# ON GRADIENT DESCENT CONVERGENCE BEYOND THE EDGE OF STABILITY

## ABSTRACT

Gradient Descent (GD) is a powerful workhorse of modern machine learning thanks to its scalability and efficiency in high-dimensional spaces. Its ability to find local minimisers is only guaranteed for losses with Lipschitz gradients, where it can be seen as a 'bona-fide' discretisation of an underlying gradient flow. Yet, many ML setups involving overparametrised models do not fall into this problem class, which has motivated research beyond the so-called "Edge of Stability" (EoS), where the step-size crosses the admissibility threshold inversely proportional to the Lipschitz constant above. Perhaps surprisingly, GD has been empirically observed to still converge regardless of local instability and oscillatory behavior.

The incipient theoretical analysis of this phenomena has mainly focused in the overparametrised regime, where the effect of choosing a large learning rate may be associated to a 'Sharpness-Minimisation' implicit regularisation within the manifold of minimisers, under appropriate asymptotic limits. In contrast, in this work we directly examine the conditions for such unstable convergence, focusing on simple, yet representative, learning problems. Specifically, we characterize a local condition involving third-order derivatives that stabilizes oscillations of GD above the EoS, and leverage such property in a teacher-student setting, under population loss. Finally, focusing on Matrix Factorization, we establish a non-asymptotic 'Local Implicit Bias' of GD above the EoS, whereby quasi-symmetric initializations converge to symmetric solutions — where sharpness is minimum amongst all minimisers.

## 1 INTRODUCTION

Given a differentiable objective function $f(\theta)$, where $\theta \in \mathbb{R}^d$ is a high-dimensional parameter vector, the most basic and widely used optimization method is gradient descent (GD), defined as

$$\theta^{(t+1)} = \theta^{(t)} - \eta \nabla_\theta f(\theta^{(t)}), \tag{1}$$

where $\eta$ is the learning rate. For all its widespread application across many different ML setups, a basic question remains: what are the convergence guarantees (even to a local minimiser) under typical objective functions, and how they depend on the (only) hyperaparameter $\eta$? In the modern context of large-scale ML applications, an additional key question is not only to understand whether or not GD converges to minimisers, but to *which* ones, since overparametrisation defines a whole manifold of global minimisers, all potentially enjoying drastically different generalisation performance.

The sensible regime to start the analysis is $\eta \to 0$, where GD inherits the local convergence properties of the Gradient Flow ODE via standard arguments from numerical integration. However, in the early phase of training, a large learning rate has been observed to result in better generalization (LeCun et al., 2012; Bjorck et al., 2018; Jiang et al., 2019; Jastrzebski et al., 2021), where the extent of "large" is measured by comparing the learning rate $\eta$ and the curvature of the loss landscape, measured with $\lambda(\theta) := \lambda_{\max}\left[\nabla_\theta^2 f(\theta)\right]$, the largest eigenvalue of the Hessian with respect to learnable parameters. Although one requires $\sup_\theta \lambda(\theta) < 2/\eta$ to guarantee the convergence of GD (Bottou et al., 2018) to (local) minimisers [1], the work of (Cohen et al., 2020) noticed a remarkable phenomena in the context

---

[1] One can replace the uniform curvature bound by $\sup_{\theta; f(\theta) \leq f(\theta^{(0)})} \lambda(\theta)$.

of neural network training: even in problems where $\lambda(\theta)$ is unbounded (as in NNs), for a fixed $\eta$, the curvature $\lambda(\theta^{(t)})$ increases along the training trajectory (1), bringing $\lambda(\theta^{(t)}) \geq 2/\eta$ (Cohen et al., 2020). After that, a surprising phenomena is that $\lambda(\theta^{(t)})$ *stably* hovers above $2/\eta$ and the neural network still eventually achieves a decreasing training loss — the so-called "Edge of Stability". We would like to understand and analyse the conditions of such convergence with a large learning rate under a variety models that capture such observed empirical behavior.

Recently, some works have built connections between EoS and implicit bias (Arora et al., 2022; Lyu et al., 2022; Damian et al., 2021; 2022) in the context of large, overparametrised models such as neural networks. In this setting, GD is expected to converge to a manifold of minimisers, and the question is to what extent a large learning rate 'favors' solutions with small curvature. In essence, these works show that under certain structural assumptions, GD is asymptotically tracking a continuous sharpness-reduction flow, in the limit of small learning rates. Compared with these, we study non-asymptotic properties of GD beyond EoS, by focusing on certain learning problems (*e.g.*, single-neuron ReLU networks and matrix factorization). In particular, we characterize a range of learning rates $\eta$ *above* the EoS such that GD dynamics hover around minimisers. Moreover, in the matrix factorization setup, where minimisers form a manifold with varying local curvature, our results give a non-asymptotic analogue of the 'Sharpness-Minimisation' arguments from Arora et al. (2022); Lyu et al. (2022); Damian et al. (2022).

The straightforward starting point for the local convergence analysis is via Taylor approximations of the loss function. However, in a quadratic Taylor expansion, gradient descent diverges once $\lambda(\theta) > 2/\eta$ (Cohen et al., 2020), indicating that a higher order Taylor approximation is required. By considering a 1-D function with local minima $\theta^*$ of curvature $\lambda^* = \lambda(\theta^*)$, we show that it is possible to stably oscillate around the minima with $\eta$ slightly above the threshold $2/\lambda^*$, provided its high order derivative satisfies mild conditions as in Theorem 1. A typical example of such functions is $f(x) = \frac{1}{4}(x^2 - \mu)^2$ with $\mu > 0$. Furthermore, we prove that it converges to an orbit of period 2 from a more global initialization rather than the analysis of high-order local approximation.

As it turns out, the analysis of such stable one-dimensional oscillations is sufficiently intrinsic to become useful in higher-dimensional problems. First, we leverage the analysis to a two-layer single-neuron ReLU network, where the task is to learn a teacher neuron with data on a uniform high-dimensional sphere. We show a convergence result under population loss with GD beyond EoS, where the direction of the teacher neuron can be learnt and the norms of two-layer weights stably oscillate. We then focus on matrix factorization, a canonical non-convex problem whose geometry is characterized by a manifold of minimisers having different local curvature. Our techniques allow us to establish a local, non-asymptotic implicit bias of GD beyond EoS, around certain quasi-symmetric initialization, by which the large learning rate regime 'attracts' the dynamics towards symmetric minimisers — precisely those where the local curvature is minimal. A further discussion is provided in Appendix M.

## 2 RELATED WORK

**Implicit regularization.** Due to its theoretical closeness to gradient descent with a small learning rate, gradient flow is a common setting to study the training behavior of neural networks. Barrett & Dherin (2020) suggests that gradient descent is closer to gradient flow with an additional term regularizing the norm of gradients. Through analysing the numerical error of Euler's method, Elkabetz & Cohen (2021) provides theoretical guarantees of a small gap depending on the convexity along the training trajectory. Neither of them fits in the case of our interest, because it is hard to track the parametric gap when $\eta > 1/\lambda$. For instance, in a quadratic function, the trajectory jumps between the two sides once $\eta > 1/\lambda$. Damian et al. (2021) shows that SGD with label noise is implicitly subjected to a regularizer penalizing sharp minimizers but the learning rate is constraint strictly below the edge of stability threshold.

**Balancing effect.** Du et al. (2018) proves that gradient flow automatically preserves the norms' differences between different layers of a deep homogeneous network. (Ye & Du, 2021) shows that gradient descent on matrix factorization with a constant small learning rate still enjoys the auto-balancing property. Also in matrix factorization, Wang et al. (2021) proves that gradient descent with a relatively large learning rate leads to a solution with a more balanced (perhaps not perfectly

balanced) solution while the initialization can be in-balanced. In a similar spirit, we extend their finding to a larger learning rate, with which the perfect balance may be achieved in our setting. We estimate our learning rate is strictly larger than theirs (Wang et al., 2021), where they show GD with large learning rates converges to a flat region in the interpolation manifold while the flat region w.r.t. our larger learing rate does not exists so GD is forced to wander around the flattest minima. Note that the implication of balancing effect is to get close to a flatter solution in the global minimum manifold, which may help improve generalization in some common arguments in the community.

**Edge of stability.**  Cohen et al. (2020) observes a two-stage process in gradient descent, where the first is loss curvature grows until the sharpness touches the bound $2/\eta$, and the second is the curvature hovers around the bound and training loss still decreases in a macro view regardless of local instability. Gilmer et al. (2021) reports similar observations in stochastic gradient descent and conducts comprehensive experiments of loss sharpness on learning rates, architecture choices and initialization. Lewkowycz et al. (2020) argues that gradient descent would "catapult" into a flatter region if loss landscape around initialization is too sharp.

Some concurrent works (Ahn et al., 2022; Ma et al., 2022; Arora et al., 2022; Damian et al., 2022) are also theoretically investigating the edge of stability. Ahn et al. (2022) suggests that unstable convergence happens when the loss landscape of neural networks forms a local forward-invariant set near the minima due to some ingredients, such as $\tanh$ as the nonlinear activation. Ma et al. (2022) empirically observes a multi-scale structure of loss landscape and, with it as an assumption, shows that gradient descent with different learning rates may stay in different levels. Arora et al. (2022) shows the training provably enters the edge of stability with modified gradient descent or modified loss, and then its associated flow goes to flat regions. Under mild conditions, Damian et al. (2022) proves that GD beyond EoS follows an optimization trajectory subjected to a sharpness constraint so that a flatter region is found.

**Learning a single neuron.**  Yehudai & Ohad (2020) studies necessary conditions on both the distribution and activation functions to guarantee a one-layer single student neuron aligning with the teacher neuron under gradient descent, SGD and gradient flow. Vardi et al. (2021) extends the investigation into a neuron with a bias term. Vardi & Shamir (2021) empirically studies the training dynamics of a two-layer single neuron, focusing on its implicit bias. In this work, we present a convergence analysis of a two-layer single-neuron ReLU network trained with population loss in a large learning rate beyond the edge of stability.

## 3 PROBLEM SETUP

We consider a differentiable objective function $f(\theta)$ with $\theta \in \mathbb{R}^d$, and the GD algorithm from (1).

**Definition 1.** *A differentiable function $f$ is L-gradient Lipschitz if*

$$\|\nabla f(\theta_1) - \nabla f(\theta_2)\| \le L \|\theta_1 - \theta_2\|, \quad \forall \, \theta_1, \theta_2. \tag{2}$$

The above definition is equivalent to saying that the spectral norm of the Hessian is bounded by $L$, or the *local curvature* at each point is bounded by $L$. Then $\eta$ needs to be bounded by $1/L$ in GD so that it is guaranteed to visit an approximate first-order stationary point (Nesterov, 1998). The perturbed GD requires $\eta = 1/L$ to visit an approximate second-order stationary point (Jin et al., 2021), and stochastic variants share similar assumptions (Ghadimi & Lan, 2013; Jin et al., 2021).

However, in practice, such an assumption may be violated, or even impossible to satisfy when $\|\nabla^2 f\|$ is not uniformly bounded. Cohen et al. (2020) observes that, with learning rate $\eta$ fixed, the largest eigenvalue $\lambda_1$ of the loss Hessian of a neural network is below $2/\eta$ at initialization, but it grows above the threshold along training. Such a phenomena is more obvious when the network is deeper or narrower. This reveals the non-smooth nature of the loss landscape of neural networks.

Furthermore, another observation from Cohen et al. (2020) is that once $\lambda_1 \ge 2/\eta$, the training loss starts to perturb sharply. This is not surprising because GD would diverge in a quadratic function with such a large curvature. However, despite of local instability, the training loss still decreases in a longer range of steps, during which the local curvature stays around $2/\eta$. A further phenomena is that, when GD is at the edge of stability, if the learning rate suddenly changes to a smaller value

$\eta_s < \eta$, then the local curvature quickly grows to $2/\eta_s$ — indicating the ability to 'manipulate' the local curvature by adjusting the learning rate.

Besides the analysis of GD, the local curvature itself has also received a lot of attention. Due to the nature of over-parameterization in modern neural networks, the global minimizers of the objective $f$ form a manifold of solutions. There have been active directions to understand the *implicit bias* of GD methods, namely where do they converge to in the manifold, and why some points in the manifold are more preferable than others. For the former question, it is believed that (stochastic) GD prefers flatter minima (Barrett & Dherin, 2020; Smith et al., 2021; Damian et al., 2021; Ma & Ying, 2021). For the latter, flatter minima brings better generalization (Hochreiter & Schmidhuber, 1997; Li et al., 2018; Keskar et al., 2016; Ma & Ying, 2021; Ding et al., 2022). It would be meaningful if flatter minima could be obtained via GD with a large learning rate.

More specifically, it has been shown that the eigenvalues of the hessian of a deep homogeneous network could be manipulated to infinity via rescaling the weights of each layer (Elkabetz & Cohen, 2021). Fortunately, gradient flow preserves the difference of norms across layers along the training (Du et al., 2018). As a result, a balanced initialization induces balanced convergence, while GD would break this balancing effect due to finite learning rate. However, recently it has been observed that GD with large learning rates enjoys a balancing effect (Wang et al., 2021), where it converges to a (not perfect) balanced result despite of inbalanced initialization.

Motivated by the connections of optimization, loss landscape and generalization, we would like to understand the training behavior of gradient descent with a large learning rate, from low-dimensional to representative models.

## 4 STABLE OSCILLATION ON 1-D FUNCTIONS

**Definition 2.** *(Period-2 stable oscillation.) Consider GD on a function $f$ in domain $\Omega$. Denote the update rule of GD as $F(x)$ for $x \in \Omega$. A period-2 stable oscillation is $\exists\, x \in \Omega$ such that $F(F(x)) = x$ and $x$ is not a minima of $f$.*

We initiate our analysis of the stable oscillation phenomenon in 1-D. Starting from a condition on general 1-D functions, we look into several specific 1-D functions to verify our arguments. Then, focusing on a function in the form of $f(x) = (x^2 - \mu)^2$, we present the convergence analysis as a foundation for the following discussions. Furthermore, to shed light on the multi-layer setting, we propose a balancing effect to make a connection to the 1-D analysis, as shown in Appendix A.1.

**General 1-D functions.** Consider a 1-D function $f(x)$ with a learnable parameter $x \in \mathbb{R}$. The parameter updates following GD with the learning rate $\eta$ as

$$x^{(t+1)} := x^{(t)} - \eta f'(x^{(t)}). \tag{3}$$

Assuming $f$ is differentiable and all derivatives are bounded, the function value in the next step can be approximated by

$$f(x^{(t+1)}) = f(x^{(t)}) - \eta [f'(x^{(t)})]^2 \Big( 1 - \frac{\eta}{2} f''(x^{(t)}) \Big) + o((x^{(t+1)} - x^{(t)})^2). \tag{4}$$

If $\eta < 2/f''(x^{(t)})$, this approximation reveals that the function monotonically decreases for each step of GD, ignoring higher terms. Such an assumption would guarantee the convergence to a global minimum in a convex function. However, our interest is what happens if $\eta > 2/f''(x)$. For instance, if $f$ is a quadratic function, the second-order derivative $f''$ is constant. As a result, once $\eta > 2/f''$, GD diverges except when being initialized at the optimum. However, when trained with a large learning rate $\eta > 2/f''(\bar{x})$, there is still some hope for a function to stay around a local minima $\bar{x}$, as stated in the following theorem.

**Theorem 1.** *Consider any 1-D differentiable function $f(x)$ around a local minima $\bar{x}$, satisfying (i) $f^{(3)}(\bar{x}) \neq 0$, and (ii) $3[f^{(3)}]^2 - f'' f^{(4)} > 0$ at $\bar{x}$. Then, there exists $\epsilon$ with sufficiently small $|\epsilon|$ and $\epsilon \cdot f^{(3)} > 0$ such that: for any point $x_0$ between $\bar{x}$ and $\bar{x} - \epsilon$, there exists a learning rate $\eta$ such that the update rule $F_\eta$ of GD satisfies $F_\eta(F_\eta(x_0)) = x_0$, and*

$$\frac{2}{f''(\bar{x})} < \eta < \frac{2}{f''(\bar{x}) - \epsilon \cdot f^{(3)}(\bar{x})}.$$

The details of proof are presented in the Appendix C. As stated in the Theorem 1, we provide a necessary condition that allows GD to stably oscillate around a local minima. But still we cannot tell whether or not some functions allow it with $f^{(3)}(\bar{x}) = 0$. For instance, a quadratic function does not satisfy this condition since $f^{(3)} = f^{(4)} \equiv 0$ and it diverges when GD is beyond the edge of stability. For $f(x) = \sin(x)$ around $\bar{x} = -\frac{\pi}{2}$ where $f^{(3)}(\bar{x}) = 0$, it turns out the sine function allows stable oscillation. Therefore, we extend the argument in Theorem 1 to a higher order case in Lemma 1.

**Lemma 1.** *Consider any 1-D differentiable function $f(x)$ around a local minima $\bar{x}$, satisfying that the lowest order non-zero derivative (except the $f''$) at $\bar{x}$ is $f^{(k)}(\bar{x})$ with $k \geq 4$. Then, there exists $\epsilon$ with sufficiently small $|\epsilon|$ such that: for any point $x_0$ between $\bar{x}$ and $\bar{x} - \epsilon$, and*

    *1. if $k$ is odd and $\epsilon \cdot f^{(k)}(\bar{x}) > 0$, $f^{(k+1)}(\bar{x}) < 0$, then there exists $\eta \in (\frac{2}{f''}, \frac{2}{f'' - f^{(k)}\epsilon^{k-2}})$,*

    *2. if $k$ is even and $f^{(k)}(\bar{x}) < 0$, then there exists $\eta \in (\frac{2}{f''}, \frac{2}{f'' + f^{(k)}\epsilon^{k-2}})$,*

*such that: the update rule $F_\eta$ of GD satisfies $F_\eta(F_\eta(x_0)) = x_0$.*

With Lemma 1, we can verify the sine function to allow stable oscillation as in Corollary 1, because its lowest order of nonzero derivative (except $f''$) at the local minima is $f^{(4)}(\bar{x}) < 0$. Meanwhile, Theorem 1 provides a guarantee that squared-loss on any function $g$ provably allows stable oscillation once $g$ satisfies some mild conditions, as stated below.

**Lemma 2.** *Consider a 1-D function $g(x)$, and define the loss function $f$ as $f(x) = (g(x) - y)^2$. Assuming (i) $g'$ is not zero when $g(\bar{x}) = y$, (ii) $g'(\bar{x})g^{(3)}(\bar{x}) < 6[g''(\bar{x})]^2$, then it satisfies the condition in Theorem 1 or Lemma 1 to allow period-2 stable oscillation around $\bar{x}$.*

This setup covers generic non-linear least squares problems, including the base model $g$ being sine, tanh, high-order monomial, exponential, logarithm, sigmoid, softplus, gaussian, etc. The proof details for these settings of $g(x)$ are provided as Corollaries 1-8 in Appendix D and E. Moreover, we provide a straightforward method to build a more complicated model from two simple base models, as follows.

**Proposition 1** (Composition Rule for Stable Oscillation). *Consider two 1-D functions $p, q$. Assume both $p(x), q(y)$ at $x = \bar{x}, y = p(\bar{x})$ satisfies the conditions of $g$ in Lemma 2 to allow stable oscillations. Then $q(p(x))$ allows stable oscillation around $x = \bar{x}$.*

Proof details of the above lemmas and proposition are presented in the Appendix D and E. Next we are going to present a careful analysis on $g(x) = x^2$.

**A special 1-D function.** Consider $f(x) = \frac{1}{4}(x^2 - \mu)^2$ with $\mu > 0$, $f^{(3)}(\sqrt{\mu}) = 6\sqrt{\mu}$, $f''(\sqrt{\mu}) = 2\mu$. Note that this function is more special to us because it can be viewed as a *symmetric scalar factorization* problem subjected to the squared loss. Later we will leverage it to gain insights for asymmetric initialization, two-layer single-neuron networks and matrix factorization. Before that, we would like to show where it converges to when $\eta > \frac{2}{f''(\sqrt{\mu})}$ as follows.

**Theorem 2.** *For $f(x) = \frac{1}{4}(x^2 - \mu)^2$, consider GD with $\eta = K \cdot \frac{1}{\mu}$ where $1 < K < \sqrt{4.5} - 1 \approx 1.121$, and initialized on any point $0 < x_0 < \sqrt{\mu}$. Then it converges to an orbit of period 2, except for a measure-zero initialization where it converges to $\sqrt{\mu}$. More precisely, the period-2 orbit are the solutions $x = \delta_1 \in (0, \sqrt{\mu}), x = \delta_2 \in (\sqrt{\mu}, 2\sqrt{\mu})$ of solving $\delta$ in*

$$\eta = \frac{1}{\delta^2 \left( \sqrt{\frac{\mu}{\delta^2} - \frac{3}{4}} + \frac{1}{2} \right)}. \tag{5}$$

The details of proof are presented in the Appendix F. As shown above, Theorem 1 and Theorem 2 stand in two different levels: Theorem 1 restricts the discussion in a local view because of Taylor approximation, while Theorem 2 starts from local convergence and then generalizes it into a global view. However, Theorem 1 builds a foundation for Theorem 2 because the latter would degenerate to the former when $K$ is extremely close to 1.

A natural follow-up question is what implications Theorem 2 brings, because 1-D is far from the practice of neural networks that contain multi-layer structures, nonlinearity and high dimensions. We precisely incorporate two layers and nonlinearity in Section 5, and high dimensions in Section 6.

## 5  ON A TWO-LAYER SINGLE-NEURON HOMOGENEOUS NETWORK

We denote a two-layer single-neuron network as $f(x;\theta) = v \cdot \sigma(w^\top x)$ where $v \in \mathbb{R}, w \in \mathbb{R}^d$, the set of trained parameters $\theta = (v, w^\top) \in \mathbb{R}^{d+1}$, and the nonlinearity $\sigma$ is ReLU. We will keep such an order in $\theta$ to view it as a vector. The input $x \in \mathbb{R}^d$ is drawn uniformly from a unit sphere $\mathcal{S}^{d-1}$. The parameters are trained by GD subjected to $L_2$ population loss, as

$$\theta_{t+1} = \theta_t - \eta \nabla_\theta L(\theta_t), \quad L(\theta_t) = \mathbb{E}_{x \in \mathcal{S}^{d-1}}\big(f(x;\theta_t) - y\big)^2.$$

We generate labels from a single teacher neuron function, as $y|x = \sigma(\tilde{w}^\top x)$. Hence $\tilde{w}$ is our target neuron to learn. We denote the angle between $w$ and $\tilde{w}$ as $\alpha \geq 0$. Note that $\alpha$ is set as non-negative because the loss function is symmetric w.r.t. the angle. Moreover, the rotational symmetry of the population data distribution results in a loss landscape that only depends on $w$ through the angle $\alpha$ and the norm $\|w\|$. Indeed, from the definition, we have

$$\nabla_\theta L = \frac{1}{d} \begin{bmatrix} v \|w\|_2^2 - \frac{\|w\|}{\pi}\big(\sin\alpha + (\pi - \alpha)\cos\alpha\big)\|\tilde{w}\| \\ v^2 w - \frac{v}{\pi}(\pi - \alpha + \frac{1}{2}\sin 2\alpha)\cdot\tilde{w} - \frac{v}{\pi}(-\frac{1}{2}\cos 2\alpha + \frac{1}{2})\|\tilde{w}\|\,\tilde{w}_\perp \end{bmatrix},$$

where we denote $\tilde{w}_\perp$ as the normalized of $w - \text{proj}_{\tilde{w}} w$. Consider the Hessian

$$H \triangleq \begin{bmatrix} \partial_v^2 L & \partial_w \partial_v L \\ \partial_v \partial_w L & \partial_w^2 L \end{bmatrix} \stackrel{\text{if } vw = \tilde{w}}{=\!=\!=\!=} \frac{1}{d}\begin{bmatrix} \|w\|^2 & vw^\top \\ vw & v^2\mathbb{I} \end{bmatrix} \in \mathbb{R}^{(d+1)\times(d+1)}. \tag{6}$$

Hence, in the global minima manifold where $vw = \tilde{w}$, the eigenvalues of the Hessian are $\lambda_1 = \frac{\|w\|^2 + v^2}{d}, \lambda_{2\ldots d} = \frac{v^2}{d}, \lambda_{d+1} = 0$. Therefore, the largest eigenvalue $\lambda_1$ measures the imbalance (i.e., $|\|w\| - v|$) between the two layers again as $\lambda_1 = \frac{(\|w\|-v)^2 + 2\|\tilde{w}\|}{d}$ similar to the 2-D case in (15) in Appendix A.1. So we would like to investigate where GD converges if $\eta > \frac{2}{2\|\tilde{w}\|/d} = d/\|\tilde{w}\|$ that is too large even for the flattest minima. Note that a key difference between the current case and the previous 2-D analysis is that the current one includes a neuron as a vector and a nonlinear ReLU unit.

From the second row of $\nabla_\theta L$, which is $\nabla_w L$, it is clear that updates of $w$ always stay in the plane spanned by $\tilde{w}$ and $w^{(0)}$. Hence, this problem can be simplified to three variables $(v, w_x, w_y)$ with the target neuron $\tilde{w} = [1, 0]$. The three variables stand for

$$v^{(t)} := v^{(t)}, \quad w_x^{(t)} := \text{proj}_{\tilde{w}} w^{(t)}, \quad w_y^{(t)} := \text{proj}_{\tilde{w}_\perp} w^{(t)} = \sqrt{\left\|w^{(t)}\right\|^2 - (w_x^{(t)})^2}.$$

We keep $w_y$ as nonnegative because the loss $L$ is invariant to its sign and our previous notation $\alpha \geq 0$ requires a non-negative $w_y$. Then we show that $w_y$ decays to 0 as follows.

**Theorem 3.** *In the above setting, consider a teacher neuron $\tilde{w} = [1, 0]$ and set the learning rate $\eta = Kd$ with $K \in (1, 1.1]$. Initialize the student as $\left\|w^{(0)}\right\| = v^{(0)} \triangleq \epsilon \in (0, 0.10]$ and $\langle w^{(0)}, \tilde{w}\rangle \geq 0$. Then, for $t \geq T_1 + 4$, $w_y^{(t)}$ decays as*

$$w_y^{(t)} < 0.1 \cdot (1 - 0.030K)^{t-T_1-4}, \quad T_1 \leq \left\lceil \log_{2.56}\frac{1.35}{\pi\beta^2}\right\rceil, \quad \beta = \left(1 + \frac{1.1}{\pi}\right)\epsilon.$$

**Proof sketch**  The details of proof are presented in the Appendix I. The proof is divided into two stages, depending on whether $w_y$ grows or not. The key is that the change of $w_y$ follows (omitting all superscripts $t$)

$$\frac{\Delta w_y}{w_y} \propto -vw_x + \frac{1}{\pi}\frac{\frac{w_y}{w_x}}{1 + (\frac{w_y}{w_x})^2}, \quad w_y^{(t+1)} = |w_y + \Delta w_y|. \tag{7}$$

where the second term in $\Delta w_y/w_y$ is bounded in $[0, \frac{1}{2\pi}]$. In stage 1 where $vw_x$ is relatively small, we show the growth ratio of $w_y$ is smaller than those of $w_x$ and $vw_x$, resulting in an upper bound of number of iterations for $vw_x$ to reach $\frac{1}{2\pi}$, so $\max(w_y)$ is bounded too. Although the initialization is balanced as $v^{(0)} = \left\|w^{(0)}\right\|$ for simplicity of proof, $v - w_x$ is also bounded at the end of stage 1. From the beginning of stage 2, thanks to the relatively narrow range of $K$, we are able to compute the

bounds of three variables (including $v - w_x$, $v w_x$ and $w_y$) and they turn out to fall into a basin in the parameter space after four iterations. In this basin, $w_y$ decays exponentially with a linear rate of 0.97 at most. □

With the guarantee of $w_y$ decaying in the above theorem, the dynamics of the single-neuron ReLU network follow the convergence of the 2-D case in Appendix A.1, with a convergence result as follows.

**Proposition 2.** *The single-neuron model in Theorem 3 converges to a period-2 orbit where $w_y = 0$ and $(v, w_x) \in \gamma_K$ with $\gamma_K = \{(\delta_1, \delta_1), (\delta_2, \delta_2)\}$. Here $\delta_1 \in (0, 1), \delta_2 \in (1, 2)$ are the solutions $\delta$ in*

$$K = \frac{1}{\delta^2 \left( \sqrt{\frac{1}{\delta^2} - \frac{3}{4}} + \frac{1}{2} \right)}. \tag{8}$$

*Remark.* Actually this convergence is close to the flattest minima because: if the learning rate decays to infinitesimal after sufficient oscillations, then the trajectory walks towards the flattest minima ($v = w_x = 1, w_y = 0$).

To summarize, the single-neuron model goes through three phases of training dynamics, with a intialization of the angle $\angle(w, \tilde{w})$ as $\frac{\pi}{2}$ at most. First, the angle decreases monotonically but, due to the growth of norms, the absolute deviation $w_y$ still increases. Meanwhile, the inbalance $v - w_x$ stays in a bounded level. Second, $w_y$ starts to decrease and the parameters fall into a basin within four steps. Third, in the basin, $w_y$ decreases exponentially and, after $w_y$ at a reasonable low level, the model approximately follows the dynamic of the 2-D case and the inbalance $v - w_x$ decreases as well, following Theorem 5. The model converges to a period-2 orbit as in the 1-D case in Theorem 2.

## 6 QUASI-SYMMETRIC MATRIX FACTORIZATION: WALKING TOWARDS FLATTEST MINIMA

Consider a matrix factorization problem, parameterized by learnable weights $\mathbf{X} \in \mathbb{R}^{m \times p}, \mathbf{Y} \in \mathbb{R}^{q \times p}$, and the target matrix is $\mathbf{C} \in \mathbb{R}^{m \times q}$. The loss $L$ is defined as

$$L(\mathbf{X}, \mathbf{Y}) = \frac{1}{2} \left\| \mathbf{X} \mathbf{Y}^\top - \mathbf{C} \right\|_F^2. \tag{9}$$

Obviously $\{\mathbf{X}, \mathbf{Y} : \mathbf{X} \mathbf{Y}^\top = \mathbf{C}\}$ forms a minimum manifold. In this context, the question is to describe GD dynamics in terms of a 'descent' phase (i.e., reaching the manifold), followed by a 'hovering' phase, where the dynamics evolve nearby the minimum manifold. Although we prove that the necessary 1-D condition holds around minimum as Theorem 6 (in Appendix A.2), it is more attracting to investigate GD in high dimensions.

A straightforward subsets of the "flattest" points in the manifold of minimisers are in fact given by symmetric matrices, *i.e.,* points of the form $(\mathbf{X}, \mathbf{X})$ with $\mathbf{X} \mathbf{X}^\top = \mathbf{C}$. As it turns out, the local behavior of GD beyond EoS in this symmetric submanifold of minimisers can be explicitly analysed. Indeed, Theorem 7 (in Appendix A.2) shows that the dynamics follows the direction of the leading eigenvector and then stably oscillates with a period-2 analogous to the 1D case in Theorem 2. Note that, although $\{\mathbf{X} : \mathbf{X} \mathbf{X}^\top = \mathbf{X}_0 \mathbf{X}_0^\top\}$ forms a manifold that contains infinite number of minimizers [2], all of them have the same sharpness due to the same leading singular values. So a natural follow-up question is to analyse minimizers with different sharpness.

The simplest setting that contains minimizers of varying-sharpness is to rescale symmetric minimizers, leading to *Quasi-symmetric* Matrix Factorization. Given a symmetric target $\mathbf{C} = \mathbf{X}_0 \mathbf{X}_0^\top$, assume that we are around the (global) minima $\mathbf{Y}_1 = \alpha \mathbf{X}_0 + \Delta \mathbf{Y}_1, \mathbf{Z}_1 = \frac{1}{\alpha} \mathbf{X}_0 + \Delta \mathbf{Z}_1$ with $\alpha > 0$ and small deviation $\|\Delta \mathbf{Y}_1\|, \|\Delta \mathbf{Z}_1\| \leq \epsilon$. The top singular value and vectors in SVD of $\mathbf{X}_0$ is $\sigma_1 u_1 v_1^\top$. Then the EoS-learning rate at $(\alpha \mathbf{X}_0, \frac{1}{\alpha} \mathbf{X}_0)$ is $\frac{2}{\sigma_1^2 (\alpha^2 + \frac{1}{\alpha^2})}$, which is largest as $\frac{1}{\sigma_1^2}$ at $\alpha = 1$. We study the convergence of GD starting from $\mathbf{Y}_1 = \alpha \mathbf{X}_0 + \Delta \mathbf{Y}_1, \mathbf{Z}_1 = \frac{1}{\alpha} \mathbf{X}_0 + \Delta \mathbf{Z}_1$ with learning rate $\eta = \frac{1}{\sigma_1^2} + \beta, \ \beta > 0$. The following theorem shows that, although starting nearby a sharper minima, GD still converges to and stably scillate around the flattest one.

---

[2] in particular, it contains the orbit $\{X_0 U; U \in \mathcal{O}(p)\}$

**Theorem 4.** *Consider the above quasi-symmetric matrix factorization with learning rate $\eta = \frac{1}{\sigma_1^2} + \beta$. Assume $0 < \beta\sigma_1^2 < \sqrt{4.5} - 1 \approx 0.121$.* *Consider a minimum $(\mathbf{Y}_0 = \alpha\mathbf{X}_0, \mathbf{Z}_0 = 1/\alpha\mathbf{X}_0), \alpha > 0$.* *The initialization is around the minimum, as $\mathbf{Y}_1 = \mathbf{Y}_0 + \Delta\mathbf{Y}_1, \mathbf{Z}_1 = \mathbf{Z}_0 + \Delta\mathbf{Z}_1$, with the deviations satisfying $u_1^\top \Delta\mathbf{Y}_1 v_1 \neq 0, u_1^\top \Delta\mathbf{Z}_1 v_1 \neq 0$ and $\|\Delta\mathbf{Y}_1\|, \|\Delta\mathbf{Z}_1\| \leq \epsilon$. The second largest singular value of $\mathbf{X}_0$ needs to satisfy*

$$\max\left\{\eta\frac{\sigma_1^2}{\alpha^2}\left(1 + \alpha^4\frac{\sigma_2^2}{\sigma_1^2}\right), \eta\sigma_1^2\alpha^2\left(1 + \frac{\sigma_2^2}{\alpha^4\sigma_1^2}\right)\right\} \leq 2. \tag{10}$$

*Then GD would converge to a period-2 orbit $\gamma_\eta$ approximately with error in $\mathcal{O}(\epsilon)$, formally written as*

$$(\mathbf{Y}_t, \mathbf{Z}_t) \to \gamma_\eta + (\Delta\mathbf{Y}, \Delta\mathbf{Z}), \qquad \|\Delta\mathbf{Y}\|, \|\Delta\mathbf{Z}\| = \mathcal{O}(\epsilon), \tag{11}$$

$$\gamma_\eta = \left\{\left(\mathbf{Y}_0 + (\rho_i - \alpha)\,\sigma_1 u_1 v_1^\top, \mathbf{Z}_0 + (\rho_i - 1/\alpha)\,\sigma_1 u_1 v_1^\top\right)\right\}, \qquad (i = 1, 2) \tag{12}$$

*where $\rho_1 \in (1, 2), \rho_2 \in (0, 1)$ are the two solutions of solving $\rho$ in*

$$1 + \beta\sigma_1^2 = \frac{1}{\rho^2\left(\sqrt{\frac{1}{\rho^2} - \frac{3}{4}} + \frac{1}{2}\right)}. \tag{13}$$

**Proof sketch**  Details of proof can be found in Appendix J.3, which shares a similar spirit with Theorem 7. The analysis consists of two phases, depending on whether $\epsilon_{y,t} \triangleq \langle\mathbf{Y}_t - \mathbf{Y}_0, u_1 v_1^\top\rangle, \epsilon_{z,t} \triangleq \langle\mathbf{Z}_t - \mathbf{Z}_0, u_1 v_1^\top\rangle$ are small or not. In Phase I, all components of $\mathbf{Y}_t - \mathbf{Y}_0$ and $\mathbf{Z}_t - \mathbf{Z}_0$ are small due to the initialization near minima, but both $\epsilon_{y,t}$ and $\epsilon_{z,t}$ are growing exponentially in a rate of $\eta\sigma_1^2\alpha^2 + \eta\frac{\sigma_1^2}{\alpha^2} - 1 \geq 2\eta\sigma_1^2 - 1 > 1$. In Phase II, both $\epsilon_{y,t}$ and $\epsilon_{z,t}$ are much larger than other components, as long as other components are still not growing. So the dynamics of them matches GD of 2-D function $f(y, z) = \frac{1}{2}(yz - 1)^2$ with learning rate $\eta' = 1 + \beta\sigma_1^2$. Following the 2-D analysis in Theorem 5, we have $\epsilon_{y,t}$ and $\epsilon_{z,t}$ converge to the same values, which degenerates the 2-D problem to 1-D function. Therefore, the proof concludes with 1-D convergence analysis of $f(x) = \frac{1}{4}(x^2 - 1)^2$ as shown in Theorem 2. □

*Remark.* Note that both $\mathbf{Y}_0 - \alpha \cdot \sigma_1 u_1 v_1^\top, \mathbf{Z}_0 - 1/\alpha \cdot \sigma_1 u_1 v_1^\top$ are residuals of $\mathbf{Y}_0, \mathbf{Z}_0$ with the top singular value eliminated. Then, compared with Theorem 7, we have $\rho_i$ corresponds to $\delta_i + 1$, which means both symmetric and quasi-symmetric cases converge to parameters with the same top singular values and wander around the flattest minima. In other words, this convergence is close to the flattest minima because: if the learning rate decays to infinitesimal after sufficient oscillations, then the trajectory walks towards the flattest minima approximately with parameter distance in $\mathcal{O}(\epsilon)$. Also note that, if $\eta < \frac{1}{\sigma_1^2}$, we anticipate it still escapes from the sharp minima and converges to a flatter one (not necessarily the flattest). The result could be obtained by tracking GD on $f(x, y) = \frac{1}{2}(xy - 1)^2$ with $\eta < 1$ slightly. But the closed form can not be expressed explicitly, because it strongly depends on initialization.

## 7 NUMERICAL EXPERIMENTS

In this section, we provide numerical experiments to verify our theorems. Additional experiments on 2-D functions, MLP and MNIST can be found in Appendix B.

**1-D functions.**  As discussed in the Section 4, we have $f(x) = \frac{1}{4}(x^2 - 1)^2$ satisfying the condition in Theorem 1 and $g(x) = 2\sin(x)$ satisfying Lemma 1, so we estimate that both $f$ and $g$ allow stable oscillation around the local minima. It turns out GD stably oscillates around the local minima on both functions, when $\eta > \frac{2}{f''(\bar{x})}$ slightly, as shown in Figure 1.

**Two-layer single-neuron model.**  As discussed in the Section 5, with a learning rate $\eta \in (d, 1.1d]$, a single-neuron network $f(x) = v \cdot \sigma(w^\top x)$ is able to align with the direction of the teacher neuron under population loss. We train such a model in empirical loss on 1000 data points uniformly sampled from a sphere $\mathcal{S}^1$, as shown in Figure 2. The student neuron is initialized orthogonal to the teacher

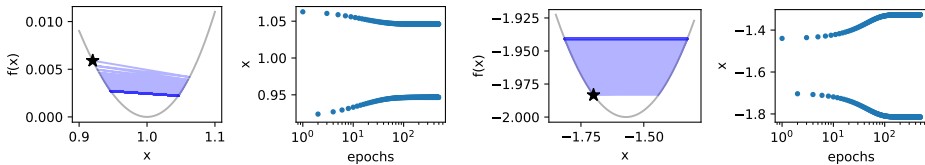

Figure 1: Running GD around the local minima of $f(x) = \frac{1}{4}(x^2 - 1)^2$ (left) and $f(x) = 2\sin(x)$ (right) with learning rate $\eta = 1.01 > \frac{2}{f''(\bar{x})} = 1$. Stars denote the start points. It turns out both functions allow stable oscillation around the local minima.

neuron. In the end of training, $w_y$ decays to a small value before the inbalance $|v - w_x|$ decays sharply, which verifies our argument in Section 5. With a small $w_y$, this nonlinear problem degenerates to a 2-D problem on $v, w_x$. Then, the balanced property makes it align with the 1-D problem where $v$ and $w_x$ converge to a period-2 orbit. Note that the small residuals of $|v - w_x|$ and $w_y$ are due to the difference between population loss and empirical loss.

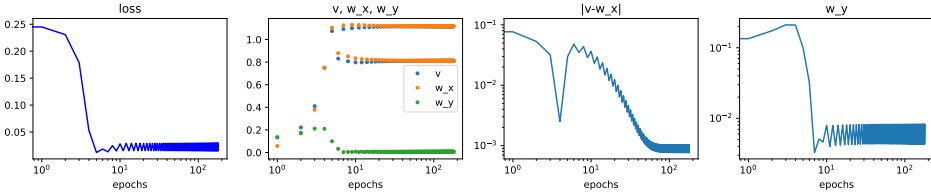

Figure 2: Running GD in the teacher-student setting with learning rate $\eta = 2.2 = 1.1d$, trained on 1000 points uniformly sampled from sphere $\mathcal{S}^1$ of $\|x\| = 1$. The teacher neuron is $\tilde{w} = [1, 0]$ and the student neuron is initialized as $w^{(0)} = [0, 0.1]$ with $v^{(0)} = 0.1$.

**Symmetric and quasi-symmetric matrix factorization.**    As discussed in the Section 6 and Appendix A.2, with mild assumptions, both symmetric and quasi-symmetric cases stably wanders around the flattest minima. We train GD on a matrix factorization problem with $\mathbf{X}_0\mathbf{X}_0^\top = \mathbf{C} \in \mathbb{R}^{8 \times 8}$. The learning rate is $1.02\times$ EoS threshold. Following the setting in Section 6, for symmetric case, the training starts near $\mathbf{X}_0$ and, for quasi-symmetric case, it starts near $(\alpha\mathbf{X}_0, 1/\alpha\mathbf{X}_0)$ with $\alpha = 0.8$, as shown in Figure 3. Although starting with a re-scaling, the quasi-symmetric case achieves the same top singular values in $\mathbf{Y}$ and $\mathbf{Z}$, which verifies the balancing effect of 2-D functions in Theorem 5. Then, the top singular values of both cases converge to the same period-2 orbit, supported by Theorem 2, 4 and 7.

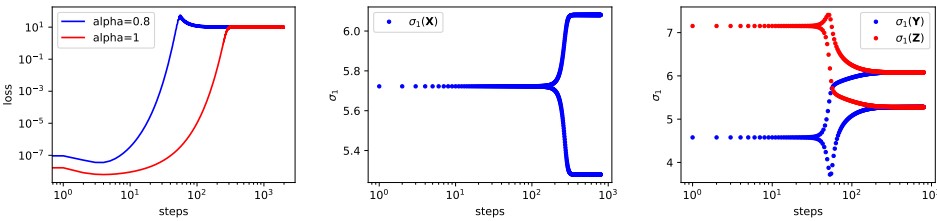

Figure 3: Symmetric and Quasi-symmetric Matrix factorization: running GD around flat ($\alpha = 1$) and sharp ($\alpha = 0.8$) minima. In both cases, their leading singular values converge to the same period-2 orbit (about 6.1 and 5.3). (Left: Training loss. Middle: Largest singular value of symmetric case. Right: Largest singular values of quasi-symmetric case.)

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

# Appendices

# A    ADDITIONAL RESULTS

## A.1    ON A 2-D FUNCTION

Similar to $f(x) = \frac{1}{4}(x^2 - \mu)^2$, consider a 2-D function $f(x, y) = \frac{1}{2}(xy - \mu)^2$. Apparently, if $x$ and $y$ initialize as the same, then $(x^{(t)}, y^{(t)})$ would always align with the 1-D case from the same initialization. Therefore, it is significant to analyze this problem under different initialization for $x$ and $y$, which we would call "in-balanced" initialization. Meanwhile, another giant difference is that all the global minima in 2-D case form a manifold $\{(x, y)|xy = \mu\}$ while the 1-D case only has two points of global minima. It would be great if we could understand which points in the global minima manifold, or in the whole parameter space, are preferable by GD.

Note that reweighting the two parameters would manipulate the curvature to infinity as in (Elkabetz & Cohen, 2021), so the inbalance strongly affects the local curvature. Viewing $f(x)$ as a symmetric scalar factorization problem, we treat $f(x, y)$ as asymmetric scalar factorization. The update rule of GD is

$$x^{(t+1)} := x^{(t)} - \eta(x^{(t)}y^{(t)} - \mu)y^{(t)}, \quad y^{(t+1)} := y^{(t)} - \eta(x^{(t)}y^{(t)} - \mu)x^{(t)}. \tag{14}$$

Consider the Hessian as

$$H \triangleq \begin{bmatrix} \partial_x^2 f & \partial_y \partial_x f \\ \partial_x \partial_y f & \partial_y^2 f \end{bmatrix} = \begin{bmatrix} y^2 & 2xy - \mu \\ 2xy - \mu & x^2 \end{bmatrix}. \tag{15}$$

When $xy = \mu$, the eigenvalues of $H$ are $\lambda_1 = x^2 + y^2, \lambda_2 = 0$. Note that $\lambda_1 = (x - y)^2 + 2\mu$. Hence, in the global minima manifold, the local curvature of each point is larger if its two parameters are more inbalanced. Among all these points, the smallest curvature appears to be $\lambda_1 = 2\mu$ when $x = y = \sqrt{\mu}$. In other words, if the learning rate $\eta > 2/2\mu$, all points in the manifold would be too sharp for GD to converge. We would like to investigate the behavior of GD in this case. It turns out the two parameters are driven to a perfect balance although they initialized differently, as follows.

**Theorem 5.** *For $f(x, y) = \frac{1}{2}(xy - \mu)^2$, consider GD with learning rate $\eta = K \cdot \frac{1}{\mu}$. Assume both $x$ and $y$ are always positive during the whole process $\{x_i, y_i\}_{i \geq 0}$. In this process, denote a series of all points with $xy > \mu$ as $\mathcal{P} = \{(x_i, y_i)|x_i y_i > \mu\}$. Then $|x - y|$ decays to 0 in $\mathcal{P}$, for any $1 < K < 1.5$.*

**Proof sketch**    The details of proof are presented in the Appendix G. Start from a point $(x^{(t)}, y^{(t)})$ where $x^{(t)}y^{(t)} > \mu$. Because $y^{(t+1)} - x^{(t+1)} = (y^{(t)} - x^{(t)})(1 + \eta(x^{(t)}y^{(t)} - \mu))$, it suffices to show

$$\left| \frac{y^{(t+2)} - x^{(t+2)}}{y^{(t)} - x^{(t)}} \right| = |(1 + \eta(x^{(t)}y^{(t)} - \mu))(1 + \eta(x^{(t+1)}y^{(t+1)} - \mu))| < 1. \tag{16}$$

Since $1 + \eta(x^{(t)}y^{(t)} - \mu) > 1$, the analysis of $1 + \eta(x^{(t+1)}y^{(t+1)} - \mu)$ is divided into three cases considering the coupling of $(x^{(t)}, y^{(t)}), (x^{(t+1)}, y^{(t+1)})$.    □

*Remark.* Actually, for a larger $K \geq 1.5$, it is possible for GD to converge to an inbalanced orbit. For instance, Figure 15 in (Wang et al., 2021) shows inbalanced orbits for $f(x) = \frac{1}{2}(xy - 1)^2$ with $K = 1.9$.

Combining with the fact that the probability of GD converging to a stationary point that has sharpness beyond the edge of stability is zero (Ahn et al., 2022), Theorem 5 reveals $x$ and $y$ would converge to a perfect balance. Note that this balancing effect is different from that of gradient flow (Du et al., 2018), where the latter states that gradient flow preserves the difference of norms of different layers along training. As a result, in gradient flow, inbalanced initialization induces inbalanced convergence, while in our case inbalanced-initialized weights converge to a perfect balance. Furthermore, Theorem 5 shows an effect that the two parameters are squeezed to a single variable, which re-directs to our 1-D analysis in Theorem 2. Therefore, actually both cases converge to the same orbit when $1 < K < 1.121$, as stated in Prop 3. Numerical results are presented in Figure 4.

**Proposition 3.** *Following the setting in Theorem 5. Further assume $1 < K < \sqrt{4.5} - 1 \approx 1.121$. Then GD converges to an orbit of period 2. The orbit is formally written as $\{(x = y = \delta_i)|i = 1, 2\}$, with $\delta_1 \in (0, \sqrt{\mu}), \delta_2 \in (\sqrt{\mu}, 2\sqrt{\mu})$ as the solutions of solving $\delta$ in*

$$\eta = \frac{1}{\delta^2 \left( \sqrt{\frac{\mu}{\delta^2} - \frac{3}{4}} + \frac{1}{2} \right)}.$$

*Remark.* Actually this convergence is close to the flattest minima because: if the learning rate decays to infinitesimal after sufficient oscillations, then the trajectory walks towards the flattest minima.

However, one thing to notice is that the inbalance at initialization needs to be bounded in Theorem 5 because both $x$ and $y$ are assumed to stay positive along the training. More precisely, we have

$$x^{(t+1)}y^{(t+1)} = x^{(t)}y^{(t)}(1 - \eta(x^{(t)}y^{(t)} - \mu))^2 - \eta(x^{(t)}y^{(t)} - \mu)(x^{(t)} - y^{(t)})^2, \quad (17)$$

and then $x^{(t+1)}y^{(t+1)} < 0$ when $|x^{(t)} - y^{(t)}|$ is large with $x^{(t)}y^{(t)} > \mu$ fixed. Therefore, we provide a condition to guarantee both $x, y$ positive as follows, with details presented in the Appendix H.

**Lemma 3.** *In the setting of Theorem 5, denote the initialization as $m = \frac{|y_0 - x_0|}{\sqrt{\mu}}$ and $x_0 y_0 > \mu$. Then, during the whole process, both $x$ and $y$ will always stay positive, denoting $p = \frac{4}{(m + \sqrt{m^2 + 4})^2}$ and $q = (1 + p)^2$, if*

$$\max\left\{\eta(x_0 y_0 - \mu), \frac{4}{27}(1 + K)^3 + \left(\frac{2}{3}K^2 - \frac{1}{3}K + \frac{qK^2}{2(K+1)}m^2\right)qm^2 - K\right\} < p.$$

### A.2 ON MATRIX FACTORIZATION

In this section, we present two additional results of matrix factorization.

#### A.2.1 ASYMMETRIC CASE: 1D FUNCTION AT THE MINIMA

Before looking into the theorem, we would like to clarify the definition of the loss Hessian. Inherently, we squeeze $\mathbf{X}, \mathbf{Y}$ into a vector $\theta = \text{vec}(\mathbf{X}, \mathbf{Y}) \in \mathbb{R}^{mp+pq}$, which vectorizes the concatnation. As a result, we are able to represent the loss Hessian w.r.t. $\theta$ as a matrix in $\mathbb{R}^{(mp+pq) \times (mp+pq)}$. Meanwhile, the support of the loss landscape is in $\mathbb{R}^{mp+pq}$. Similarly, we use $(\Delta\mathbf{X}, \Delta\mathbf{Y})$ in the same shape of $(\mathbf{X}, \mathbf{Y})$ to denote . In the following theorem, we are to show the leading eigenvector $\Delta \triangleq \text{vec}(\Delta\mathbf{X}, \Delta\mathbf{Y}) \in \mathbb{R}^{mp+pq}$ of the loss Hessian. Since the cross section of the loss landscape and $\Delta$ forms a 1D function $f_\Delta$, we would also show the stable-oscillation condition on 1D function holds at the minima of $f_\Delta$.

**Theorem 6.** *For a matrix factorization problem, assume $\mathbf{XY} = \mathbf{C}$. Consider SVD of both matrices as $\mathbf{X} = \sum_{i=1}^{\min\{m,p\}} \sigma_{x,i} u_{x,i} v_{x,i}^\top$ and $\mathbf{Y} = \sum_{i=1}^{\min\{p,q\}} \sigma_{y,i} u_{y,i} v_{y,i}^\top$, where both groups of $\sigma_{\cdot,i}$'s are in descending order and both top singular values $\sigma_{x,1}$ and $\sigma_{y,1}$ are unique. Also assume $v_{x,1}^\top u_{y,1} \neq 0$. Then the leading eigenvector of the loss Hessian is $\Delta = \text{vec}(C_1 u_{x,1} u_{y,1}^\top, C_2 v_{x,1} v_{y,1}^\top)$ with $C_1 = \frac{\sigma_{y,1}}{\sqrt{\sigma_{x,1}^2 + \sigma_{y,1}^2}}, C_2 = \frac{\sigma_{x,1}}{\sqrt{\sigma_{x,1}^2 + \sigma_{y,1}^2}}$. Denote $f_\Delta$ as the 1D function at the cross section of the loss landscape and the line following the direction of $\Delta$ passing $\text{vec}(\Delta\mathbf{X}, \Delta\mathbf{Y})$. Then, at the minima of $f_\Delta$, it satisfies*

$$3[f_\Delta^{(3)}]^2 - f_\Delta^{(2)} f_\Delta^{(4)} > 0. \quad (18)$$

The proof is provided in Appendix J.1. This theorem aims to generalize our 1-D analysis into higher dimension, and it turns out the 1-D condition is sastisfied around any minima for two-layer matrix factorization. In Theorem 1 and Lemma 1, if such 1-D condition holds, there must exist a period-2 orbit around the minima for GD beyond EoS. However, this is not straightforward to generalize to high dimensions, because 1) directions of leading eigenvectors and (nearby) gradient are not necessarily aligned, and 2) it is more natural and practical to consider initialization *in any direction* around the minima instead of strictly along leading eigenvectors. Therefore, below we present a convergence analysis with initialization near the minima, but in any direction instead.

#### A.2.2 SYMMETRIC CASE: CONVERGENCE ANALYSIS AROUND THE MINIMA

In this section, we focus on the symmetric case of matrix factorization where $\mathbf{Y} = \mathbf{X}^\top$. Accordingly, we rescale the loss function as $L(\mathbf{X}, \mathbf{X}) = \frac{1}{4}\|\mathbf{XX}^\top - \mathbf{C}\|_F^2$. Denote the target as $\mathbf{C} = \mathbf{X}_0\mathbf{X}_0^\top$, and

assume we are around the minima $\mathbf{X}_1 = \mathbf{X}_0 + \Delta\mathbf{X}_1$ with small $\|\Delta\mathbf{X}_1\| \le \epsilon$. Consider SVD as $\mathbf{X}_1 = \sum_{i=1}^{\min\{m,p\}} \sigma_i u_i v_i^\top \triangleq \sigma_1 u_1 v_1^\top + \widetilde{\mathbf{X}}_0$. Then the EoS-learning-rate threshold at $\mathbf{X} = \mathbf{X}_0$ is $\eta = \frac{1}{\sigma_1^2}$. Therefore, we are to show the convergence of GD starting from $\mathbf{X}_1 = \mathbf{X}_0 + \Delta\mathbf{X}_1$ with learning rate $\eta = \frac{1}{\sigma_1^2} + \beta$ where $\beta > 0$.

**Theorem 7.** *Consider the above symmetric matrix factorization with learning rate $\eta = \frac{1}{\sigma_1^2} + \beta$. Assume $0 < \beta\sigma_1^2 < \sqrt{4.5} - 1 \approx 1.121$ and $\eta\sigma_2^2 < 1$. The initialization is around the minimum, as $\mathbf{X}_1 = \mathbf{X}_0 + \Delta\mathbf{X}_1$, with the deviation satisfying $u_1^\top \Delta\mathbf{X}_1 v_1 \ne 0$ and $\|\Delta\mathbf{X}_1\| \le \epsilon$ bounded by a small value. Then GD would converge to a period-2 orbit $\gamma_\eta$ approximately by a small margin in $O(\epsilon)$, formally written as*

$$\mathbf{X}_t \to \gamma_\eta + \Delta\mathbf{X}, \quad \|\Delta\mathbf{X}\| = O(\epsilon), \tag{19}$$

$$\gamma_\eta = \{\mathbf{X}_0 + \delta_1\sigma_1 u_1 v_1^\top, \mathbf{X}_0 + \delta_2\sigma_1 u_1 v_1^\top\}, \tag{20}$$

*where $\delta_1 \in (0,1), \delta_2 \in (-1,0)$ are the two solutions of solving $\delta$ in*

$$1 + \beta\sigma_1^2 = \frac{1}{(\delta+1)^2\left(\sqrt{\frac{1}{(\delta+1)^2} - \frac{3}{4}} + \frac{1}{2}\right)}. \tag{21}$$

**Proof sketch** The proof is provided in Appendix J.2. The analysis consists of two phases, depending on whether $\epsilon_t \triangleq \langle \mathbf{X}_t - \mathbf{X}_0, u_1 v_1^\top \rangle$ is small or not. In Phase I, all components of $\mathbf{X}_t - \mathbf{X}_0$ are small due to the initialization near minima, but only $\epsilon_t$ is growing exponentially in a rate of $1 + \beta\sigma_1^2$. In Phase II, $\epsilon_t$ is much larger than other components, as long as other components are still not growing. So the dynamics of $\epsilon_t$ matches GD of 1-D function $f(\alpha) = \frac{1}{4}((\alpha + \sigma_1)^2 - \sigma_1^2)^2$ with learning rate $\eta' = \frac{1}{\sigma_1^2} + \beta > \frac{2}{f''(0)} = \frac{1}{\sigma_1^2}$. The proof concludes with 1-D convergence analysis of $f(x) = \frac{1}{4}(x^2 - \mu)^2$ as shown in Theorem 2. □

*Remark.* Theorem 7 assumes GD starts from any point in an $\epsilon$-ball near the minima, except $u_1^\top \Delta\mathbf{X}_1 v_1 \ne 0$. Note that this exception is with Lebesgue measure zero and, even if it is unfortunately satisfied, GD still has a chance to reach $u_1^\top \Delta\mathbf{X}_t v_1 \ne 0$ after several steps due to some higher-order small noise. Then, this assumption could be relaxed as $\|\Delta\mathbf{X}_1\| \ne 0$.

# B ADDITIONAL EXPERIMENTS

## B.1 2-D FUNCTION

As discussed in the Section A.1, on the function $f(x,y) = \frac{1}{2}(xy - 1)^2$, we estimate that $|x - y|$ decays to 0 when $\eta \in (1, 1.5)$, as shown in Figure 4. Since it achieves a perfect balance, the two parameters follows convergence of the corresponding 1-D function $f(x) = \frac{1}{4}(x^2 - 1)^2$. As shown in Figure 4, $xy$ with $\eta = 1.05$ converges to a period-2 orbit, as stated in the 1-D discussion of Theorem 2 while $xy$ with $\eta = 1.25$ converges to a period-4 orbit, which is out of our range in the theorem. But still it falls into the range for balance in Theorem 5.

## B.2 HIGH DIMENSION AND MNIST

We perform two experiments in relatively higher dimension settings. We are to show two observations that coincides with our discussions in the low dimension:

**Observation 1**: GD beyond EoS drives to flatter minima.

**Observation 2**: GD beyond EoS is in a similar style with the low dimension.

### B.2.1 2-LAYER HIGH-DIM HOMOGENEOUS RELU NNS WITH PLANTED TEACHER NEURONS

We conduct a synthetic experiment in the high-dimension teacher-student framework. The teacher network is in the form of

$$y|x \coloneqq f_{\text{teacher}}(x; \tilde{\theta}) = \sum_{i=1}^{16} \mathsf{ReLU}(\mathbf{e}_i^\top x), \tag{22}$$

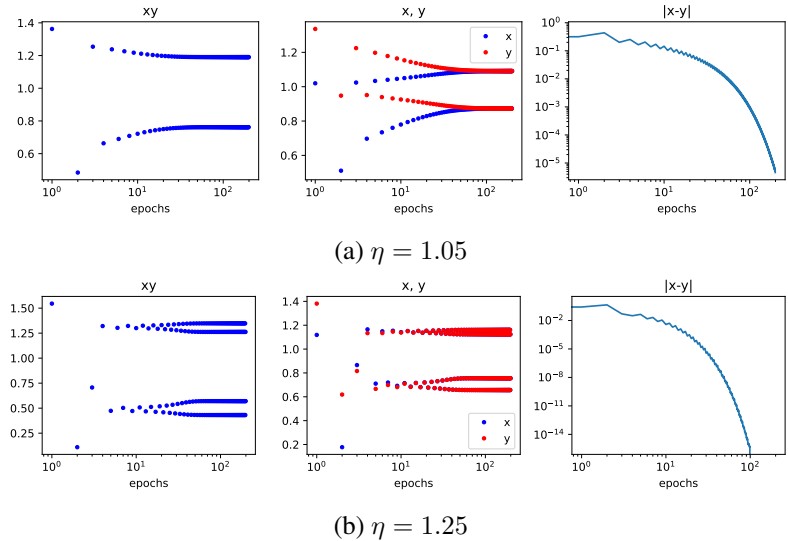

Figure 4: Running GD on $f(x, y) = \frac{1}{2}(xy - 1)^2$ with learning rate $\eta = 1.05$ (top) and $\eta = 1.25$ (bottom). When $\eta = 1.05$, it converges to a period-2 orbit. When $\eta = 1.25$, it converges to a period-4 orbit. In both cases, $|x - y|$ decays sharply.

where $x \in \mathbb{R}^{16}$ and $\mathbf{e_i}$ is the $i$-th vector in the standard basis of $\mathbb{R}^{16}$. The student and the loss are in forms of

$$f(x; \theta) = \sum_{i=1}^{16} v_i \cdot \mathsf{ReLU}(w_i^\top x), \tag{23}$$

$$L(\theta; \tilde{\theta}) = \frac{1}{m} \sum_{i}^{16} \left( f(x; \theta) - y|x_i \right)^2. \tag{24}$$

Apparently, the global minimum manifold contains the following set $\mathcal{M}$ as (w.l.o.g., ignoring any permutation)

$$\mathcal{M} = \{(v_i, w_i)_{i=1}^{16} \mid \forall i \in [16], w_i = k_i \cdot \mathbf{e}_i, v_i = \frac{1}{k_i}, k_i > 0\}. \tag{25}$$

However, different choices of $\{k_i\}_{i=1}^{16}$ induce different extents of sharpness around each minima. Our aim is to show that **GD with a large learning rate beyond the edge of stability drives to the flattest minima from sharper minima**.

**Initialization.** We initialize all student neurons directionally aligned with the teachers as $w_i \parallel \mathbf{e}_i$ but choose various $k_i$, as $k_i = 1 + 0.0625(i - 1)$. Obviously, such a choice of $\{k_i\}_{i=1}^{16}$ is not at the flattest minima, due to the isotropy of teacher neurons. Also we add small noise to $w_i$ to make the training start closely (but not exactly) from a sharp minima, as

$$w_i = k_i \cdot (\mathbf{e}_i + 0.01\epsilon), \quad \epsilon \sim \mathcal{N}(0, I). \tag{26}$$

**Data.** We uniformly sample 10000 data points from the unit sphere $\mathcal{S}^{15}$.

**Training.** We run gradient descent with two learning rates $\eta_1 = 0.5, \eta_2 = 2.6$. Later we will show with experiments that the EoS threshold of learning rate is around 2.5, so $\eta_2$ is beyond the edge of stability. GD with these two learning rates starts from the same initialization for 100 epochs. Then we extend another 20 epochs with learning rate decay to 0.5 from 2.6 for the learning-rate case.

**Results.** All results are provided in Figure 5. Both Figure 5 (a, b) present the gap between these two trajectories, where GD with a small learning rate stays around the sharp minima, while that with a larger one drives to flatter minima. Then GD stably oscillates around the flatter minima.

Meanwhile, from Figure 5 (b), when we decrease the learning rate from 2.6 to 0.5 after 100 epochs, GD converges to a nearby minima which is significantly flatter, compared with that of lr=0.5.

Figure 5 (c) provides a more detailed view of $\frac{\|w_i\|}{v_i}$ for all 16 neurons. All neurons with lr=0.5 stay at the original ratio $k_i^2$. But those with lr=2.6 all converge to the same ratio around $k^2 = \frac{\|w\|}{v} = 1.21$, as shown in Figure 5 (d). We compute the relationship between the sharpness of global minima in $\mathcal{M}$ and different choices of $k$, as shown in Figure 5 (e, f). Actually, $k^2 = 1.21$ is the best choice of $\{k_i\}_{i=1}^{16}$ such that the minima is the flattest.

Therefore, we have shown that, **in such a setting of high-dimension teacher-student network, GD beyond the edge of stability drives to the flattest minima**.

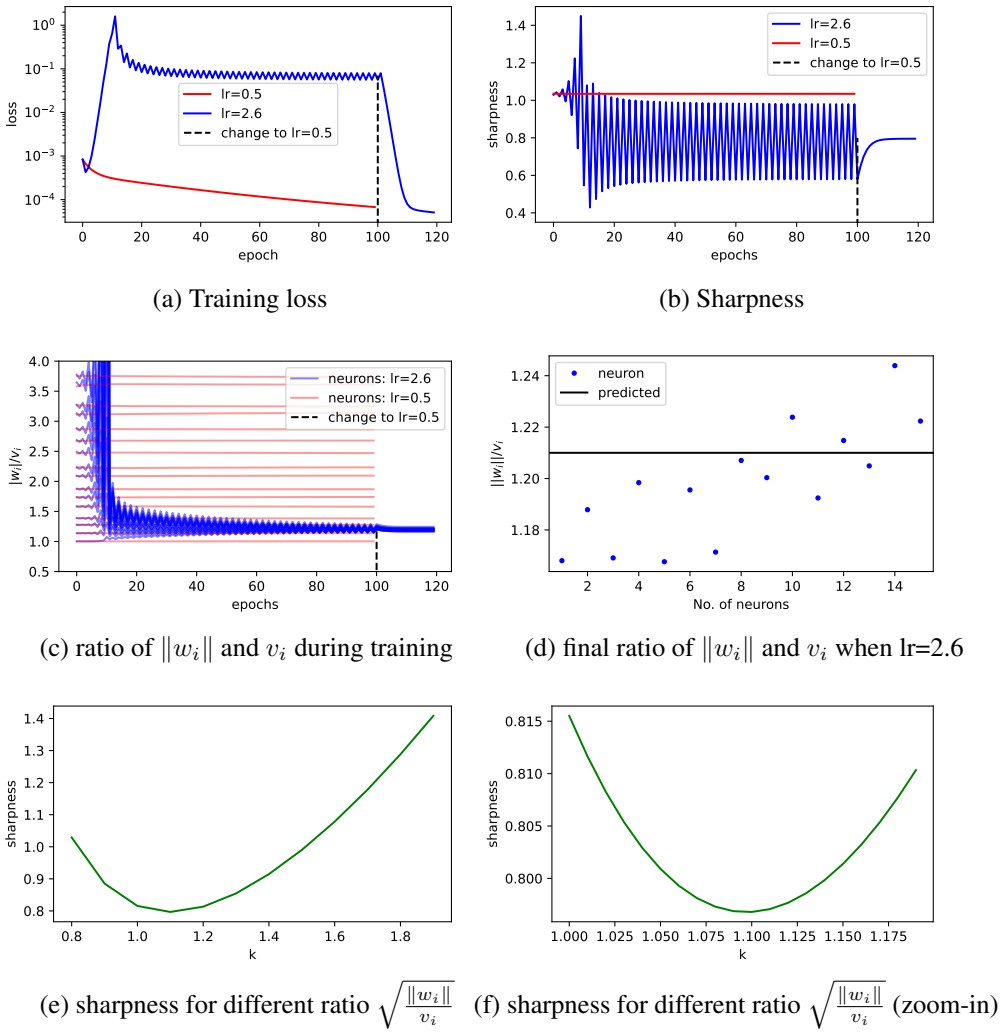

(a) Training loss

(b) Sharpness

(c) ratio of $\|w_i\|$ and $v_i$ during training

(d) final ratio of $\|w_i\|$ and $v_i$ when lr=2.6

(e) sharpness for different ratio $\sqrt{\frac{\|w_i\|}{v_i}}$

(f) sharpness for different ratio $\sqrt{\frac{\|w_i\|}{v_i}}$ (zoom-in)

Figure 5: Result of 2-layer 16-neuron teacher-student experiment.

### B.2.2   3, 4, 5-LAYER NON-HOMOGENEOUS MLPS ON MNIST

We conduct an experiment on real data to show that **our finding in the low-dimension setting in Theorem 1 is possible to generalize to high-dimensional setting**. More precisely, our goals are to show, when GD is beyond EoS,

1. the oscillation direction (gradient) aligns with the top eigenvector of Hessian.

2. the 1D function at the cross-section of oscillation direction and high-dim loss landscape satisfies the conditions in Theorem 1.

**Network, dataset and training.**   We run 3, 4, 5-layer ReLU MLPs on MNIST LeCun et al. (1998). The networks have 16 neurons in each layer. To make it easier to compute high-order derivatives, we simplify the dataset by 1) only using 2000 images from class 0 and 1, and 2) only using significant input channels where the standard deviation over the dataset is at least 110, which makes the network input dimension as 79. We train the networks using MSE loss subjected to GD with large learning rates $\eta = 0.5, 0.4, 0.35$ and a small rate $\eta = 0.1$ (for 3-layer). Note that the larger ones are beyond EoS.

**Definition 3** (line search minima). *Consider a function $f$, learning rate $\eta$ and a point $x \in domain(f)$. We call $\tilde{x}$ as the line search minima of $x$ if*

$$\tilde{x} = x - c^* \cdot \eta \nabla f(x), \tag{27}$$

$$c^* = argmin_{c \in [0,1]} \, f\left(x - c \cdot \eta \nabla f(x)\right). \tag{28}$$

The line search minima $\tilde{x}$ can interpreted as the lowest point on the 1D function induced by the gradient at $x$. If GD is beyond EoS, $\tilde{x}$ stays in the valley below the oscillation of $x$.

**Results.** All results are presented in Figure 6, 7 and 8.

Take the 3-layer as an example. From Figure 6 (a, b), GD is beyond EoS during epochs 10-14 and 21-60. For these epochs, cosine similarity between the top Hessian eigenvector $v_1$ and the gradient is pretty close to 1, as shown in Figure 6 (c), which verifies our goal 1.

In Figure 6 (d), we compute $3[f^{(3)}]^2 - f^{(2)}f^{(4)}$ at line search minima along training, which is required to be positive in Theorem 1 to allow stable oscillation. Then it turns out most points have $3[f^{(3)}]^2 - f^{(2)}f^{(4)} > 0$ except a few points, all of which are not in the EoS regime, and these few exceptional points might be due to approximation error to compute the fourth-order derivative since their negativity is quite small. This verifies our goal 2.

Both the above arguments are the same in the cases of 4 and 5 layers as shown in Figure 7 and 8.

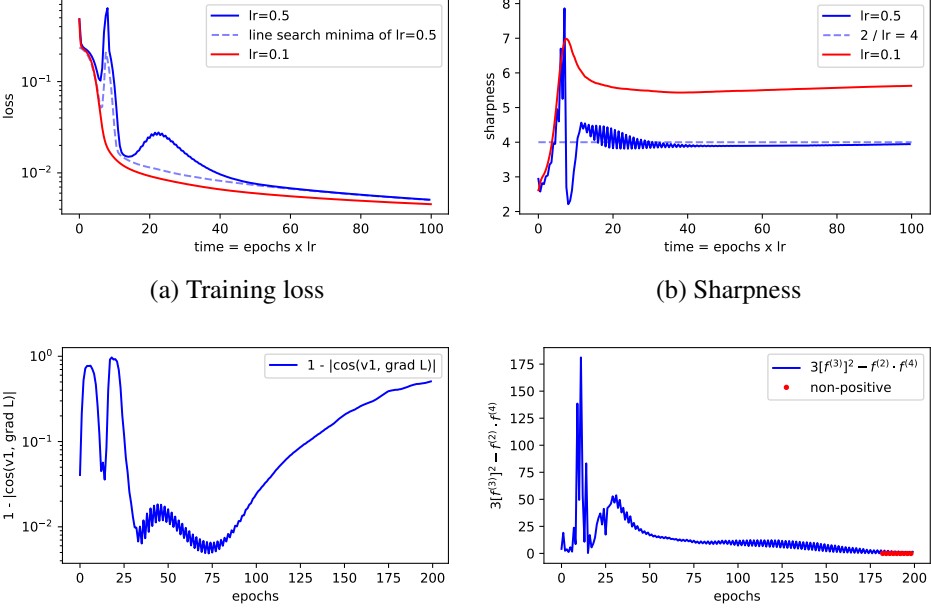

(a) Training loss

(b) Sharpness

(c) similarity of gradient and top eig-vector $v_1$

(d) $3[f^{(3)}]^2 - f^{(2)}f^{(4)}$ at line search minima

Figure 6: Result of **3-layer** ReLU MLPs on MNIST. Both (c) and (d) are for learning rate as 0.5.

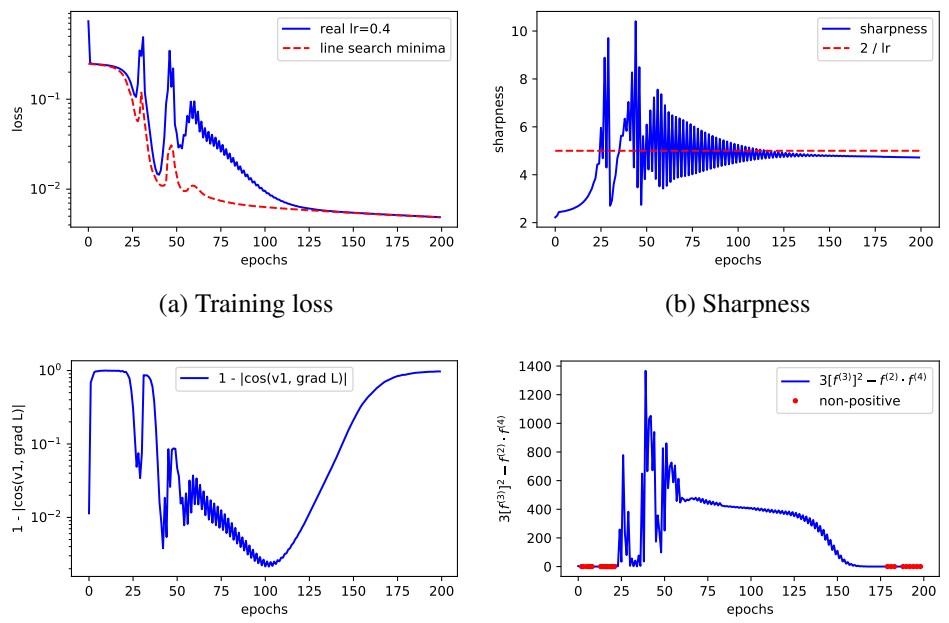

(a) Training loss

(b) Sharpness

(c) similarity of gradient and top eig-vector $v_1$

(d) $3[f^{(3)}]^2 - f^{(2)}f^{(4)}$ at line search minima

Figure 7: Result of **4-layer** ReLU MLPs on MNIST.

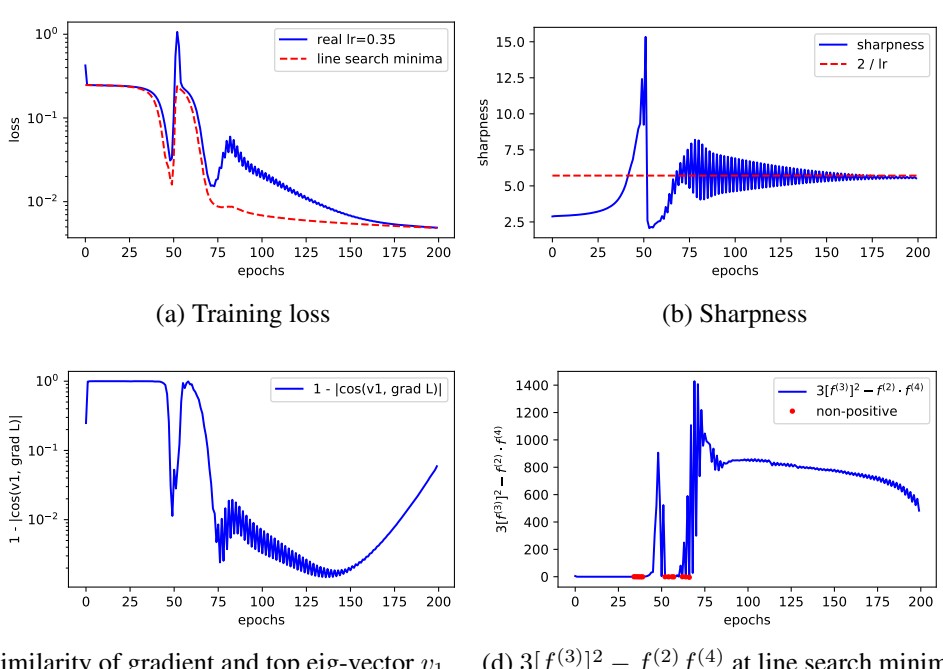

(a) Training loss

(b) Sharpness

(c) similarity of gradient and top eig-vector $v_1$

(d) $3[f^{(3)}]^2 - f^{(2)}f^{(4)}$ at line search minima

Figure 8: Result of **5-layer** ReLU MLPs on MNIST.

## C    PROOF OF THEOREM 1

**Theorem 8** (Restatement of Theorem 1). *Consider any 1-D differentiable function $f(x)$ around a local minima $\bar{x}$, satisfying (i) $f^{(3)}(\bar{x}) \neq 0$, and (ii) $3[f^{(3)}]^2 - f'' f^{(4)} > 0$ at $\bar{x}$. Then, there exists $\epsilon$ with sufficiently small $|\epsilon|$ and $\epsilon \cdot f^{(3)} > 0$ such that: for any point $x_0$ between $\bar{x}$ and $\bar{x} - \epsilon$, there exists a learning rate $\eta$ such that the update rule $F_\eta$ of GD satisfies $F_\eta(F_\eta(x_0)) = x_0$, and*

$$\frac{2}{f''(\bar{x})} < \eta < \frac{2}{f''(\bar{x}) - \epsilon \cdot f^{(3)}(\bar{x})}.$$

*Proof.* For simplicity, we assume $f^{(3)}(\bar{x}) > 0$. Imagine a starting point $x_0 = \bar{x} - \epsilon, \epsilon > 0$. We omit $f'(\bar{x}), f''(\bar{x}), f^{(3)}(\bar{x}), f^{(4)}(\bar{x})$ as $f', f'', f^{(3)}, f^{(4)}$. After running two steps of gradient descent, we have

$$x_0 = \bar{x} - \epsilon,$$

$$f'(x_0) = f' - f''\epsilon + \frac{1}{2}f^{(3)}\epsilon^2 - \frac{1}{6}f^{(4)}\epsilon^3 + \mathcal{O}(\epsilon^4)$$

$$= -f''\epsilon + \frac{1}{2}f^{(3)}\epsilon^2 - \frac{1}{6}f^{(4)}\epsilon^3 + \mathcal{O}(\epsilon^4),$$

$$x_1 = x_0 - \eta f'(x_0) = \bar{x} - \epsilon - \eta\left(-f''\epsilon + \frac{1}{2}f^{(3)}\epsilon^2 - \frac{1}{6}f^{(4)}\epsilon^3\right) + \mathcal{O}(\epsilon^4),$$

$$f'(x_1) = f'' \cdot (x_1 - \bar{x}) + \frac{1}{2}f^{(3)} \cdot (x_1 - \bar{x})^2 + \frac{1}{6}f^{(4)} \cdot (x_1 - \bar{x})^3 + \mathcal{O}(\epsilon^4),$$

$$x_2 = x_1 - \eta f'(x_1),$$

$$\frac{x_2 - x_0}{\eta} = -\left(-f''\epsilon + \frac{1}{2}f^{(3)}\epsilon^2 - \frac{1}{6}f^{(4)}\epsilon^3\right) - f'' \cdot \left(-\epsilon - \eta\left(-f''\epsilon + \frac{1}{2}f^{(3)}\epsilon^2 - \frac{1}{6}f^{(4)}\epsilon^3\right)\right)$$

$$- \frac{1}{2}f^{(3)}\left(-\epsilon - \eta\left(-f''\epsilon + \frac{1}{2}f^{(3)}\epsilon^2 - \frac{1}{6}f^{(4)}\epsilon^3\right)\right)^2 - \frac{1}{6}f^{(4)} \cdot (-\epsilon - \eta(-f''\epsilon))^3 + \mathcal{O}(\epsilon^4)$$

$$= (2f'' - \eta f'' f'')\epsilon + \left(-\frac{1}{2}f^{(3)} + \frac{1}{2}\eta f'' f^{(3)} - \frac{1}{2}f^{(3)}(-1 + \eta f'')^2\right)\epsilon^2$$

$$+ \left(\frac{1}{6}f^{(4)} - \frac{1}{6}\eta f'' f^{(4)} + \frac{1}{2}(-1 + \eta f'')\eta f^{(3)} f^{(3)} - \frac{1}{6}(-1 + \eta f'')^3 f^{(4)}\right)\epsilon^3 + \mathcal{O}(\epsilon^4).$$

When $\eta = \frac{2}{f''}$, it holds

$$\frac{x_2 - x_0}{\eta} = \left(\frac{1}{2}\eta f^{(3)} f^{(3)} - \frac{1}{3}f^{(4)}\right)\epsilon^3 + \mathcal{O}(\epsilon^4), \tag{29}$$

which would be positive if $\frac{1}{2}\eta f^{(3)} f^{(3)} - \frac{1}{3}f^{(4)} = \frac{1}{3f''}(3[f^{(3)}]^2 - f'' f^{(4)}) > 0$ and $|\epsilon|$ is sufficiently small.

When $\eta = \frac{2}{f'' - \epsilon \cdot f^{(3)}}$ then $\eta f'' = 2 + 2\frac{f^{(3)}}{f''}\epsilon + \mathcal{O}(\epsilon^2)$, it holds

$$\frac{x_2 - x_0}{\eta} = -2f^{(3)}\epsilon^2 + \left(-\frac{1}{2}f^{(3)} + f^{(3)} - \frac{1}{2}f^{(3)}\right)\epsilon^2 + \mathcal{O}(\epsilon^3) = -2f^{(3)}\epsilon^2 + \mathcal{O}(\epsilon^3), \tag{30}$$

which is negative when $|\epsilon|$ is sufficiently small.

Therefore, there exists a learning rate $\eta \in (\frac{2}{f''}, \frac{2}{f'' - \epsilon \cdot f^{(3)}})$ such that $x_2 = x_0$ due to the continuity of $(x_2 - x_0)$ with respect to $\eta$.

The above proof can be generalized to the case of $x_0 = \bar{x} - \epsilon'$ with $\epsilon' \in (0, \epsilon]$ and the learning rate is still bounded as $\eta \in (\frac{2}{f''}, \frac{2}{f'' - \epsilon \cdot f^{(3)}})$. $\square$

## D    PROOF OF LEMMA 1

**Lemma 4** (Restatement of Lemma 1). *Consider any 1-D differentiable function $f(x)$ around a local minima $\bar{x}$, satisfying that the lowest order non-zero derivative (except the $f''$) at $\bar{x}$ is $f^{(k)}(\bar{x})$ with*

$k \geq 4$. *Then, there exists $\epsilon$ with sufficiently small $|\epsilon|$ such that: for any point $x_0$ between $\bar{x}$ and $\bar{x} - \epsilon$, and*

1. *if $k$ is odd and $\epsilon \cdot f^{(k)}(\bar{x}) > 0$, $f^{(k+1)}(\bar{x}) < 0$, then there exists $\eta \in (\frac{2}{f''}, \frac{2}{f'' - f^{(k)} \epsilon^{k-2}})$,*

2. *if $k$ is even and $f^{(k)}(\bar{x}) < 0$, then there exists $\eta \in (\frac{2}{f''}, \frac{2}{f'' + f^{(k)} \epsilon^{k-2}})$,*

*such that: the update rule $F_\eta$ of GD satisfies $F_\eta(F_\eta(x_0)) = x_0$.*

*Proof.* (1) If $k$ is odd, assuming $f^{(k)} > 0$ for simplicity, we have

$$x_0 = \bar{x} - \epsilon,$$

$$f'(x_0) = -f'' \epsilon + \frac{1}{(k-1)!} f^{(k)} \epsilon^{k-1} - \frac{1}{k!} f^{(k+1)} \epsilon^k + \mathcal{O}(\epsilon^{k+1}),$$

$$x_1 = x_0 - \eta f'(x_0) = \bar{x} - \epsilon + \eta f'' \epsilon - \frac{1}{(k-1)!} \eta f^{(k)} \epsilon^{k-1} + \frac{1}{k!} \eta f^{(k+1)} \epsilon^k + \mathcal{O}(\epsilon^{k+1}),$$

$$f'(x_1) = f'' \cdot (x_1 - \bar{x}) + \frac{1}{(k-1)!} f^{(k)} \cdot (x_1 - \bar{x})^{k-1} + \frac{1}{k!} f^{(k+1)} \cdot (x_1 - \bar{x})^k + \mathcal{O}(\epsilon^{k+1}),$$

$$\begin{aligned}
\frac{x_2 - x_0}{\eta} &= \frac{x_1 - \eta f'(x_1) - x_0}{\eta} = -f'(x_0) - f'(x_1) \\
&= (2f'' - \eta f'' f'') \epsilon \\
&\quad + \left( -\frac{1}{(k-1)!} f^{(k)} + \frac{1}{(k-1)!} \eta f'' f^{(k)} - \frac{1}{(k-1)!} f^{(k)} \cdot (-1 + \eta f'')^{k-1} \right) \epsilon^{k-1} \\
&\quad + \left( \frac{1}{k!} f^{k+1} - \frac{1}{k!} \eta f'' f^{(k+1)} - \frac{1}{k!} f^{(k+1)} \cdot (-1 + \eta f'')^k \right) \epsilon^k + \mathcal{O}(\epsilon^{k+1})
\end{aligned}$$

When $\eta = \frac{2}{f''}$, it holds

$$\frac{x_2 - x_0}{\eta} = -\frac{2}{k!} f^{(k+1)} \epsilon^k + \mathcal{O}(\epsilon^{k+1}). \tag{31}$$

When $\eta = \frac{2}{f'' - f^{(k)} \epsilon^{k-2}}$ then $\eta f'' = 2 + 2 \frac{f^{(k)}}{f''} \epsilon^{k-2} + \mathcal{O}(\epsilon^{2k-4})$, then it holds

$$\frac{x_2 - x_0}{\eta} = -2 f^{(k)} \epsilon^{k-1} + \mathcal{O}(\epsilon^k). \tag{32}$$

Since $k$ is odd and $\epsilon \cdot f^{(k)}(\bar{x}) > 0$, $f^{(k+1)}(\bar{x}) < 0$, the above two estimations of $x_2 - x_0/\eta$ have one positive and one negative exactly. Therefore, due to the continuity of $x_2 - x_0$ wrt $\eta$, there exists a learning rate $\eta \in (\frac{2}{f''}, \frac{2}{f'' - f^{(k)} \epsilon^{k-2}})$ such that $x_2 = x_0$.

The above proof can be generalized to any $x_0$ between $\bar{x}$ and $\bar{x} - \epsilon$ with the same bound for $\eta$.

(2) If $k$ is even, we have

$$x_0 = \bar{x} - \epsilon,$$

$$f'(x_0) = -f'' \epsilon - \frac{1}{(k-1)!} f^{(k)} \epsilon^{k-1} + \mathcal{O}(\epsilon^k),$$

$$x_1 = x_0 - \eta f'(x_0) = \bar{x} - \epsilon + \eta f'' \epsilon + \frac{1}{(k-1)!} \eta f^{(k)} \epsilon^{k-1} + \mathcal{O}(\epsilon^k),$$

$$f'(x_1) = f'' \cdot (x_1 - \bar{x}) + \frac{1}{(k-1)!} f^{(k)} \cdot (x_1 - \bar{x})^{k-1} + \mathcal{O}(\epsilon^k),$$

$$\begin{aligned}
\frac{x_2 - x_0}{\eta} &= \frac{x_1 - \eta f'(x_1) - x_0}{\eta} = -f'(x_0) - f'(x_1) \\
&= (2f'' - \eta f'' f'') \epsilon \\
&\quad + \left( \frac{1}{(k-1)!} f^{(k)} - \frac{1}{(k-1)!} \eta f'' f^{(k)} - \frac{1}{(k-1)!} (-1 + \eta f'')^{k-1} \right) \epsilon^{k-1} + \mathcal{O}(\epsilon^k).
\end{aligned}$$

When $\eta = \frac{2}{f''}$, it holds

$$\frac{x_2 - x_0}{\eta} = -\frac{2}{(k-1)!}f^{(k)}\epsilon^{k-1} + \mathcal{O}(\epsilon^k).$$

When $\eta = \frac{2}{f'' + c \cdot f^{(k)}\epsilon^{k-2}}$ with $c > 0$ as some constant implying $\eta f'' = 2(1 - c\frac{f^{(k)}}{f''}\epsilon^{k-2}) + \mathcal{O}(\epsilon^{2k-4})$, then it holds

$$\frac{x_2 - x_0}{\eta} = 2\left(c - \frac{1}{(k-1)!}\right)f^{(k)}\epsilon^{k-1} + \mathcal{O}(\epsilon^k),$$

where we then set $c = 1$.

Hence, the above two estimations of $x_2 - x_0 / \eta$ have one positive and one negative with sufficiently small $|\epsilon|$. Therefore, due to the continuity of $x_2 - x_0$, there exists a learning rate $\eta \in (\frac{2}{f''}, \frac{2}{f'' + f^{(k)}\epsilon^{k-2}})$ such that $x_2 = x_0$.

The above proof can be generalized to any $x_0$ between $\bar{x}$ and $\bar{x} - \epsilon$ with the same bound for $\eta$. □

**Corollary 1.** $f(x) = \sin(x)$ *allows stable oscillation around its local minima* $\bar{x}$.

*Proof.* Its lowest order nonzero derivative (expect $f''$) is $f^{(4)}\bar{x} = \sin(\bar{x}) = -1 < 0$ and the order 4 is even. Then Lemma 1 gives the result. □

## E  PROOF OF LEMMA 2

**Lemma 5** (Restatement of Lemma 2). *Consider a 1-D function $g(x)$, and define the loss function $f$ as $f(x) = (g(x) - y)^2$. Assuming (i) $g'$ is not zero when $g(\bar{x}) = y$, (ii) $g'(\bar{x})g^{(3)}(\bar{x}) < 6[g''(\bar{x})]^2$, then it satisfies the condition in Theorem 1 or Lemma 1 to allow period-2 stable oscillation around $\bar{x}$.*

*Proof.* From the definition, we have

$$f''(x) = 2[g(x) - y]g''(x) + 2[g'(x)]^2, \tag{33}$$

$$f^{(3)}(x) = 2[g(x) - y]g^{(3)}(x) + 6g''(x)g'(x), \tag{34}$$

$$f^{(4)}(x) = 2[g(x) - y]g^{(4)}(x) + 6g''(x)g''(x) + 8g'(x)g^{(3)}(x). \tag{35}$$

Then at the global minima where $g(x) = y$, we have $f''(x) = 2[g'(x)]^2$ and $f^{(3)}(x) = 6g''(x)g'(x)$. If we assume $y$ is not a trivial value for $g(x)$, which means $g'(x) \neq 0$ at the minima, and $g$ is not linear around the minima (implies $g'' \neq 0$), then $f$ satisfies $f^{(3)}(\bar{x}) \neq 0$ in Theorem 1. Meanwhile, we need $3f^{(3)}f^{(3)} - f''f^{(4)} > 0$ as in Theorem 1, hence it requires

$$\frac{1}{2g'(x)g'(x)}36g''(x)g''(x)g'(x)g'(x) - \frac{1}{3}\left(6g''(x)g''(x) + 8g'(x)g^{(3)}(x)\right) > 0 \tag{36}$$

$$6g''(x)g''(x) > g'(x)g^{(3)}(x). \tag{37}$$

The remaining case is, if $g'(x) \neq 0$ and $g'' = 0$ at the minima, it satisfies the condition for Lemma 1 with $k = 4$, because $f^{(3)} = 0$ and $f^{(4)} < 0$ due to (35, 37) □

**Corollary 2.** $f(x) = (x^2 - 1)^2$ *allows stable oscillation around the local minima* $\bar{x} = 1$.

*Proof.* With $g(x) = x^2$, it has $g'(1) = 2 \neq 0$, $g''(1) = 2 \neq 0$. All higher order derivatives of $g$ are zero. Then Lemma 2 gives the result. □

**Corollary 3.** $f(x) = (\sin(x) - y)^2$ *allows stable oscillation around the local minima* $\bar{x} = \arcsin(y)$ *with* $y \in (-1, 1)$.

*Proof.* With $g(x) = \sin(x)$, it has $g'(\bar{x}) = \cos(\bar{x}) \neq 0$, $g^{(3)}(\bar{x}) = -\cos(\bar{x})$. We have $g^{(3)}(\bar{x})$ is bounded as $g'g^{(3)} - 6[g'']^2 = -\cos^2(\bar{x}) - 6\sin^2(\bar{x}) < 0$. Then Lemma 2 gives the result. □

**Corollary 4.** $f(x) = (\tanh(x) - y)^2$ *allows stable oscillation around the local minima* $\bar{x} = \tanh^{-1}(y)$ *with* $y \in (-1, 1)$.

*Proof.* With $g(x) = \tanh(x)$, it has $g'(\bar{x}) = \text{sech}^2(\bar{x}) \neq 0$, and $g^{(3)}(\bar{x}) = -2\text{sech}^4(\bar{x}) + 4\text{sech}^2(\bar{x})\tanh^2(\bar{x})$ is bounded as

$$g'g^{(3)} - 6[g'']^2 = -2\text{sech}^6 + 4\text{sech}^4\tanh^2 - 24\text{sech}^4\tanh^2 = -2\text{sech}^6 - 20\text{sech}^4\tanh^2 < 0.$$

Then Lemma 2 gives the result. □

**Corollary 5.** $f(x) = (x^\alpha - y)^2$ *(with* $k \in \mathbb{Z}, k \geq 2$*) allows stable oscillation around the local minima* $\bar{x} = y^{1/\alpha}$ *except* $y = 0$.

*Proof.* With $g(x) = x^\alpha$, it has $g'(\bar{x}) = \alpha x^{\alpha-1}, g''(\bar{x}) = \alpha(\alpha-1)x^{\alpha-2}, g^{(3)}(\bar{x}) = \alpha(\alpha-1)(\alpha-2)x^{\alpha-3}$. Then we have $g'g^{(3)} - 6[g'']^2 = \alpha^2(\alpha-1)(-5\alpha+4)x^{2\alpha-4} < 0$. Then Lemma 2 gives the result. □

**Corollary 6.** $f(x) = (\exp(x) - y)^2$ *allows stable oscillation around the local minima* $\bar{x} = \log y$ *for* $y > 0$.

*Proof.* With $g(x) = \exp x$, it has $g'(\bar{x}) = g''(\bar{x}) = g^{(3)}(\bar{x}) = \exp(\bar{x})$. Then we have $g'g^{(3)} - 6[g'']^2 < 0$. Then Lemma 2 gives the result. □

**Corollary 7.** $f(x) = (\log(x) - y)^2$ *allows stable oscillation around the local minima* $\bar{x} = \exp y$.

*Proof.* With $g(x) = log x$, it has $g'(\bar{x}) = \frac{1}{\bar{x}}, g''(\bar{x}) = -\frac{1}{\bar{x}^2}, g^{(3)}(\bar{x}) = -\frac{2}{\bar{x}^3}$. Then we have $g'g^{(3)} - 6[g'']^2 < 0$. Then Lemma 2 gives the result. □

**Corollary 8.** $f(x) = (\frac{1}{1+\exp(-x)} - y)^2$ *allows stable oscillation around the local minima* $\bar{x} = sigmoid^{-1}(y)$ *for* $y \in (0, 1)$.

*Proof.* With $g(x) = \frac{1}{1+\exp(-x)}$, it has $g'(\bar{x}) = \frac{\exp(-x)}{(\exp(-x)+1)^2}, g''(\bar{x}) = -\frac{\exp(x)(\exp(x)-1)}{(\exp(x)+1)^3}, g^{(3)}(\bar{x}) = \frac{\exp(x)(-4\exp(x)+\exp(2x)+1)}{(\exp(x)+1)^4}$. Then we have $g'g^{(3)} - 6[g'']^2 \propto -4\exp(x) + \exp(2x) + 1 - 6(\exp(x) - 1)^2 < 0$. Then Lemma 2 gives the result. □

**Proposition 4** (Restatement of Prop 1). *Consider two functions* $f, g$. *Assume both* $f(x), g(y)$ *at* $x = \bar{x}, y = f(\bar{x})$ *satisfies the conditions in Lemma 2 to allow stable oscillations. Then* $g(f(x))$ *allows stable oscillation around* $x = \bar{x}$.

*Proof.* Denote $F(x) \triangleq g(f(x))$. Then we have

$$F'(x) = g'(f(x))f'(x),$$
$$F''(x) = g''(f(x))[f'(x)]^2 + g'(f(x))f''(x),$$
$$F^{(3)}(x) = g^{(3)}(f(x))[f'(x)]^3 + 3g''(f(x))f'(x)f''(x) + g'(f(x))f^{(3)}(x).$$

Thus, omitting all variables $\bar{x}$ and $f(\bar{x})$ in the derivatives, it holds

$$F'(\bar{x})F^{(3)}(\bar{x}) - 6[F''(\bar{x})]^2 = g'f'\left(g^{(3)}(f')^3 + 3g''f'f'' + g'f^{(3)}\right) - 6\left(g''(f')^2 + g'f''\right)^2$$
$$\leq -9g'g''(f')^2 f'',$$

where the inequality is due to all conditions in Lemma 2. So the only problem is whether we can achieve $g'g''f'' > 0$. The good news is that, even if it holds $g'g''f'' < 0$, we can still find functions to re-represent $g(f(x))$ as $\hat{g}(\hat{f}(x))$ such that $\hat{g}'\hat{g}''\hat{f}'' < 0$ and all other conditions in Lemma 2 are satisfied by $\hat{g}, \hat{f}$.

For $g'g''f'' < 0$, construct $\hat{g}(y) \triangleq g(-y), \hat{f}(x) \triangleq -f(x)$. In this sense, it holds $\hat{g}(\hat{f}(\bar{x})) = g(f(\bar{x}))$. It is easy to verify that both $\hat{g}, \hat{f}$ at $y = -f(\bar{x}), x = \bar{x}$ satisfy the conditions in Lemma 2, because

$$\hat{g}'(y) = -g'(-y) = -g'(f(\bar{x})), \ \hat{g}''(y) = g''(-y) = g''(f(\bar{x})), \ \hat{g}^{(3)}(y) = -g^{(3)}(-y) = -g^{(3)}(f(\bar{x})),$$
$$\hat{f}'(\bar{x}) = -f'(\bar{x}), \ \hat{f}''(\bar{x}) = -f''(\bar{x}), \ \hat{f}^{(3)}(y) = -f^{(3)}(\bar{x}).$$

Then, it has $\hat{g}'(y)\hat{g}''(y)\hat{f}''(x) = -g'g''f'' > 0$ at $y = -f(\bar{x}), x = \bar{x}$. Therefore, we have $F'(\bar{x})F^{(3)}(\bar{x}) - 6[F''(\bar{x})]^2 < 0$ and Lemma 2 gives the result. $\qquad\square$

## F  PROOF OF THEOREM 2

**Theorem 9** (Restatement of Theorem 2). *For $f(x) = \frac{1}{4}(x^2 - \mu)^2$, consider GD with $\eta = K \cdot \frac{1}{\mu}$ where $1 < K < \sqrt{4.5} - 1 \approx 1.121$, and initialized on any point $0 < x_0 < \sqrt{\mu}$. Then it converges to an orbit of period 2, except for a measure-zero initialization where it converges to $\sqrt{\mu}$. More precisely, the period-2 orbit are the solutions $x = \delta_1 \in (0, \sqrt{\mu}), x = \delta_2 \in (\sqrt{\mu}, 2\sqrt{\mu})$ of solving $\delta$ in*

$$\eta = \frac{1}{\delta^2\left(\sqrt{\frac{\mu}{\delta^2} - \frac{3}{4}} + \frac{1}{2}\right)}. \tag{38}$$

*Proof.* Assume the 2-period orbit is $(\bar{x}_0, \bar{x}_1)$, which means

$$\bar{x}_1 = \bar{x}_0 - \eta \cdot f'(\bar{x}_0) = \bar{x}_0 + \eta \cdot (\mu - \bar{x}_0^2)\bar{x}_0,$$
$$\bar{x}_0 = \bar{x}_1 - \eta \cdot f'(\bar{x}_1) = \bar{x}_1 + \eta \cdot (\mu - \bar{x}_1^2)\bar{x}_1.$$

First, we show the existence and uniqueness of such an orbit when $K \in (1, 1.5]$ via solving a high-order equation, some roots of which can be eliminated. Then, we conduct an analysis of global convergence by defining a special interval $I$. GD starting from any point following our assumption will enter $I$ in some steps, and any point in $I$ will back to this interval after two steps of iteration. Finally, any point in $I$ will converge to the orbit $(\bar{x}_0, \bar{x}_1)$.

Before diving into the proof, we briefly show it always holds $x > 0$ under our assumption. If $x_{t-1} > 0$ and $x_t \leq 0$, the GD rule reveals $\eta(\mu - x_{t-1}^2) \leq -1$ which implies $x_{t-1}^2 \geq \mu + \frac{1}{\eta}$. However, the maximum of $x + \eta(\mu - x^2)x$ on $x \in (0, \sqrt{\mu + \frac{1}{\eta}})$ is achieved when $x^2 = \frac{1}{3}(\mu + \frac{1}{\eta})$ so the maximum value is $\sqrt{\frac{1}{3}(\mu + \frac{1}{\eta})}(\frac{2}{3} + \frac{2}{3}\eta\mu) \leq 1.4\sqrt{\frac{1}{3}(\mu + \frac{1}{\eta})} < \sqrt{\mu + \frac{1}{\eta}}$. As a result, it always holds $x > 0$.

**Part I. Existence and uniqueness of $(\bar{x}_0, \bar{x}_1)$.**

In this part, we simply denote both $\bar{x}_0, \bar{x}_1$ as $x_0$. This means $x_0$ in all formulas in this part can be interpreted as $\bar{x}_0$ and $\bar{x}_1$. Then the GD update rule tells, for the orbit in two steps,

$$x_0 \mapsto x_1 := x_0 + \eta(\mu - x_0^2)x_0,$$
$$x_1 \mapsto x_0 = x_1 + \eta(\mu - x_1^2)x_1,$$

which means

$$0 = \eta(\mu - x_0^2)x_0 + \eta\left(\mu - \left(x_0 + \eta(\mu - x_0^2)x_0\right)^2\right)\left(x_0 + \eta(\mu - x_0^2)x_0\right),$$
$$0 = \mu - x_0^2 + \left(\mu - \left(x_0 + \eta(\mu - x_0^2)x_0\right)^2\right)\left(1 + \eta(\mu - x_0^2)\right).$$

Denote $z := 1 + \eta(\mu - x_0^2)$, it is equivalent to

$$0 = \mu - x_0^2 + (\mu - z^2 x_0^2)z = (z + 1)(-x_0^2 z^2 + x_0^2 z + \mu - x_0^2)$$
$$= (z + 1)\left(-x_0^2(z - \frac{1}{2})^2 + \mu - \frac{3}{4}x_0^2\right).$$

If $z + 1 = 0$, it means $x_1 = -x_0$ which is however out of the range of our discussion on the $x > 0$ domain. So we require $-x_0^2(z - \frac{1}{2})^2 + \mu - \frac{3}{4}x_0^2 = 0$. To ensure the existence of solutions $z$, it is natural to require

$$\mu - \frac{3}{4}x_0^2 \geq 0.$$

Then, the solutions are

$$z = \frac{1}{2} \pm \sqrt{\frac{\mu}{x_0^2} - \frac{3}{4}}.$$

However, $z = \frac{1}{2} - \sqrt{\frac{\mu}{x_0^2} - \frac{3}{4}}$ can be ruled out. If it holds, $\eta(\mu - x_0^2) = z - 1 < -\frac{1}{2}$ which means $x_0^2 > \mu + \frac{1}{2\eta}$. Since we restrict $\eta\mu \in (1, 1.121]$, it tells $x_0^2 > \mu(1 + \frac{1}{1.242})$ contradicting with $\mu \geq \frac{3}{4}x_0^2$.

Hence, $z = \frac{1}{2} + \sqrt{\frac{\mu}{x_0^2} - \frac{3}{4}}$ is the only reasonable solution, which is saying

$$\eta(\mu - x_0^2) = -\frac{1}{2} + \sqrt{\frac{\mu}{x_0^2} - \frac{3}{4}}.$$

Given a certain $\eta$, the above expression is a third-order equation of $x_0^2$ to solve. Apparently $x_0^2 = \mu$ is one trivial solution, since for any learning rate, the gradient descent stays at the global minimum. Then the two other solutions are exactly the orbit $(\bar{x}_0, \bar{x}_1)$, if the equation does have three different roots. This also guarantees the uniqueness of such an orbit.

Assuming $x_0^2 \neq \mu$, the above expression can be reformulated as

$$\eta = \frac{1}{x_0^2 \left( \sqrt{\frac{\mu}{x_0^2} - \frac{3}{4}} + \frac{1}{2} \right)}. \tag{39}$$

One necessary condition for existence is $\mu \geq \frac{3}{4}x_0^2$. Note that here $x_0$ can be both $\bar{x}_0, \bar{x}_1$, one of which is larger than $\sqrt{\mu}$. For simplicity, we assume $\bar{x}_0 < \sqrt{\mu} < \bar{x}_1$. Since $\eta$ from Eq(39) is increasing with $x_0^2$ when $\mu < x_0^2$, let $x_0^2 = \frac{4}{3}\mu$ and achieve the upper bound as

$$\eta\mu \leq \frac{3}{2}, \tag{40}$$

which is satisfied by our assumption $1 < \eta\mu < \sqrt{4.5} - 1 \approx 1.121$.

Therefore, we have shown the existence and uniqueness of a period-2 orbit.

**Part II. Global convergence to $(\bar{x}_0, \bar{x}_1)$.**

The proof structure is as follows:

1. There exists a special interval $I := [x_s, \sqrt{\mu})$ such that any point in $I$ will back to this interval surely after two steps of gradient descent. And $\bar{x}_0 \in I$.

2. Initialized from any point in $I$, the gradient descent process will converge to $\bar{x}_0$ (every two steps of GD).

3. Initialized from any point between $0$ and $\sqrt{\mu}$, the gradient descent process will fall into $I$ in some steps.

**(II.1)** Consider a function $F_\eta(x) = x + \eta(\mu - x^2)x$ performing one step of gradient descent. Since $F'_\eta(x) = 1 + \eta\mu - 3\eta x^2$, we have $F'_\eta(x) > 0$ for $0 < x^2 < \frac{1}{3}\left(\mu + \frac{1}{\eta}\right)$ and $F'_\eta(x) < 0$ otherwise.

It is obvious that the threshold has $x_s^2 := \frac{1}{3}\left(\mu + \frac{1}{\eta}\right) < \mu$. In the other words, for any point on the right of $x_s$, GD returns a point in a decreasing manner.

To prove anything further, we would like to restrict $\bar{x}_0 \geq x_s$, which is

$$\bar{x}_0^2 \geq \frac{1}{3}\left(\mu + \frac{1}{\eta}\right) = \frac{1}{3}\left(\mu + \bar{x}_0^2\left(\sqrt{\frac{\mu}{\bar{x}_0^2} - \frac{3}{4}} + \frac{1}{2}\right)\right).$$

Solving this inequality tells

$$\bar{x}_0^2 \geq \frac{3 + \sqrt{2}}{7}\mu. \tag{41}$$

Consequently, by applying Eq(39), we have

$$\eta\mu \leq \sqrt{4.5} - 1 \approx 1.121. \tag{42}$$

With the above discussion of $x_s$, we are able to define the special internal $I := [x_s, \sqrt{\mu})$. From the definition of $F_\eta$, consider a function representing two steps of gradient descent $F_\eta^2(x) := F_\eta(F_\eta(x))$. From previous discussion, we know $F_\eta^2(\bar{x}_0) = \bar{x}_0$. What about $F_\eta^2(x_s)$?

It turns out $F_\eta^2(x_s) > x_s$: we have $F_\eta(x_s) = x_s(1 + \eta\mu - \eta x_s^2) = x_s \cdot \frac{2}{3}(1 + \eta\mu)$ and, furthermore, $F_\eta^2(x_s) = F_\eta(x_s \cdot \frac{2}{3}(1+\eta\mu)) = x_s \cdot \frac{2}{3}(1+\eta\mu) \cdot \left(1 + \eta\mu - \frac{4}{27}(1+\eta\mu)^3\right)$. Then we get $F_\eta^2(x_s) > x_s$ because

$$\frac{2}{3}(1+\eta\mu) \cdot \left(1 + \eta\mu - \frac{4}{27}(1+\eta\mu)^3\right) > 1 \quad \text{if} \quad \eta\mu \in (1, \sqrt{4.5} - 1). \tag{43}$$

Combining the following facts, i) $F_\eta^2(x) - x$ is continous wrt $x$, ii) $F_\eta^2(x_s) - x_s > 0$, and iii) $F_\eta^2(\bar{x}_0) - \bar{x}_0 = 0$ is the only zero point on $x \in [x_s, \bar{x}_0]$, we can conclude that

$$F_\eta^2(x) > x, \quad \forall x \in [x_s, \bar{x}_0). \tag{44}$$

Meanwhile, we can prove $F_\eta^2(x) < x$ for any $x \in (\bar{x}_0, \sqrt{\mu})$. Since $F_\eta^2(\mu) - \mu = 0$ and $F_\eta^2(\bar{x}_0) - \bar{x}_0 = 0$ are the only two zero cases, we only need to show $\exists \hat{x} \in (\bar{x}_0, \sqrt{\mu})$, such that $F_\eta^2(\hat{x}) < \hat{x}$. We compute the derivative of $F_\eta^2(x) - x$ at $x^2 = \mu$, which is $\frac{d}{dx}F_\eta^2(x) - x|_{x^2=\mu} = -1 + F'(F(x))F'(x)|_{x^2=\mu} = -1 + [F'(\sqrt{\mu})]^2 = -1 + (1 - 2\eta\mu)^2 > 0$. Then combining it with $F_\eta^2(\bar{x}_0) = \bar{x}_0$, there exists a point $\hat{x} \in (\bar{x}_0, \sqrt{\mu})$ that is very close to $\sqrt{\mu}$ such that $F_\eta^2(\hat{x}) < \hat{x}$. Hence, we can conclude that

$$F_\eta^2(x) < x, \quad \forall x \in (\bar{x}_0, \sqrt{\mu}). \tag{45}$$

Since $F_\eta(\cdot)$ is decreasing on $[x_s, \infty)$ and $F_\eta(x) > x_s$ for $x \in [x_s, \sqrt{\mu}]$, it is fair to say $F_\eta^2(x)$ is increasing on $x \in [x_s, \sqrt{\mu}]$. Hence, we have $F_\eta^2(x) \leq F_\eta^2(\bar{x}_0) = \bar{x}_0, \forall x \in [x_s, \bar{x}_0]$. And $F_\eta^2(x) \geq F_\eta^2(\bar{x}_0) = \bar{x}_0, \forall x(\bar{x}_0, \sqrt{\mu})$

Combining the above results, we have

$$F_\eta^2(x) \in (x, \bar{x}_0], \quad \forall x \in [x_s, \bar{x}_0), \tag{46}$$

$$F_\eta^2(x) \in [\bar{x}_0, x), \quad \forall x \in (\bar{x}_0, \sqrt{\mu}). \tag{47}$$

**(II.2)** A consequence of Exp(46, 47) is that any point in $I$ will converge to $\bar{x}_0$ with even steps of gradient descent. For simplicity, we provide the proof for $x \in [x_s, \bar{x}_0)$.

Denote $a_0 \in [x_s, \bar{x}_0)$ and $a_n := F_\eta^2(a_{n-1}), n \geq 1$. The series $\{a_i\}_{i \geq 0}$ satisfies

$$\bar{x}_0 \geq a_{n+1} > a_n > a_0. \tag{48}$$

Since the series is bounded and strictly increasing, it is converging. Assume it is converging to $a$. If $a < \bar{x}_0$, then

$$\bar{x}_0 \geq F_\eta^2(a) > a > F_\eta^2(a_n).$$

Since $F_\eta^2(\cdot)$ is continuous, so $\exists\, \delta > 0$, such that, when $|x - a| < \delta$, we have

$$|F_\eta^2(x) - F_\eta^2(a)| < F_\eta^2(a) - a. \tag{49}$$

Since $a$ is the limit, so $\exists\, N > 0$, such that, when $n > N$, $0 < a - F_\eta^2(a_n) < \delta$. So, combining with Exp(49), we have

$$|F_\eta^2(F_\eta^2(a_n)) - F_\eta^2(a)| < F_\eta^2(a) - a.$$

But LHS $= F_\eta^2(a) - a_{n+2} > F_\eta^2(a) - a$, so we reach a contradiction.

Hence, we have $\{a_i\}$ converges to $\bar{x}_0$.

**(II.3)** Obviously, any initialization in $(0, \sqrt{\mu})$ will have gradient descent run into (i) the interval $I$, or (ii) the interval on the right of $\sqrt{\mu}$, *i.e.*, $(\sqrt{\mu}, \infty)$. The first case is exactly our target.

Now consider the second case. From the definition of $x_s$ in part III.1, we know $F_\eta(x_s) = \max_{x \in [0, \sqrt{\mu}]} F_\eta(x)$. So it is fair to say this case is $x_n \in (\sqrt{\mu}, F_\eta(x_s)]$. Then the next step will go into the interval $I$, because

$$F_\eta(x_n) \geq F_\eta(F_\eta(x_s)) = F_\eta^2(x_s) > x_s,$$

where the first inequality is from the decreasing property of $F_\eta(\cdot)$ and the second inequality is due to $F_\eta^2(x) > x$ on $x \in [x_s, \bar{x}_0)$. $\qquad\square$

## G    PROOF OF THEOREM 5

**Theorem 10** (Restatement of Theorem 5). *For $f(x, y) = \frac{1}{2}(xy - \mu)^2$, consider GD with learning rate $\eta = K \cdot \frac{1}{\mu}$. Assume both $x$ and $y$ are always positive during the whole process $\{x_i, y_i\}_{i \geq 0}$. In this process, denote a series of all points with $xy > \mu$ as $\mathcal{P} = \{(x_i, y_i) | x_i y_i > \mu\}$. Then $|x - y|$ decays to 0 in $\mathcal{P}$, for any $1 < K < 1.5$.*

*Proof.* Consider the current step is at $(x_t, y_t)$ with $x_t y_t > \mu$. After two steps of gradient descent, we have

$$x_{t+1} = x_t + \eta(\mu - x_t y_t)y_t \tag{50}$$
$$y_{t+1} = y_t + \eta(\mu - x_t y_t)x_t \tag{51}$$
$$x_{t+2} = x_{t+1} + \eta(\mu - x_{t+1}y_{t+1})y_{t+1} \tag{52}$$
$$y_{t+2} = y_{t+1} + \eta(\mu - x_{t+1}y_{t+1})x_{t+1}, \tag{53}$$

with which we have the difference evolve as

$$y_{t+1} - x_{t+1} = (y_t - x_t)\left(1 - \eta\left(\mu - x_t y_t\right)\right) \tag{54}$$
$$y_{t+2} - x_{t+2} = (y_{t+1} - x_{t+1})\left(1 - \eta\left(\mu - x_{t+1}y_{t+1}\right)\right). \tag{55}$$

Meanwhile, we have

$$\begin{aligned}
x_{t+1}y_{t+1} &= x_t y_t + \eta\left(\mu - x_t y_t\right)\left(x_t^2 + y_t^2\right) + \eta^2\left(\mu - x_t y_t\right)^2 x_t y_t \\
&= x_t y_t \left(1 + \eta\left(\mu - x_t y_t\right)\right)^2 + \eta\left(\mu - x_t y_t\right)\left(x_t - y_t\right)^2
\end{aligned} \tag{56}$$

Note that the second term in Eq(56) vanishes when $x$ and $y$ are balanced. When they are not balanced, if $x_t y_t > \mu$, it holds $x_{t+1}y_{t+1} < x_t y_t \left(1 + \eta\left(\mu - x_t y_t\right)\right)^2$. Incorporating this inequality into Eq(54, 55) and assuming $y_t - x_t > 0$, it holds

$$y_{t+2} - x_{t+2} < (y_t - x_t)\left(1 - \eta\left(\mu - x_t y_t\right)\right)\left(1 - \eta\left(\mu - x_t y_t\left(1 + \eta\left(\mu - x_t y_t\right)\right)^2\right)\right). \tag{57}$$

To show that $|x - y|$ is decaying as in the theorem, we are to show

1. $y_{t+2} - x_{t+2} < y_t - x_t$

2. $y_{t+2} - x_{t+2} > -(y_t - x_t)$

Note that, although $x_t y_t > \mu$, it is not sure to have $x_{t+2} y_{t+2} > \mu$. However, for any $0 < x_i y_i < \mu$ and $K < 2$, we have

$$\frac{|x_{i+1} - y_{i+1}|}{|x_i - y_i|} = |1 - \eta(\mu - x_i y_i)| < 1, \tag{58}$$

which is saying $|x - y|$ decays until it reaches $xy > \mu$. So it is enough to prove the above two inequalities, whether or not $x_{t+2} y_{t+2} > \mu$.

**Part I. To show** $y_{t+2} - x_{t+2} < y_t - x_t$

Since we wish to have $y_{t+2} - x_{t+2} < y_t - x_t$, it is sufficient to require

$$(1 - \eta(\mu - x_t y_t))\left(1 - \eta\left(\mu - x_t y_t(1 + \eta(\mu - x_t y_t))^2\right)\right) < 1. \tag{59}$$

Since we assume $x_{t+1}, y_{t+1} > 0$, Eq (50, 51) tells $\eta(\mu - x_t y_t) > -\min\{\frac{x_t}{y_t}, \frac{y_t}{x_t}\}$, which is equivalent to $1 - \eta(\mu - x_t y_t) < 1 + \min\{\frac{x_t}{y_t}, \frac{y_t}{x_t}\}$.

**(I.1) If** $\eta(\mu - x_{t+1} y_{t+1}) \geq \frac{1}{2}$

Then we have $1 - \eta(\mu - x_{t+1} y_{t+1}) \leq \frac{1}{2}$. As a result,

$$\frac{y_{t+2} - x_{t+2}}{y_t - x_t} = (1 - \eta(\mu - x_t y_t))(1 - \eta(\mu - x_{t+1} y_{t+1})) < \left(1 + \min\{\frac{x_t}{y_t}, \frac{y_t}{x_t}\}\right) \times \frac{1}{2} \tag{60}$$

$$= \frac{1}{2} + \frac{1}{2}\min\{\frac{x_t}{y_t}, \frac{y_t}{x_t}\} \tag{61}$$

**(I.2) If** $\eta(\mu - x_{t+1} y_{t+1}) < \frac{1}{2}$ **and** $x_{t+1} y_{t+1} \leq x_s^2 = \frac{1}{3}\left(\mu + \frac{1}{\eta}\right)$

The second condition reveals

$$\frac{y_{t+2} - x_{t+2}}{y_{t+1} - x_{t+1}} = 1 - \eta(\mu - x_{t+1} y_{t+1}) \leq 1 - \eta\left(\mu - \frac{1}{3}\left(\mu + \frac{1}{\eta}\right)\right)$$

$$= \frac{4}{3} - \frac{2}{3}K. \tag{62}$$

The first condition is equivalent to $x_{t+1} y_{t+1} > \mu - \frac{1}{2\eta}$. Since the second term in Eq(56) is negative, we have

$$x_t y_t(1 + \eta(\mu - x_t y_t))^2 > \mu - \frac{1}{2\eta}, \tag{63}$$

with which we would like to find an upper bound of $x_t y_t$.

Denoting $b = x_t y_t$, consider a function $q(b) = b(1 + \eta(\mu - b))^2$. Obviously $q(\mu) = \mu$. Its derivative is $q'(b) = (1 + \eta\mu - \eta b)(1 + \eta\mu - 3\eta b) < 0$ on the domain of our interest. If we can show an (negative) upper bound for the derivative as $q'(b) < -1$ on a proper domain, then it is fair to say that, from Exp(63), $x_t y_t < \mu + \frac{1}{2\eta}$. Then we have

$$\frac{y_{t+1} - x_{t+1}}{y_t - x_t} = 1 - \eta(\mu - x_t y_t) < 1 - \eta\left(\mu - \left(\mu - \frac{1}{2\eta}\right)\right) = \frac{3}{2}. \tag{64}$$

Then, combining Exp(64, 62), it tells

$$\frac{y_{t+2} - x_{t+2}}{y_t - x_t} < 2 - K. \tag{65}$$

The remaining is to show $q'(b) < -1$ on a proper domain. We have $q'(b) = (1 + \eta\mu - 2\eta b)^2 - (\eta b)^2$, which is equal to $1 - 2\eta\mu < -1$ when $b = \mu$. Meanwhile, the derivative of $q'(b)$ is $q''(b) = -2\eta(\eta b + (1 + \eta\mu - 2\eta b)) = -2\eta(1 + \eta\mu - \eta b)$, which is negative when $b < \mu + \frac{1}{\eta}$. As a result, it always holds $q'(b) < -1$ when $b < \mu + \frac{1}{\eta}$.

**(I.3) If** $x_{t+1}y_{t+1} \geq x_s^2$

Denoting again $b = x_t y_t$, the above inequality in is saying, with $b > \mu$,

$$p(b) = (1 - \eta(\mu - b))\left(1 - \eta\left(\mu - b(1 + \eta(\mu - b))^2\right)\right) < 1. \tag{66}$$

After expanding $p(\cdot)$, we have

$$p(b) - 1 = \eta(\mu - b)\left(-2 + \eta(\mu - b) + 2\eta b - \eta^2 b(\mu - b) - \eta^3 b(\mu - b)^2\right).$$

Apparently $p(\mu) = 1$. So it is necessary to investigate whether $p'(b) < 0$ on $b > \mu$, as

$$p'(b) = 2 - 2\eta b + (\mu - b)\left(\eta^2(1 + \eta(\mu - b))(-\mu + 3b) + \eta^3 b(\mu - b)\right).$$

Since $\eta b > 1$ and $b > \mu$, it is enough to require

$$(1 + \eta(\mu - b))(-\mu + 3b) + \eta b(\mu - b) > 0$$
$$(1 + \eta(\mu - b))(-\mu + b) + \eta b(\mu - b) + 2b(1 + \eta(\mu - b)) > 0.$$

It suffices to show

$$\eta(\mu - b) + 2(1 + \eta(\mu - b)) = 2 + 3\eta(\mu - b) > 0. \tag{67}$$

Since $x_{t+1}y_{t+1} \geq x_s^2 = \frac{1}{3}\left(\mu + \frac{1}{\eta}\right)$, it holds

$$b(1 + \eta(\mu - b))^2 \geq \frac{1}{3}\left(\mu + \frac{1}{\eta}\right)$$

$$2 + 3\eta(\mu - b) \geq \sqrt{\frac{3\left(\mu + \frac{1}{\eta}\right)}{b}} - 1 > 0,$$

where the last inequality holds because: if $b \geq 3\left(\mu + \frac{1}{\eta}\right)$, then $1 + \eta(\mu - b) \leq -2\eta\mu - 2 < 0$, which contradicts with the assumption that both $x_{t+1}, y_{t+1}$ are positive. As a result, the above argument gives

$$\frac{y_{t+2} - x_{t+2}}{y_t - x_t} < p(b) < 1 - 2(K-1)(b - \mu). \tag{68}$$

**Part II. To show** $y_{t+2} - x_{t+2} > -(y_t - x_t)$

Since $x_t y_t > \mu$, we have $1 - \eta(\mu - x_t y_t) > 1$. Combining with $1 - \eta(\mu - x_t y_t) < 2$, it holds

$$\frac{y_{t+1} - x_{t+1}}{y_t - x_t} = 1 - \eta(\mu - x_t y_t) \in (1, 2).$$

So the remaining is to have $\frac{y_{t+2} - x_{t+2}}{y_{t+1} - x_{t+1}} > -0.5$. Actually it is $1 - \eta(\mu - x_{t+1}y_{t+1}) \geq 1 - \eta\mu = 1 - K$. Therefore, we have

$$\frac{y_{t+2} - x_{t+2}}{y_t - x_t} > -1 + (3 - 2K), \tag{69}$$

as required.

**Part III. To show** $y_t - x_t$ **converges to 0**

From Exp (61, 65, 68, 69), we have for points in $\mathcal{P}$, $|y - x|$ is a monotone strictly decreasing sequence lower bounded by 0. Hence it is convergent. Actually it converges to 0. If not, assuming it converges to $\epsilon > 0$, the next point will have the difference as $\tilde{\epsilon} < \epsilon$ as well as all following points. Hence, the contradiction gives the convergence to 0. □

# H    PROOF OF LEMMA 3

**Lemma 6** (Restatement of Lemma 3). *In the setting of Theorem 5, denote the initialization as* $m = \frac{|y_0 - x_0|}{\sqrt{\mu}}$ *and* $x_0 y_0 > \mu$. *Then, during the whole process, both* $x$ *and* $y$ *will always stay positive, denoting* $p = \frac{4}{\left(m + \sqrt{m^2 + 4}\right)^2}$ *and* $q = (1 + p)^2$, *if*

$$\max\left\{\eta(x_0 y_0 - \mu), \frac{4}{27}(1 + K)^3 + \left(\frac{2}{3}K^2 - \frac{1}{3}K + \frac{qK^2}{2(K+1)}m^2\right)qm^2 - K\right\} < p.$$

*Proof.* Considering $x_t y_t > \mu$, one step of gradient descent returns

$$x_{t+1} = x_t + \eta(\mu - x_t y_t)y_t$$
$$y_{t+1} = y_t + \eta(\mu - x_t y_t)x_t.$$

To have both $x_{t+1} > 0, y_{t+1} > 0$, it suffices to have

$$\eta(x_t y_t - \mu) < \min\left\{\frac{y_t}{x_t}, \frac{x_t}{y_t}\right\}. \tag{70}$$

This inequality will be the main target we need to resolve in this proof.

First, we are to show

$$\min\left\{\frac{y_0}{x_0}, \frac{x_0}{y_0}\right\} > \frac{4}{\left(m + \sqrt{m^2 + 4}\right)^2}.$$

With the difference fixed as $m = (y_0 - x_0)/\sqrt{\mu}$, assuming $y_0 > x_0$, we have $m/y_0 = (1 - x_0/y_0)/\sqrt{\mu}$. if $x_0 y_0$ increases, both $x_0$ and $y_0$ increase then $m/y_0$ decreases, which means $x_0/y_0$ increases. As a result, we have

$$\min\left\{\frac{y_0}{x_0}, \frac{x_0}{y_0}\right\} > \min\left\{\frac{y_0}{x_0}, \frac{x_0}{y_0}\right\}\bigg|_{x_0 y_0 = \mu} = \frac{4}{\left(m + \sqrt{m^2 + 4}\right)^2}.$$

Therefore, at initialization, to have positive $x_1$ and $y_1$, it is enough to require

$$\eta(x_0 y_0 - \mu) < \frac{4}{\left(m + \sqrt{m^2 + 4}\right)^2} \triangleq r.$$

From Theorem 5, it is guaranteed that $|x_t - y_t| < |x_0 - y_0|$ with $t \geq 2$ until it reaches $x_t y_t > \mu$, with which $r$ is still a good lower bound for $\min\{y_t/x_t, x_t/y_t\}$. So what remains to show is it satisfies $\eta(x_t y_t - \mu) < r$ for the next first time $x_t y_t > \mu$. If this holds, we can always iteratively show, for any $x_t y_t > \mu$ along gradient descent,

$$\eta(x_t y_t - \mu) < r < \min\left\{\frac{y_t}{x_t}, \frac{x_t}{y_t}\right\}.$$

Note that $r$ itself is independent of $x_t y_t$ and all the history, so it is ideal to compute a uniform upper bound of $\eta(x_t y_t - \mu)$ with any pair of $(x_{t-1}, y_{t-1})$ satisfying $x_{t-1} y_{t-1} < \mu$. Actually it is possible, since we have $|x_{t-1} - y_{t-1}|$ bounded as in Theorem 5.

Assume $x_i y_i > \mu$ and it satisfies the condition of $\eta(x_i y_i - \mu) < r$ and $|x_i - y_i| < |x_0 - y_0|$. As in (54), we have

$$\frac{x_{i+1} - y_{i+1}}{x_i - y_i} = 1 - \eta(\mu - x_i y_i) \in (1, 1 + r). \tag{71}$$

Hence, it suffices to get the maximum value of $g(z)$, with $z \in (0, \mu)$, as

$$g(z) = z(1 + \eta(\mu - z))^2 + \eta(\mu - z)(1 + r)^2(x_0 - y_0)^2, \tag{72}$$

which is from (56). Denote $\bar{z} = \arg\max g(z)$. Obviously $\bar{z} < \frac{1}{3}(\mu + \frac{1}{\eta}) \triangleq z_b$, because the first term of $g(z)$ achieves maximum at $z = \frac{1}{3}(\mu + \frac{1}{\eta})$ and the second term is in a decreasing manner with $z$. Then let's take the derivative of $g(z)$ as

$$g'(z) = (1 + \eta(\mu - z))(1 + \eta\mu - 3\eta z) - \eta(1 + r)^2(x_0 - y_0)^2$$

$$= (1 + \eta(\mu - z))\left(1 + \eta\mu - 3\eta z - \frac{\eta(1 + r)^2(x_0 - y_0)^2}{1 + \eta(\mu - z)}\right),$$

where the first term is always positive, so we have

$$1 + \eta\mu - 3\eta\bar{z} - \frac{\eta(1 + r)^2(x_0 - y_0)^2}{1 + \eta(\mu - \bar{z})} = 0, \tag{73}$$

which means

$$\bar{z} = \frac{1}{3\eta}\left(1 + \eta\mu - \frac{\eta(1 + r)^2(x_0 - y_0)^2}{1 + \eta(\mu - \bar{z})}\right) \tag{74}$$

$$> \frac{1}{3\eta}\left(1 + \eta\mu - \frac{\eta(1 + r)^2(x_0 - y_0)^2}{1 + \eta(\mu - \frac{1}{3}(\mu + \frac{1}{\eta}))}\right) \tag{75}$$

$$= \frac{1}{3}\left(\mu + \frac{1}{\eta} - \frac{3(1 + r)^2}{2(\eta + 1)}(x_0 - y_0)^2\right) \tag{76}$$

$$\triangleq z_s, \tag{77}$$

where the inequality is from $\bar{z} < \frac{1}{3}(\mu + \frac{1}{\eta})$. As a result, it is safe to say

$$g(z) \le z\left(1 + \eta(\mu - z)\right)^2\bigg|_{z=z_b} + \eta(\mu - z)(1 + r)^2(x_0 - y_0)^2\bigg|_{z=z_s} \tag{78}$$

$$= \frac{4}{27}(1 + \eta\mu)^3 \cdot \frac{1}{\eta} + \eta(1 + r)^2\left(\frac{2}{3}\mu - \frac{1}{3\eta} + \frac{2}{\eta\mu + 1}(x_0 - y_0)^2\right)(x_0 - y_0)^2, \tag{79}$$

with which we are able to compute $\max \eta(g(z) - \mu)$, which is exactly the final result. $\square$

## I   PROOF OF THEOREM 3

**Theorem 11** (Restatement of Theorem 3)**.** *In the above setting, consider a teacher neuron $\tilde{w} = [1, 0]$ and set the learning rate $\eta = Kd$ with $K \in (1, 1.1]$. Initialize the student as $\|w^{(0)}\| = v^{(0)} \triangleq \epsilon \in (0, 0.10]$ and $\langle w^{(0)}, \tilde{w}\rangle \ge 0$. Then, for $t \ge T_1 + 4$, $w_y^{(t)}$ decays as*

$$w_y^{(t)} < 0.1 \cdot (1 - 0.030K)^{t - T_1 - 4}, \quad T_1 \le \left\lceil \log_{2.56} \frac{1.35}{\pi\beta^2} \right\rceil, \quad \beta = \left(1 + \frac{1.1}{\pi}\right)\epsilon.$$

*Proof.* We restate the update rules as

$$\Delta v^{(t)} := v^{(t+1)} - v^{(t)} = Kw_x^{(t)} \left[ (-v^{(t)} w_x^{(t)} + 1) - v^{(t)} w_y^{(t)} \frac{w_y^{(t)}}{w_x^{(t)}} - \frac{1}{\pi} \left( \arctan\left( \frac{w_y^{(t)}}{w_x^{(t)}} \right) - \frac{w_y^{(t)}}{w_x^{(t)}} \right) \right],$$

$$= Kw_x^{(t)} \left[ (-v^{(t)} w_x^{(t)} + 1) - \frac{1}{\pi} \left( \arctan\left( \frac{w_y^{(t)}}{w_x^{(t)}} \right) - \frac{w_x^{(t)} w_y^{(t)}}{\left\| w^{(t)} \right\|^2} \right) \right]$$

$$+ K \frac{(w_y^{(t)})^2}{v^{(t)}} \left( -(v^{(t)})^2 + \frac{v^{(t)} w_y^{(t)}}{\pi \left\| w^{(t)} \right\|^2} \right) \tag{80}$$

$$\Delta w_x^{(t)} := w_x^{(t+1)} - w_x^{(t)} = Kv^{(t)} \left[ (-v^{(t)} w_x^{(t)} + 1) - \frac{1}{\pi} \left( \arctan\left( \frac{w_y^{(t)}}{w_x^{(t)}} \right) - \frac{w_x^{(t)} w_y^{(t)}}{\left\| w^{(t)} \right\|^2} \right) \right],$$
$$\tag{81}$$

$$\Delta w_y^{(t)} = w_y^{(t)} \cdot K \left( -(v^{(t)})^2 + \frac{v^{(t)} w_y^{(t)}}{\pi \left\| w^{(t)} \right\|^2} \right), \tag{82}$$

$$w_y^{(t+1)} = \left| w_y^{(t)} + \Delta w_y^{(t)} \right|. \tag{83}$$

For simplicity, we will omit all superscripts of time $t$ unless clarification is necessary. From (83), if the target is to show $w_y$ decaying with a linear rate, it suffices to bound the factor term in (82) (by a considerable margin) as

$$-2 < K \left( -v^2 + \frac{vw_y}{\pi \left\| w \right\|^2} \right) < 0. \tag{84}$$

The technical part is the second inequality of (84). If $v, w_x > 0$, it is equivalent to

$$vw_x > \frac{w_x w_y}{\pi \left\| w \right\|^2} = \frac{w_x w_y}{\pi (w_x^2 + w_y^2)},$$

where the RHS is smaller than or equal to $\frac{1}{2\pi}$. Hence, $\frac{1}{2\pi}$ is a special threshold with which we will frequently compare $vw_x$. Another important variable to control is $v - w_x$ that reveals how the two layers are balanced. If it is too large, for the iteration $v^{(t+1)} w_x^{(t+1)}$ may explode as shown in the 2-D case.

The main idea of our proof is that

- Stage 1 with $vw_x \leq \frac{w_x w_y}{\pi \left\| w \right\|^2}$: in this stage, $w_y$ grows but it grows in a smaller rate than that of $v$ and $w_x$. Therefore, since we have an upper bound for $vw_x$ to stay in this stage, we are able to compute the upper bound of #iterations to finish this stage, which is $T_1$ in the theorem. At the end of this stage, both of $v - w_x$ and $w_y$ are bounded under our assumption of initialization.

- Stage 2 with $vw_x > \frac{w_x w_y}{\pi \left\| w \right\|^2}$: in this stage, $w_y$ decreases. Since our range of a large learning rate is relatively narrow ($1 < K \leq 1.1$), we are able to compute bounds of $vw_x, v - w_x$ and $w_y$. After eight iterations, it falls into (and stays in) a bounded basin of these three terms, in which $w_y$ decays at least in a linear rate.

## Stage 1.

We are to show that, in the last iteration of this stage, there are three facts: 1) $vw_x \leq \frac{1}{2\pi}$, 2) $v - w_x \in [-0.017, 0.17]$, and 3) $w_y \leq 0.44$.

At initialization, we assume $v^{(0)} = \left\| w^{(0)} \right\|$. Denote $\alpha_0 = \arctan(w_y^{(0)}/w_x^{(0)}) \in [0, \pi/2]$. So for next iteration we have

$$w_y^{(1)} = v^{(0)} \left( 1 + K \left( -(v^{(0)})^2 + \frac{1}{\pi} \sin \alpha_0 \right) \right), \tag{85}$$

$$w_x^{(1)} = v^{(0)} \left[ \cos \alpha_0 + K \left( 1 - (v^{(0)})^2 \cos \alpha_0 + \frac{\cos \alpha_0 \sin \alpha_0 - \alpha_0}{\pi} \right) \right]. \tag{86}$$

Apparently $w_y^{(1)}$ increases with $\alpha_0$ increasing. And

$$
\partial_{\alpha_0} w_x^{(1)} = v^{(0)} \left[ -\sin \alpha_0 + K \left( (v^{(0)})^2 \sin \alpha_0 + \frac{-\sin^2 \alpha_0 + \cos^2 \alpha_0 - 1}{\pi} \right) \right]
$$

$$
= v^{(0)} \left[ -\sin \alpha_0 + K \left( \left( (v^{(0)})^2 - \frac{\sin \alpha_0}{\pi} \right) \sin \alpha_0 + \frac{-\sin^2 \alpha_0}{\pi} \right) \right].
$$

Since in stage 1 it holds $\Delta w_y > 0$ which means $-(v^{(0)})^2 + \frac{1}{\pi} \sin \alpha_0 > 0$ in (85). So it follows $\partial_{\alpha_0} w_x^{(1)} \leq 0$. Combining the above arguments, we have

$$
w_x^{(1)} \geq w_x^{(1)}|_{\alpha_0=\frac{\pi}{2}} = \frac{K}{2} v^{(0)},
$$

$$
w_y^{(1)} \leq w_y^{(1)}|_{\alpha_0=\frac{\pi}{2}} = \left( 1 + \frac{K}{\pi} - K(v^{(0)})^2 \right) v^{(0)} \leq \left( 1 + \frac{K}{\pi} \right) v^{(0)},
$$

$$
\frac{w_y^{(1)}}{w_x^{(1)}} \leq \frac{2 + \frac{2K}{\pi}}{K} \leq 2.7.
$$

Regarding $\frac{v}{w_y}$, it has $v^{(0)} \geq w_y^{(0)}$ at initialization due to $v^{(0)} = \left\| w^{(0)} \right\|$. From (80, 81, 82), we have $v\Delta v = w_x \Delta w_x + w_y \Delta w_y$. So it holds $v\Delta v \geq y\Delta y$. Meanwhile, $\frac{\Delta w_y}{v} = K(-vw_y + \frac{w_y^2}{\pi \|w\|^2}) \in [0, \frac{K}{\pi}]$. From Lemma 7, given $v^{(t)} \geq w_y^{(t)}$ and $\frac{\Delta w_y}{v} \in [0, 1]$ for any $t$ in this stage, it always holds $v^{(t+1)} \geq w_y^{(t+1)}$.

Therefore, it is fair to say

$$
\frac{v^{(1)} w_x^{(1)}}{(w_y^{(1)})^2} \geq \frac{1}{2.7}.
$$

Additionally, to bound the term $vw_y / \|w\|^2$ in $\Delta w_y$, we would like to show it always has $vw_y \leq \|w\|^2$. At initialization, it naturally holds. Then, for the every next iteration, given it holds in the last iteration, we have

$$
(v + \Delta v)(w_y + \Delta w_y) - [(w_x + \Delta w_x)^2 + (w_y + \Delta w_y)^2]
$$

$$
= (v + \frac{w_x \Delta w_x + w_y \Delta w_y}{v})(w_y + \Delta w_y) - [(w_x + \Delta w_x)^2 + (w_y + \Delta w_y)^2]
$$

$$
= vw_y + v\Delta w_y + w_x \Delta w_x (\frac{w_y}{v} + \frac{\Delta w_y}{v}) + (w_y \Delta w_y + (\Delta w_y)^2) \frac{w_y}{v} - [(w_x + \Delta w_x)^2 + (w_y + \Delta w_y)^2]
$$

$$
\leq vw_y + v\Delta w_y + w_y \Delta w_y \frac{w_y}{v} - (w_x^2 + w_y^2 + 2w_y \Delta w_y + (\Delta w_x)^2)
$$

$$
\leq v\Delta w_y + w_y \Delta w_y \frac{w_y}{v} - 2w_y \Delta w_y - (\Delta w_x)^2
$$

$$
= v\Delta w_y (1 - \frac{w_y}{v})^2 - (\Delta w_x)^2
$$

$$
\leq v\Delta w_y - (\Delta w_x)^2
$$

where the first equality uses $v\Delta v = w_x \Delta w_x + w_y \Delta w_y$, the first inequality uses the proven $v \geq w_y$ and $v \geq \Delta w_y$, the second inequality uses the assumption $vw_y \leq \|w\|^2$. Now we are to show $v\Delta w_y - (\Delta w_x)^2 \leq 0$. We have

$$
v\Delta w_y - (\Delta w_x)^2 \leq Kv^2 \frac{w_y^2}{\pi \|w\|^2} - K^2 v^2 \left( 1 - \frac{1}{2\pi} - \gamma^{(t)} \right)^2,
$$

$$
\gamma^{(t)} = \frac{1}{\pi} \left( \arctan \left( \frac{w_y^{(t)}}{w_x^{(t)}} \right) - \frac{w_x^{(t)} w_y^{(t)}}{\left\| w^{(t)} \right\|^2} \right).
$$

Since we have proven $w_y^{(1)}/w_x^{(1)} \le 2.7$, it is easy to check that

$$\frac{1}{\pi\left(1 + (\frac{w_x^{(1)}}{w_y^{(1)}})^2\right)} \le (1 - \frac{1}{2\pi} - \gamma^{(1)})^2.$$

As a result, $v\Delta w_y - (\Delta w_x)^2 \le 0$ at time 1. Furthermore, by checking each term, $v\Delta w_y - (\Delta w_x)^2$ decreases with $w_y/w_x$ decreasing. We will soon show that $w_y/w_x$ itself decreases, by showing the growth ratio of $w_x$ is larger than that of $w_y$.

Our target lower bound of the growth ratio of $w_x$ is that

$$\frac{\Delta w_x}{w_x} \ge 1 - \frac{1}{\pi} - \gamma, \tag{87}$$

which is larger than the growth ratio of $w_y$ bounded by $\frac{1}{\pi}$ due to $v\Delta w_y < \|w\|^2$. So it suffices to show $Kv/w_x \ge 1$. Assuming $Kv/w_x \ge 1$ for the current step, we need to show $Kv^{(t+1)}/w_x^{(t+1)} \ge 1$ also holds for the next step. Let's denote

$$A^{(t)} = K\left[(-v^{(t)}w_x^{(t)} + 1) - \frac{1}{\pi}\left(\arctan\left(\frac{w_y^{(t)}}{w_x^{(t)}}\right) - \frac{w_x^{(t)}w_y^{(t)}}{\|w^{(t)}\|^2}\right)\right]. \tag{88}$$

Then

$$(v + \Delta v) - \frac{1}{K}(w_x + \Delta w_x) \ge v + Aw_x - \frac{w_x}{K} - \frac{Av}{K}$$

$$= (v - \frac{w_x}{K})(1 - KA) + v(K - \frac{1}{K})A. \tag{89}$$

If $KA \le 1$, since $K > 1$ and $A > 0$, we have (89) as positive, which is what we need. If $KA > 1$, then

$$(89) \ge (v - \frac{w_x}{K})(1 - K^2) + v(K - \frac{1}{K})A$$

$$= ((-K + A)v + w_x)(K - \frac{1}{K}),$$

where the first inequality is due to $A \le K$ and the assumption of $Kv^{(t)}/w_x^{(t)} \ge 1$. Then it suffices to show $(-K + A)v + w_x \ge (-K + \frac{1}{K})v + w_x \ge 0$. Note that $-K + 1/K \in (-0.2, 0]$ when $K \in (1, 1.1]$. It is easy to verify that $v^{(1)} \le 5w_x^{(1)}$. Then, for the next step, we need to show $v^{(t+1)} \le 5w_x^{(t+1)}$ given $v^{(t)} \le 5w^{(t+1)}$. To prove this, we are to bound $v - w_x$, as

$$v^{(t+1)} - w_x^{(t+1)} = (1 - A)(v - w) + K\frac{w_y^2}{v}(-v^2 + \frac{vw_y}{\pi\|w\|^2})$$

$$\le 0.4(v - w) + Kw_y\frac{w_y^2}{\pi\|w\|^2} \le 0.4(v - w) + \frac{Kw_y}{\pi}, \tag{90}$$

where the first inequality is due to, when $w_y/w_x \le 2.7$,

$$A = K\left[-v^{(t)}w_x^{(t)} + \frac{1}{\pi}\frac{w_x^{(t)}w_y^{(t)}}{\|w^{(t)}\|^2}\right] + K\left[1 - \frac{1}{\pi}\arctan\left(\frac{w_y^{(t)}}{w_x^{(t)}}\right)\right]$$

$$\ge K\left[1 - \frac{1}{\pi}\arctan\left(\frac{w_y^{(t)}}{w_x^{(t)}}\right)\right] \ge 0.6.$$

We will later show that $v^{(t+1)} - w^{(t+1)} \ge -0.1(v^{(t)} - w^{(t)})$. Combining this with (90), it is safe to say

$$v^{(t+1)} - w^{(t+1)} \le 0.4(v - w) + \frac{Kw_y}{\pi} \le 0.4 \times 4w + \frac{K \times 5w}{\pi} \le 4w,$$

where the second inequality is due to $v \leq 5w$ and $v \geq w_y$. Since $w^{(t+1)} \geq w^{(t)}$ (due to $A > 0$) in this stage, we have $v^{(t+1)} \leq 5w_x^{(t+1)}$.

Combining the above discussion, we have prove (87). Obviously, when $w_y/w_x \leq 2.7$, RHS of (87) is at least 0.55, larger than $1.1/\pi$, which is the upper bound of the $\Delta w_y/w_y$. As a result, $w_y/w_x$ keeps decreasing.

The next step is to show the growing ratio of $vw_x$ is much larger than that of $w_y$. From (81, 82), it holds

$$v^{(t+1)}w_x^{(t+1)} = (v + \Delta v)(w_x + \Delta w_x) \geq vw_x + KA(v^2 + w_x^2) + K^2A^2vw_x$$
$$\geq vw_x(1 + A)^2,$$

where the first inequality is due to $\Delta w_y \geq 0$. It follows $v^{(t+1)}w_x^{(t+1)}/v^{(t)}w_x^{(t)} \geq 1.6^2 = 2.56$.

So far, we have shown the following facts: under the defined initialization at time 0, starting from time 1, we have

1. $vw_x \leq 1/2\pi$.

2. $\Delta w_x/w_x + 1 \geq 1.55$.

3. $\Delta w_y/w_y + 1 \leq 1 + K/\pi$.

4. $w_y/w_x \leq 2.7$ and keeps decreasing.

5. $v^{(t+1)}w_x^{(t+1)}/v^{(t)}w_x^{(t)} \geq 2.56$.

6. $v \geq w_y$.

7. $v\Delta w_y < (\Delta w_x)^2$.

Now we are to use the above facts to bound $vw_x, w_y$ and $v - w_x$ to the end of stage 1.

For $vw_x$, in previous discussion, we have shown that $vw_x \leq \frac{1}{2\pi}$. Actually, there is another special value

$$\frac{w_x w_y}{\pi(w_x^2 + w_y^2)} = 0.104 \text{ when } w_y/w_x = 2.7. \tag{91}$$

This value is slightly larger than $1/4\pi$. Hence, we would like to split the analysis into three parts: in the **first step of stage 2**,

1. $vw_x \geq \frac{1}{2\pi}$.

2. $\frac{1}{4\pi} \leq vw_x < \frac{1}{2\pi}$.

3. $vw_x < \frac{1}{4\pi}$.

Note that, although we are discussing the stage 1 in this section, investigating the lower bound of the first step in stage 2 helps calculate the number of iterations in stage 1. Furthermore, it helps bound several variables in stage 1.

**Case (I). If $vw_x \geq \frac{1}{2\pi}$ in first step of stage 2:**

Since we have prove $\frac{v^{(1)}w_x^{(1)}}{(w_y^{(1)})^2} \geq 1/2.7$ and $v^{(t+1)}w_x^{(t+1)}/v^{(t)}w_x^{(t)} \geq 2.56$, the number of iterations for $vw_x$ to reach $1/2\pi$ is at most

$$T_1 \leq \left\lceil \log_{2.56} \frac{\frac{1}{2\pi}}{(w_y^{(1)})^2/2.7} \right\rceil. \tag{92}$$

Meanwhile, starting from time 1, the growth ratio of $w_y$ is

$$(w_y + \Delta w_y)/w_y \leq 1 + K(-v^2 + 1/\pi) \leq 1 + 1.1/\pi - (v^{(1)})^2 \leq 1 + 1.1/\pi - (w_y^{(1)})^2, \quad (93)$$

where the first inequality is due to $vw_y \leq \|w\|^2$, the second is due to $K > 1$ and the third is from $v \geq w_y$. Therefore, combining with (92), we can bound $w_y$ in the end of stage 1 as

$$w_y \leq \left(1 + 1.1/\pi - (w_y^{(1)})^2\right)^{\left\lceil \log_{2.56} \frac{\frac{1}{2\pi}}{(w_y^{(1)})^2/2.7} \right\rceil}. \quad (94)$$

Since it initializes as $\|w^{(0)}\| \leq 0.1$, we have $w_y^{(1)} \leq 0.1(1 + 1.1/\pi) = 0.135$. Then, it can be verified that, when $w_y^{(1)} \in (0, 0.135]$, it holds

$$w_y \leq 0.44. \quad (95)$$

The next is to bound $v - w_x$. Combining the update rules of $v$ and $w_x$ in (80, 81), we have

$$\Delta(v - w_x) := (v^{(t+1)} - w_x^{(t+1)}) - (v^{(t)} - w_x^{(t)})$$
$$= K(v - w_x)\left(vw_x - 1 + \frac{\arctan(w_y/w_x) - \frac{w_x w_y}{\|w\|^2}}{\pi}\right) + K\frac{w_y^2}{v}\left(-v^2 + \frac{vw_y}{\pi\|w\|^2}\right). \quad (96)$$

Note that

$$-1 \leq vw_x - 1 + \frac{\arctan(w_y/w_x) - \frac{w_x w_y}{\|w\|^2}}{\pi} \leq -1 + \frac{\arctan(w_y/w_x)}{\pi}, \quad (97)$$

where the left is due to $vw_x > 0$ and , the right is from $\Delta w_y \geq 0$. When $w_y/w_x \leq 2.7$, the RHS follows $-1 + \frac{\arctan(w_y/w_x)}{\pi} \leq -0.6$. So combining both sides tells

$$1 + K\left(vw_x - 1 + \frac{\arctan(w_y/w_x) - \frac{w_x w_y}{\|w\|^2}}{\pi}\right) \in [-K + 1, 0.4] \subset [-0.1, 0.4]. \quad (98)$$

Since $\Delta w_y \geq 0$, we have $0 \leq K\frac{w_y^2}{v}(-v^2 + \frac{vw_y}{\pi\|w\|^2}) \leq \frac{K}{\pi}w_y\frac{w_y^2}{\|w\|^2}$. Note that at initialization $w_x^{(0)} \leq v^{(0)}$. Then it is easy to verify that

$$-0.01 \leq -0.1(v^{(0)} - w^{(0)}) \leq v^{(1)} - w^{(1)} \leq (1 + \frac{K}{\pi} - \frac{K}{2})v^{(0)} \leq 0.082. \quad (99)$$

Because the coefficient on the positive side in (98) is larger than $0.4 > 0.1$, it is appropriate to upper bound the $v - w_x$ as

$$v - w_x \leq \max\left\{0.082, 0.082 \cdot 0.4^T + \sum_{t=1}^{T} 0.4^{t-1}\frac{K}{\pi}w_y^{(t)}\frac{(w_y^{(t)})^2}{\|w^{(t)}\|^2}\right\}$$

$$\leq \max\left\{0.082, 0.082 \cdot 0.4^T + \sum_{t=1}^{T} 0.4^{t-1}\frac{K}{\pi}w_y^{(t)}\frac{1}{1 + \frac{1}{2.7}\left(\frac{1.55}{1+K/\pi}\right)^{2(t-1)}}\right\}$$

$$\leq \max\left\{0.082, 0.082 \cdot 0.4^T + \sum_{t=1}^{T} 0.4^{t-1}\frac{1.1 \cdot 4.4}{\pi}\frac{1}{1 + \frac{1}{2.7}\left(\frac{1.55}{1+1.1/\pi}\right)^{2(t-1)}}\right\},$$

where the second inequality is from the different growth ratios of $w_x$ and $w_y$. Note that here we take all $T \geq 1$ and pick the largest value of RHS to bound $w_y$. It turns out

$$v - w_x \leq 0.17. \quad (100)$$

Furthermore, to lower bound $v - w_x$, since obviously $|v - w_x| \leq 0.17$, it follows

$$v - w_x \geq -0.1 \cdot |v - w_x|_{\max} \geq -0.017. \tag{101}$$

**Case (II). If $\frac{1}{4\pi} \leq vw_x < \frac{1}{2\pi}$ in first step of stage 2:**

Similar to the discussion in Case (I), we are able to compute the number of iterations for $vw_x$ to reach $1/4\pi$. It is at most

$$T_1 \leq \lceil \log_{2.56} \frac{\frac{1}{4\pi}}{(w_y^{(1)})^2/2.7} \rceil. \tag{102}$$

Accordingly, $w_y$ is bounded as

$$w_y \leq \left(1 + 1.1/\pi - (w_y^{(1)})^2\right)^{\lceil \log_{2.56} \frac{\frac{1}{4\pi}}{(w_y^{(1)})^2/2.7} \rceil} \leq 0.37. \tag{103}$$

For simplicity, we just keep the bounds for $v - w_x$ as in Case (I), as

$$v - w_x \in [-0.017, 0.17]. \tag{104}$$

**Case (III). If $vw_x < \frac{1}{4\pi}$ in first step of stage 2:**

From the condition, we know $vw_x < \frac{1}{4\pi}$ as well in the last step of stage 1. Since $\Delta w_y > 0$ in stage 1, it tells

$$\frac{1}{\pi} \frac{w_x w_y}{\|w\|^2} < vw_x \leq \frac{1}{4\pi}, \tag{105}$$

which means

$$\max\{\frac{w_x}{w_y}, \frac{w_y}{w_x}\} \geq 2 + \sqrt{3}. \tag{106}$$

Since $2 + \sqrt{3} > 2.7$, if $w_y/w_x \geq 2 + \sqrt{3}$, then for time 1, $(v^{(1)}, w_x^{(1)}, w_y^{(1)})$ is already in the stage 2. However, it is not possible because $\|w^{(0)}\| = v^{(0)} \leq 0.1$, which means $v^{(1)} w_x^{(1)}$ can not reach $\frac{1}{\pi} \frac{2.7}{1 + 2.7^2}$.

Therefore, the only possible is $\frac{w_x}{w_y} \geq 2 + \sqrt{3}$. In this case, we are able to bound $w_y$ as

$$w_y \leq (2 - \sqrt{3})w_x \leq (2 - \sqrt{3})\left(\sqrt{\frac{1}{4\pi} + 0.0085^2} + 0.0085\right) \leq 0.078, \tag{107}$$

where the second inequality is due to $vw_x \leq \frac{1}{4\pi}$ and $v - w_x \geq -0.017$. Note that here we still use the bound of $v - w_x$ from Case (I), although it is loose somehow but it is enough for our analysis.

We leave the analysis of the bound of number of iterations to the end of this section.

## Stage 2.

In the case (I) of stage 1, where the first step in stage 2 is with $vw_x \geq \frac{1}{2\pi}$, it has $v - w_x \in [-0.017, 0.17]$ and $w_y \leq 0.44$. In the case (II), where the first step of stage 2 is with $vw_x \in [\frac{1}{4\pi}, \frac{1}{2\pi}]$, it has $v - w_x \in [-0.017, 0.17]$ and $w_y \leq 0.37$. In the case (III), where the first step of stage 2 is with $vw_x \in [\frac{1}{4\pi}, \frac{1}{2\pi}]$, it has $v - w_x \in [-0.017, 0.17]$ and $w_y \leq 0.078$.

To upper bound $vw_x$ in the first step of stage 2, there are two candidates. One is from the case (I),

$$v^{(t+1)}w_x^{(t+1)} = vw_x \left(1 + K(1 - vw_x - \frac{\arctan(\frac{w_y}{w_x}) - \frac{w_y/w_x}{1+(w_y/w_x)^2}}{\pi})\right)^2 + K\frac{w_x w_y^2}{v}\left(-v^2 + \frac{vw_y}{\pi \|w\|^2}\right)$$

$$+ K(v - w_x)^2 \left(1 + K(1 - vw_x - \frac{\arctan(\frac{w_y}{w_x}) - \frac{w_y/w_x}{1+(w_y/w_x)^2}}{\pi})\right)$$

$$\leq vw_x \left(1 + K(1 - vw_x)\right)^2 + K\frac{w_x w_y^2}{w_x}\left(-vw_x + \frac{w_x w_y}{\pi \|w\|^2}\right)$$

$$+ K(v - w_x)^2 \left(1 + K(1 - vw_x)\right)$$

$$\leq \frac{1}{2\pi}\left(1 + 1.1(1 - \frac{1}{2\pi})\right)^2 + 1.1 \cdot 0.44^2\left(-\frac{1}{4\pi} + \frac{1}{2\pi}\right) + 1.1 \cdot 0.17^2\left(1 + 1.1(1 - \frac{1}{2\pi})\right)$$

$$\leq 0.668, \tag{108}$$

where we use $vw_x \geq 1/4\pi$, $x/(1 + x^2) \leq 0.5$ for any $x$.

One is from the case (II),

$$v^{(t+1)}w_x^{(t+1)} \leq vw_x \left(1 + K(1 - vw_x)\right)^2 + K\frac{w_x w_y^2}{w_x}\left(-vw_x + \frac{w_x w_y}{\pi \|w\|^2}\right)$$

$$+ K(v - w_x)^2 \left(1 + K(1 - vw_x)\right)$$

$$\leq \frac{1}{4\pi}\left(1 + 1.1(1 - \frac{1}{4\pi})\right)^2 + 1.1 \cdot 0.37^2\left(\frac{1}{2\pi}\right) + 1.1 \cdot 0.17^2\left(1 + 1.1(1 - \frac{1}{4\pi})\right)$$

$$\leq 0.48, \tag{109}$$

where we use $vw_x \leq 1/4\pi$, $x/(1 + x^2) \leq 0.5$ for any $x$.

Therefore, we can see that, in the first step of stage 2,

$$vw_x \leq 0.668. \tag{110}$$

Next we are going to show how the iteration goes in the stage 2. In Case (I), there are three facts:

1. $w_y \leq 0.44$.

2. $v - w_x \in [-0.017, 0.17]$.

3. $vw_x \in [\frac{1}{2\pi}, 0.668]$.

Similarly, in Case (II), there are three facts as well:

1. $w_y \leq 0.37$.

2. $v - w_x \in [-0.017, 0.17]$.

3. $vw_x \in [\frac{1}{4\pi}, \frac{1}{2\pi}]$.

The main idea is to find a basin that any iteration with the above properties (*i.e.*, in the interval) will converge to and then stay in. The method is to iteratively compute the ranges of the variables for several steps, thanks to the narrow range of $K$. Before explicitly computing the ranges, let's write down the computing method, depending on whether or not $vw_x \geq 1$.

Consider any iteration with $vw_x \in [m_1, m_2]$, $v - w_x \in [d_1, d_2]$, $w_y \leq e$, we compute the bounds of $v^{(t+1)}w_x^{(t+1)}, v^{(t+1)} - w_x^{(t+1)}, w_y^{(t+1)}$ in the following process (naturally assuming $d_1 < 0 < d_2$)

1. If $m_1 \geq 1$:

(a) Compute $w_x \geq \sqrt{m_1 + (d_2/2)^2} - d_2/2 \triangleq f$.

(b) Compute $\frac{w_y}{w_x} \leq e/f \triangleq g$.

(c) Compute $\frac{\arctan(w_y/w_x) - \frac{w_x w_y}{\|w\|^2}}{\pi} \leq \frac{\arctan(g) - g/(1+g^2)}{\pi} \triangleq h$.

(d) Compute $v^{(t+1)} w_x^{(t+1)} \geq m_2(1 + 1.1(1 - m_2 - h))^2 + 1.1(1 - m_2 - h)\max\{|d_1|, |d_2|\}^2 - 1.1e^2 m_2$. This is from

$$v^{(t+1)} w_x^{(t+1)} \geq vw_x \left(1 + K(1 - vw_x - h)\right)^2 + K\frac{w_x w_y^2}{v}\left(-v^2 + \frac{vw_y}{\pi \|w\|^2}\right)$$

$$+ K(v - w_x)^2 \left(1 + K(1 - vw_x - h)\right)$$

$$\geq vw_x \left(1 + K(1 - vw_x - h)\right)^2 - Kw_y^2 \cdot vw_x$$

$$+ K(v - w_x)^2 \left(1 + K(1 - vw_x - h)\right).$$

(e) Compute $v^{(t+1)} w_x^{(t+1)} \leq m_1(1 + 1.0(1 - m_1))^2$. This is due to $x(1 + K(1 - x))^2$ decreases with $x$ increasing when $x \geq 1$.

(f) Compute $v^{(t+1)} - w_x^{(t+1)} \in [d_1(1 + 1.1(m_2 - 1 + h) - 1.1e^2 \cdot (\sqrt{m_2 + (d_2/2)^2} + d_2/2)), d_2(1 + 1.1(m_2 - 1 + h))]$. This is due to

$$\Delta v - \Delta w_x = K(v - w_x)\left(vw_x - 1 + \frac{1}{\pi}(\arctan(\alpha) - \frac{w_x w_y}{\|w\|^2})\right) + K\frac{w_y^2}{v}\left(-v^2 + \frac{vw_y}{\pi \|w\|^2}\right),$$

where $vw_x \geq 1$, the last term is between $-Kvw_y^2$ and $0$.

(g) Compute $w_y^{(t+1)} \leq e \cdot \max\{|j_1|, |j_2|\}$, where

$$j_1 = 1 + 1.1\frac{\sqrt{m_1 + (d_2/2)^2} + d_2/2}{\sqrt{m_1 + (d_2/2)^2} - d_2/2} \cdot (-m_2), \tag{111}$$

$$j_2 = 1 + 1.0\frac{\sqrt{m_1 + (d_1/2)^2} - d_1/2}{\sqrt{m_1 + (d_1/2)^2} + d_1/2} \cdot (-m_1 + \frac{1}{2\pi}). \tag{112}$$

This is due to

$$\frac{\Delta w_y}{w_y} = K\frac{v}{w_x}(-vw_x + \frac{1}{\pi}\frac{w_x w_y}{\|w\|^2}),$$

then we would like to have the smallest value as $j_1 - 1$ and the largest value as $j_2 - 1$. Since $w_y$ is always non-negative, taking the maximum absolute value gives the upper bound.

2. If $m_2 < 1$:

(a) Compute $w_x \geq \sqrt{m_1 + (d_2/2)^2} - d_2/2 \triangleq f$.

(b) Compute $\frac{w_y}{w_x} \leq e/f \triangleq g$.

(c) Compute $\frac{\arctan(w_y/w_x) - \frac{w_x w_y}{\|w\|^2}}{\pi} \leq \frac{\arctan(g) - g/(1+g^2)}{\pi} \triangleq h$.

(d) Compute $v^{(t+1)} w_x^{(t+1)} \geq \min_{x \in [m_1, m_2]} x(1 + 1.0(1 - x - h))^2 - 1.1e^2 x$. Compared with the case of $m_1 \geq 1$, we drop the term $1.1(1 - m_2 - h)\max\{|d_1|, |d_2|\}^2$ because it is possible to have $v - w_x = 0$ in some iterations.

(e) Compute $v^{(t+1)} w_x^{(t+1)} \leq \max_{x \in [m_1, m_2]} x(1 + 1.1(1 - x))^2 + 1.1(1 - x)\max\{|d_1|, |d_2|\}^2$. Compared with the case of $m_1 \geq 1$, we add a term depending on the $|v - w_x|_{\max}$ because it enlarges $vw_x$ in the in-balanced case.

(f) Compute $v^{(t+1)} - w_x^{(t+1)} \in [d_1(1 + 1.1(m_2 - 1 + h) - 1.1e^2 \cdot (\sqrt{m_2 + (d_2/2)^2} + d_2/2)), d_2(1 + 1.1(m_2 - 1 + h))]$. In fact, a rigorous left bound should include more terms to select a minimum from. Here it is simple because it keeps $1 + K(m_1 - 1) \geq 0$ in the following computing, so we do not need to worry about the flipping sign of $d_1$ and $d_2$.

(g) Compute $w_y^{(t+1)} \le e \cdot \max\{|j_1|, |j_2|\}$, where $j_1, j_2$ are the same with those in the case of $m_1 \ge 1$.

Therefore, with the above process, we are able to brutally compute the ranges of $v^{(t+1)}w_x^{(t+1)}, v^{(t+1)} - w_x^{(t+1)}, w_y^{(t+1)}$ from the current ranges. Note that this process plays a role of building a mapping from one interval to another interval, which covers all points from the source interval. However, it is loose to some extent because gradient descent is a mapping from a point to another point. The advantage of such a loose method is feasibility of obtaining bounds while losing tightness. To achieve tightness, later we will also include some wisdom in a point-to-point style.

Also note that, a nice way to combine tightness and efficiency in this method is to split and to merge intervals when necessary.

**For Case (I):**

Now we are to compute the ranges starting from the interval where $I = \{w_y \le 0.44, v - w_x \in [-0.017, 0.17], vw_x \in [\frac{1}{2\pi}, 0.668]\}$. First, we split it into three intervals:

1. $I_1 = \{w_y \le 0.44, v - w_x \in [-0.017, 0.17], vw_x \in [0.213, 0.4]\}$.

2. $I_2 = \{w_y \le 0.44, v - w_x \in [-0.017, 0.17], vw_x \in [0.4, 0.668]\}$.

3. $I_{30} = \{w_y \le 0.44, v - w_x \in [-0.017, 0.17], vw_x \in [\frac{1}{2\pi}, 0.213]\}$.

Then, following the above method with splitting and merging intervals, we have

1. Starting from $I_1$,

    (a) Step 1: $I_1$ mapps to $I_3 = \{w_y \le 0.416, v - w_x \in [-0.162, 0.068], vw_x \in [0.55, 1.12131]\}$.

    (b) Step 2: Splitting $I_3$, we have
       i. $I_4 = \{w_y \le 0.416, v - w_x \in [-0.162, 0.068], vw_x \in [0.55, 0.8]\}$.
       ii. $I_5 = \{w_y \le 0.416, v - w_x \in [-0.162, 0.068], vw_x \in [0.8, 0.9]\}$.
       iii. $I_6 = \{w_y \le 0.416, v - w_x \in [-0.162, 0.068], vw_x \in [0.9, 1.0]\}$.
       iv. $I_7 = \{w_y \le 0.416, v - w_x \in [-0.162, 0.068], vw_x \in [1.0, 1.12131]\}$.
       Then, we have
       i. $I_4$ mapps to
          $I_8 = \{w_y \le 0.214, v - w_x \in [-0.309, 0.0545], vw_x \in [0.942, 1.25786]\}$.
       ii. $I_5$ mapps to
          $I_9 = \{w_y \le 0.0966, v - w_x \in [-0.335, 0.0613], vw_x \in [0.880, 1.19649]\}$.
       iii. $I_6$ mapps to
          $I_{10} = \{w_y \le 0.0756, v - w_x \in [-0.362, 0.068], vw_x \in [0.777894, 1.11178]\}$.
       iv. $I_7$ mapps to
          $I_{11} = \{w_y \le 0.134, v - w_x \in [-0.394, 0.0782], vw_x \in [0.595, 1]\}$.

    (c) Step 3: Splitting and merging $I_8, I_9, I_{10}, I_{11}$, we have
       i. $I_{12} = \{w_y \le 0.134, v - w_x \in [-0.394, 0.078], vw_x \in [0.595, 0.777]\}$.
       ii. $I_{13} = \{w_y \le 0.214, v - w_x \in [-0.394, 0.078], vw_x \in [0.777, 1]\}$.
       iii. $I_{14} = \{w_y \le 0.214, v - w_x \in [-0.362, 0.068], vw_x \in [1, 1.11178]\}$.
       iv. $I_{15} = \{w_y \le 0.214, v - w_x \in [-0.309, 0.061], vw_x \in [1.11178, 1.25786]\}$.
       Then, we have
       i. $I_{12}$ mapps to
          $I_{16} = \{w_y \le 0.0372, v - w_x \in [-0.317, 0.061], vw_x \in [1.14493, 1.31246]\}$.
       ii. $I_{13}$ mapps to
          $I_{17} = \{w_y \le 0.0432, v - w_x \in [-0.448, 0.078], vw_x \in [0.943633, 1.24393]\}$.
       iii. $I_{14}$ mapps to
          $I_{18} = \{w_y \le 0.0662, v - w_x \in [-0.462, 0.077], vw_x \in [0.77846, 1]\}$.
       iv. $I_{15}$ mapps to
          $I_{20} = \{w_y \le 0.0998, v - w_x \in [-0.456, 0.0785], vw_x \in [0.550, 0.878]\}$.

2. Starting from $I_2$,

    (a) Step 1: $I_2$ mapps to $I_{21} = \{w_y \leq 0.332, v - w_x \in [-0.205, 0.114], vw_x \in [0.864, 1.25894]\}$

    (b) Step 2: Splitting $I_{21}$, we have

        i. $I_{22} = \{w_y \leq 0.332, v - w_x \in [-0.205, 0.114], vw_x \in [0.864, 1]\}$.

        ii. $I_{23} = \{w_y \leq 0.332, v - w_x \in [-0.205, 0.114], vw_x \in [1, 1.125894]\}$.

        Then, we have

        i. $I_{22}$ mapps to
$I_{24} = \{w_y \leq 0.081, v - w_x \in [-0.336, 0.114], vw_x \in [0.858, 1.14813]\}$.

        ii. $I_{23}$ mapps to
$I_{25} = \{w_y \leq 0.184, v - w_x \in [-0.409, 0.148], vw_x \in [0.463, 1]\}$.

    (c) Step 3: Splitting and merging $I_{24}, I_{25}$, we have

        i. $I_{26} = \{w_y \leq 0.184, v - w_x \in [-0.409, 0.148], vw_x \in [0.463, 1]\}$.

        ii. $I_{27} = \{w_y \leq 0.081, v - w_x \in [-0.336, 0.114], vw_x \in [1, 1.14813]\}$.

        Then, we have

        i. $I_{26}$ mapps to
$I_{28} = \{w_y \leq 0.083, v - w_x \in [-0.452, 0.148], vw_x \in [0.952783, 1.31778]\}$.

        ii. $I_{27}$ mapps to
$I_{29} = \{w_y \leq 0.034, v - w_x \in [-0.399, 0.133], vw_x \in [0.777, 1]\}$.

3. Starting from $I_{30}$,

    (a) Step 1: $I_{30}$ mapps to $I_{31} = \{w_y \leq 0.44, v - w_x \in [-0.124, 0.037], vw_x \in [0.422, 0.767]\}$

    (b) Step 2: Splitting $I_{31}$, we have

        i. $I_{32} = \{w_y \leq 0.44, v - w_x \in [-0.124, 0.037], vw_x \in [0.422, 0.5]\}$.

        ii. $I_{33} = \{w_y \leq 0.44, v - w_x \in [-0.124, 0.037], vw_x \in [0.5, 0.6]\}$.

        iii. $I_{34} = \{w_y \leq 0.44, v - w_x \in [-0.124, 0.037], vw_x \in [0.6, 0.767]\}$.

        Then, we have

        i. $I_{32}$ mapps to
$I_{35} = \{w_y \leq 0.301, v - w_x \in [-0.218, 0.0185], vw_x \in [0.901, 1.20971]\}$.

        ii. $I_{33}$ mapps to
$I_{36} = \{w_y \leq 0.262, v - w_x \in [-0.245, 0.023], vw_x \in [0.96322, 1.25093]\}$.

        iii. $I_{34}$ mapps to
$I_{37} = \{w_y \leq 0.213, v - w_x \in [-0.288, 0.029], vw_x \in [0.947, 1.25345]\}$.

    (c) Step 3: Splitting and merging $I_{35}, I_{36}, I_{37}$, we have

        i. $I_{38} = \{w_y \leq 0.301, v - w_x \in [-0.288, 0.029], vw_x \in [0.901, 1]\}$.

        ii. $I_{39} = \{w_y \leq 0.301, v - w_x \in [-0.288, 0.029], vw_x \in [1, 1.1]\}$.

        iii. $I_{40} = \{w_y \leq 0.301, v - w_x \in [-0.288, 0.029], vw_x \in [1.1, 1.25093]\}$.

        iv. $I_{41} = \{w_y \leq 0.262, v - w_x \in [-0.245, 0.029], vw_x \in [1.25093, 1.25345]\}$.

        Then, we have

        i. $I_{38}$ mapps to
$I_{42} = \{w_y \leq 0.0404, v - w_x \in [-0.392, 0.029], vw_x \in [0.888, 1.11696]\}$.

        ii. $I_{39}$ mapps to
$I_{43} = \{w_y \leq 0.0740, v - w_x \in [-0.428, 0.033], vw_x \in [0.741, 1]\}$.

        iii. $I_{40}$ mapps to
$I_{44} = \{w_y \leq 0.125, v - w_x \in [-0.482, 0.038], vw_x \in [0.497, 0.891]\}$.

        iv. $I_{41}$ mapps to
$I_{45} = \{w_y \leq 0.109, v - w_x \in [-0.400, 0.038], vw_x \in [0.534, 0.702]\}$.

    (d) Step 4: Splitting and merging $I_{42}, I_{43}, I_{44}, I_{45}$, we have

        i. $I_{46} = \{w_y \leq 0.125, v - w_x \in [-0.482, 0.038], vw_x \in [0.497, 0.891]\}$.

        ii. $I_{47} = \{w_y \leq 0.074, v - w_x \in [-0.428, 0.033], vw_x \in [0.891, 1]\}$.

        iii. $I_{48} = \{w_y \leq 0.041, v - w_x \in [-0.40, 0.029], vw_x \in [1, 1.11696]\}$.

        Then, we have

    i. $I_{46}$ mapps to
    $I_{49} = \{w_y \leq 0.0424, v - w_x \in [-0.442, 0.034], vw_x \in [1.07853, 1.34708]\}$.
    ii. $I_{47}$ mapps to
    $I_{50} = \{w_y \leq 0.0110, v - w_x \in [-0.435, 0.033], vw_x \in [0.993, 1.13943]\}$.
    iii. $I_{48}$ mapps to
    $I_{51} = \{w_y \leq 0.0109, v - w_x \in [-0.454, 0.033], vw_x \in [0.497, 0.891]\}$.

**For Case (II):**

Now we are to compute the ranges starting from the interval where $I = \{w_y \leq 0.37, v - w_x \in [-0.017, 0.17], vw_x \in [\frac{1}{4\pi}, \frac{1}{2\pi}]\}$. First, we denote it as

1. $I_{52} = \{w_y \leq 0.37, v - w_x \in [-0.017, 0.17], vw_x \in [\frac{1}{4\pi}, \frac{1}{2\pi}]$.

Then, following the above method with splitting and merging intervals, we have

1. Starting from $I_{52}$,

    (a) Step 1: $I_{52}$ mapps to $I_{53} = \{w_y \leq 0.37, v - w_x \in [-0.079, 0.0271], vw_x \in [0.222, 0.616]\}$.
    (b) Step 2: $I_{53}$ mapps to $I_{54} = \{w_y \leq 0.343, v - w_x \in [-0.171, 0.017], vw_x \in [0.621, 1.24894]\}$.
    (c) Step 3: Splitting $I_{54}$, we have
        i. $I_{55} = \{w_y \leq 0.343, v - w_x \in [-0.171, 0.017], vw_x \in [0.621, 1]\}$.
        ii. $I_{56} = \{w_y \leq 0.343, v - w_x \in [-0.171, 0.017], vw_x \in [1, 1.24894]\}$.
        Then, we have
        i. $I_{55}$ mapps to
        $I_{57} = \{w_y \leq 0.150, v - w_x \in [-0.305, 0.017], vw_x \in [0.840, 1.25908]\}$.
        ii. $I_{56}$ mapps to
        $I_{58} = \{w_y \leq 0.137, v - w_x \in [-0.367, 0.022], vw_x \in [0.472, 1]\}$.
    (d) Step 4: Splitting and merging $I_{57}, I_{58}$, we have
        i.
        ii. $I_{59} = \{w_y \leq 0.150, v - w_x \in [-0.367, 0.022], vw_x \in [0.472, 1]\}$.
        iii.
        iv. $I_{60} = \{w_y \leq 0.150, v - w_x \in [-0.305, 0.017], vw_x \in [1, 1.25908]\}$.
        Then, we have
        i. $I_{59}$ mapps to
        $I_{61} = \{w_y \leq 0.0705, v - w_x \in [-0.393, 0.022], vw_x \in [0.971, 1.304]\}$.
        ii. $I_{60}$ mapps to
        $I_{62} = \{w_y \leq 0.0613, v - w_x \in [-0.421, 0.0219], vw_x \in [0.583, 1]\}$.

**For both Cases (I, II):**

From $I_{16-20}, I_{28}, I_{29}, I_{49-51}, I_{61}, I_{62}$, we can see that it has fallen into an interval $I_f = \{w_y < 0.1, v - w_x \in [-0.462, 0.148], vw_x \in [0.497, 1.34078]\}$. Something special here is that $w_y$ has been much smaller than $w_x$. More broadly, let's define an interval $I_s$ generated by $I_g = \{w_y = 0, v - w_x \in [-0.464, 0.148], vw_x \in [1, 1.5]\}$. Here "generated" means

$$I_s = \bigcup_{T \geq t} \{(v^{(T)}, w_x^{(T)}, w_y^{(T)}) | (v_t^{(t)}, w_x^{(t)}, w_y^{(t)}) \in I_g\}. \tag{113}$$

Then each element $(v, w_x, w_y) \in I_s$ has the following properties:

1. $w_y = 0$.

2. $vw_x \in [0.181, 1.5]$.

3. If $vw_x \leq 1$, then $v - w_x \in [-0.735, 0.23]$. If $vw_x > 1$, then $v - w_x \in [-0.474, 0.148]$.

The first property is obvious. The third can be proven as follows: for each element $(v, w_x, w_y) \in I_g$, it has $v^{(t+1)} - w_x^{(t+1)} = (v - w_x)(1 + K(vw_x - 1))$, where the ratio $1 + K(vw_x - 1) \in [1, 1 + 1.1(1.5 - 1)]$ when $vw_x \in [1, 1.5]$. Furthermore, in the proven 2-D case, we have shown that "if $vw_x > 1$ with some mild conditions, then $\frac{v^{(t+2)} - w_x^{(t+2)}}{v - w_x} \in (-1, 1)$". Actually it can be tighter as $\frac{v^{(t+2)} - w_x^{(t+2)}}{v - w_x} \in (-0.2, 1)$ because here $K \leq 1.1$ while the original bound is for $K \leq 1.5$. The condition of bounded $|v - w_x|$ can also be verified, the purpose of which is to keep $v, w_x$ always positive. Then the bound $[-0.2, 1]$ will tell $v - w_x \in [-0.474, 0.148]$ on $vw_x \geq 1$, because

$$\frac{0.148}{0.474} > 0.2, \qquad \frac{0.474}{0.148} > 0.2.$$

For the second property, the left bound can be verified as

$$\min_{x \in [1, 1.5]} x(1 + 1.1(1 - x))^2 + 1.1(1 - x) \cdot 0.474^2 = \left( x(1 + 1.1(1 - x))^2 + 1.1(1 - x) \cdot 0.474^2 \right)\Big|_{x=1.5}$$
$$\geq 0.181.$$

The right bound can be verified as

$$\max_{x \in [0, 1]} x(1 + 1.1(1 - x))^2 + 1.1(1 - x) * 0.735^2 < 1.5.$$

After proving these three properties, we would like to bound how far $I_f$ is away from $I_s$. More precisely, the distance is measured by $w_y$. We are going to show $w_y$ decays exponentially.

Remind the update rules in (80, 81). Denote $\gamma = \frac{1}{\pi}(\arctan(\alpha) - \frac{w_x w_y}{\|w\|^2})$ again and $\delta = K \frac{w_y^2}{v}(-v^2 + \frac{vw_y}{\pi\|w\|^2})$, then it is

$$\Delta v = Kw_x(-vw_x + 1) - Kw_x\gamma + \delta, \tag{114}$$
$$\Delta w_x = Kv(-vw_x + 1) - Kv\gamma, \tag{115}$$
$$\delta \in [-Kvw_y^2, 0]. \tag{116}$$

Note that both $\gamma$ and $\delta$ are very small, so we are to show their effects separately, which is enough to be a good approximation.

Consider an iteration where $v^{(t)}w_x^{(t)} > 1$ and the corresponding $\gamma^{(t)}$. Let's denote $v^{(t+1)}, w_x^{(t+1)}$ as the next parameters with **no corruption** from $\gamma^{(t)}$. Similarly, we denote $\hat{v}^{(t+1)}, \hat{w}_x^{(t+1)}$ are corrupted with $\gamma^{(t)}$. From the 2-D analysis, we know

$$\frac{v^{(t+2)} - w_x^{(t+2)}}{v^{(t)} - w_x^{(t)}} = (1 + K(v^{(t)}w_x^{(t)} - 1))(1 + K(v^{(t+1)}w_x^{(t+1)} - 1)) < 1. \tag{117}$$

We would like to show, with a small $\gamma^{(t)}$ and ignoring $\delta$,

$$\frac{\hat{v}^{(t+2)} - \hat{w}_x^{(t+2)}}{v^{(t)} - w_x^{(t)}} = (1 + K(v^{(t)}w_x^{(t)} - 1 + \gamma^{(t)}))(1 + K(\hat{v}^{(t+1)}\hat{w}_x^{(t+1)} - 1 + \gamma^{(t+1)})) \lessapprox 1, \tag{118}$$

where $\gamma^{(t+1)}$ is in time $(t+1)$ accordingly. The difference of LHS of the above two expressions turns out to be

$$\begin{aligned}
(118) - (117) &= K\gamma^{(t)}(1 + K(v^{(t+1)}w_x^{(t+1)} - 1)) \\
&\quad + (1 + K(v^{(t)}w_x^{(t)} - 1))K(\hat{v}^{(t+1)}\hat{w}_x^{(t+1)} - v^{(t+1)}w_x^{(t+1)} + \gamma^{(t+1)}) + \mathcal{O}(\gamma^2) \\
&= K\gamma^{(t)}(1 + K(v^{(t+1)}w_x^{(t+1)} - 1)) \\
&\quad + K(1 + K(v^{(t)}w_x^{(t)} - 1))(-K(v^{(t)})^2\gamma^{(t)} - K(w_x^{(t)})^2\gamma^{(t)} + \gamma^{(t+1)}) + \mathcal{O}(\gamma^2) \\
&\leq K\gamma^{(t)}\left(1 + (1 + K(v^{(t)}w_x^{(t)} - 1))\left(-K(v^{(t)})^2 - K(w_x^{(t)})^2 + \frac{\gamma^{(t+1)}}{\gamma^{(t)}}\right)\right) + \mathcal{O}(\gamma^2) \\
&\leq K\gamma^{(t)}\left(1 + (1 + K(v^{(t)}w_x^{(t)} - 1))\left(-2Kv^{(t)}w_x^{(t)} + \frac{\gamma^{(t+1)}}{\gamma^{(t)}}\right)\right) + \mathcal{O}(\gamma^2).
\end{aligned} \tag{119}$$

Since $\frac{\Delta w_x}{w_x} = K \frac{v}{w_x}(-vw_x + 1 - \gamma)$, we have

$$\frac{w_x^{(t+1)}}{w_x^{(t)}} = 1 + K \frac{v^{(t)}}{w_x^{(t)}}(-v^{(t)}w_x^{(t)} + 1 - \gamma^{(t)}) < 1. \tag{120}$$

Also we have

$$\frac{\gamma^{(t+1)}}{\gamma^{(t)}} = \frac{\arctan(\frac{w_y^{(t+1)}}{w_x^{(t+1)}}) - \frac{w_x^{(t+1)}w_y^{(t+1)}}{\|w^{(t+1)}\|^2}}{\arctan(\frac{w_y^{(t)}}{w_x^{(t)}}) - \frac{w_x^{(t)}w_y^{(t)}}{\|w^{(t)}\|^2}}. \tag{121}$$

Since $w_y^{(t+1)} \leq w_y^{(t)}$ and

$$\frac{\arctan(mx) - \frac{mx}{1+m^2x^2}}{\arctan(x) - \frac{x}{1+x^2}} \leq m^3, \text{ for any } m > 0, x > 0, \tag{122}$$

we have

$$\frac{\gamma^{(t+1)}}{\gamma^{(t)}} \leq \frac{1}{\left(1 + K \frac{v^{(t)}}{w_x^{(t)}}(-v^{(t)}w_x^{(t)} + 1 - \gamma^{(t)})\right)^3}. \tag{123}$$

For general $vw_x \in (1, 1.5]$, (123) holds as

$$\frac{\gamma^{(t+1)}}{\gamma^{(t)}} \lesssim \frac{1}{(1 + 1.1\frac{\sqrt{1+0.074^2}+0.074}{\sqrt{1+0.074^2}-0.074}(-1.5+1))^3} \leq 22. \tag{124}$$

Since $1 + K(v^{(t)}w_x^{(t)} - 1) \leq 1 + 1.1 * 0.5 = 1.55$, it is fair to say

$$(118) - (117) \lesssim K\gamma^{(t)}(1 + 1.55 * (-2 + 22)) + \mathcal{O}(\gamma^2) = 35.2\gamma^{(t)} + \mathcal{O}(\gamma^2). \tag{125}$$

Actually $\gamma^{(t)}$ is bounded by

$$\frac{w_y^{(t)}}{w_x^{(t)}} \leq \frac{0.099}{\sqrt{1 + 0.074^2} - 0.074} = 0.1066, \tag{126}$$

$$\gamma^{(t)} \leq \frac{\arctan(x) - \frac{x}{1+x^2}}{\pi} \leq 2.6 \times 10^{-4}. \tag{127}$$

As a result,

$$(118) - (117) \lesssim 0.0084. \tag{128}$$

Note that this small value is very easy to cover in (117), requiring

$$1 - \frac{v^{(t+2)} - w_x^{(t+2)}}{v^{(t)} - w_x^{(t)}} \geq 0.0084, \tag{129}$$

except when $vw_x$ is pretty close to 1. When $vw_x \to 1$, from the analysis of 2-D case, (derived from the case of $x_{t+1}y_{t+1} \geq x_s^2$)

$$1 - \frac{v^{(t+2)} - w_x^{(t+2)}}{v^{(t)} - w_x^{(t)}} \geq (2K - 2)(v^{(t)}w_x^{(t)} - 1). \tag{130}$$

For $(118) - (117)$, denote a function $p(x)$ as

$$p(x) = 1 + (1 + Kx)\left(-2K(x + 1) + \frac{1}{\left(1 + K\frac{v}{w_x}(-x)\right)^3}\right), \tag{131}$$

where $x = v^{(t)} w_x^{(t)} - 1$ in (119, 123). It is obvious that $p(0) = 1 + (-2K + 1) < 0$. When $x$ is small, it turns out

$$p(x) = -2K + 2 + K\left(-2K - 1 + 3\frac{v^{(t)}}{w_x^{(t)}}\right)x + \mathcal{O}(x^2) \tag{132}$$

As a result, $(118) - (117) < 0$ when $vw_x - 1 = o(K - 1)$. What if $vw_x - 1 = \Omega(K - 1)$? Actually, we can get a better bound by a more care analysis, as

$$\frac{(118) - (117)}{K\gamma^{(t)}} \leq 1 + (1 + K(v^{(t)} w_x^{(t)} - 1))\left(-K(v^{(t)})^2 - K(w_x^{(t)})^2 + \frac{\gamma^{(t+1)}}{\gamma^{(t)}}\right)$$
$$+ K\left[v^{(t)} w_x^{(t)}(1 + K(1 - v^{(t)} w_x^{(t)}))^2 - 1\right], \tag{133}$$

where the last term is due to $v^{(t+1)} w_x^{(t+1)} \leq v^{(t)} w_x^{(t)}(1 + K(1 - v^{(t)} w_x^{(t)}))^2$. Hence, with this bound, by expanding the last term, (132) becomes

$$p(x) = -2K + 2 + K\left(-2K - 1 + 3\frac{v^{(t)}}{w_x^{(t)}}\right)x + K(1 - 2K)x + \mathcal{O}(x^2) \tag{134}$$

$$= -2K + 2 + K\left(-4K + 3\frac{v^{(t)}}{w_x^{(t)}}\right)x + \mathcal{O}(x^2), \tag{135}$$

which is definitely negative because

$$\frac{v^{(t)}}{w_x^{(t)}} \leq \frac{\sqrt{1 + 0.074^2} + 0.074}{\sqrt{1 + 0.074^2} - 0.074} < 1.16 < \frac{4}{3}. \tag{136}$$

Meanwhile, we are to prove the $\delta$ in (114) will not make $\tilde{I}_s$ make $v - w_x < -0.474$ starting from $v - w_x \geq -0.462$. First, in the region of $\{vw_x \in [1, 1.5], v - w_x \leq 0.148\}$, we have $Kvw_y^2 \leq 1.1 \cdot (\sqrt{1.5 + 0.074^2} + 0.074) * 0.1^2 \leq 0.0144$. Also note that in this region with $v - w_x \geq -0.462$, we have

$$\frac{w_y^{(t+1)}}{w_y} \leq 1 - \frac{\sqrt{1 + 0.231^2} - 0.231}{\sqrt{1 + 0.231^2} + 0.231} = 0.37. \tag{137}$$

Hence $Kv(w_y^{(t)})^2 + Kv(w_y^{(t+1)})^2 \leq 0.0144 * (1 + 0.37^2) = 0.0164$. Since $|v^{(t+2)} - w^{(t+2)}| < |v^{(t)} - w^{(t)}|$ if there is no $\delta$, we shall see that there is no need to discuss the case of $v - w_x \geq -0.462 + 0.0164 = -0.4456$ because it still holds $v^{(t+1)} - w_x^{(t+1)} > -0.462$. When $v^{(t)} - w_x^{(t)} \in [-0.462, -0.4456]$, we shall see that in (59), after adding the term of $\delta$ in $v$,

$$\frac{v^{(t+2)} - w_x^{(t+2)}}{v^{(t)} - w_x^{(t)}} \leq 1 - (1 + K(vw_x - 1)) \cdot Kw_x\delta, \tag{138}$$

which means the absolute value of $v - w_x$ decays at least by a margin depending on $\delta$. After multiplying the current difference $v^{(t)} - w_x^{(t)}$ on both side, it gives

$$(v^{(t+2)} - w_x^{(t+2)}) - (v^{(t)} - w_x^{(t)}) \geq v^{(t)} w_x^{(t)} w_x^{(t)} \delta. \tag{139}$$

Note that here $v^{(t+2)} - w_x^{(t+2)}$ does not include $\delta^{(t)}$ and $\delta^{(t+1)}$. As stated above, we have $\frac{\delta^{(t+1)}}{\delta^{(t)}} \leq 0.37^2 \leq 0.16$ due to the decay of $w_y$. So it is safe to say $\delta^{(t)} + \delta^{(t+1)} \geq 1.16\delta^{(t)}$. Combining with the above inequality, it gives

$$(v^{(t+2)} - w_x^{(t+2)}) - (v^{(t)} - w_x^{(t)}) + \delta^{(t)} + \delta^{(t+1)} \geq (v^{(t)} w_x^{(t)} w_x^{(t)} + 1.16)\delta^{(t)}, \tag{140}$$

where

$$v^{(t)} w_x^{(t)} w_x^{(t)} + 1.16 \leq vw_x \cdot \left(\sqrt{vw_x + \left(\frac{0.4456}{2}\right)^2} - \frac{0.4456}{2}\right) + 1.16 \leq 0.6. \tag{141}$$

Furthermore, from our previous discussion, $w_y^{(t+2)} < w_y^{(t+2)}$ gives that the sum of (140) is bounded by

$$\frac{0.6}{1-0.16}\delta^{(t)} \geq \frac{0.6}{1-0.16} \cdot (-0.0144) \geq -0.0103. \tag{142}$$

Since $-0.474 - (-0.462) < -0.0103$, we shall see that the term of $\delta$ cannot drive $v - w_x < -0.472$. Note that (140) shall include a factor $(< 1)$ in front of $\delta^{(t)}$, but we have ignored it to show a more aggressive bound.

Therefore, we are able to say an Interval $\hat{I}_s$ generated by $I_f$ also has the following properties: for each element $(v, w_x, w_y) \in \hat{I}_s$,

1. $vw_x \in [0.181, 1.5]$.

2. If $vw_x \leq 1$, then $v - w_x \in [-0.735, 0.23]$. If $vw_x > 1$, then $v - w_x \in [-0.472, 0.148]$.

Then the decreasing ratio of $\Delta w_y / w_y$ is bounded by

$$\frac{\Delta w_y}{w_y} = K\frac{v}{w_x}\left(-vw_x + \frac{1}{\pi}\frac{w_x w_y}{\|w\|^2}\right) \tag{143}$$

$$\in \left[-1.1(\sqrt{1.5 + 0.074^2} + 0.074)^2, -0.030K\right] \tag{144}$$

$$= [-1.87, -0.030K]. \tag{145}$$

Hence, $w_y$ decays with a linear ratio of $0.97$ (or $1 - 0.030K$) at most for Cases (I, II) in stage 2.

For Case (III), in the first step of stage 2, it already has $w_y \leq 0.078$ and $v - w_x \in [-0.017, 0.17]$. So surely it will also converge to $I_s$.

Here we present the time analysis for Case (III) of both stages. The number of iterations in the first stage is apparently similar to that of case (I, II), as

$$T_1 \leq \log_{2.56}\left\lceil\frac{2.7\psi}{\beta^2}\right\rceil, \tag{146}$$

where $\psi < \frac{1}{4\pi}$ is the value of $vw_x$ in the first step of stage 2. In stage 2, since our target is to find how many steps are necessary to get $vw_x \geq 0.181$, so it is

$$v^{(t+1)}w_x^{(t+1)} \geq v^{(t)}w_x^{(t)}\left(1 - 0.181 + 1 - \frac{\arctan(2-\sqrt{3}) - \frac{2-\sqrt{3}}{1+(2-\sqrt{3})^2}}{\pi} - 1.1w_y^2\right) \tag{147}$$

$$\geq 3.28v^{(t)}w_x^{(t)}. \tag{148}$$

where obviously it still holds $\frac{w_y}{w_x} \leq 2 - \sqrt{3}$ and $w_y^2 < 0.1^2$ in stage 2. Since $3.28 > 2.56$, we have the total number of steps to have $vw_x > 0.181$ bounded as

$$\left\lceil\log_{2.56}\frac{2.7\psi}{\beta^2}\right\rceil + \left\lceil\log_{3.28}\frac{0.181}{\psi}\right\rceil \leq \left\lceil\log_{2.56}\frac{0.675}{\pi\beta^2}\right\rceil + \left\lceil\log_{3.28}\frac{0.181}{\frac{1}{4\pi}}\right\rceil + 2$$

$$\leq \left\lceil\log_{2.56}\frac{0.675}{\pi\beta^2}\right\rceil + 3$$

$$< \left\lceil\log_{2.56}\frac{1.35}{\pi\beta^2}\right\rceil + 4,$$

which is not beyond the bound for Cases (I, II). $\square$

## J    PROOF OF MATRIX FACTORIZATION

Consider a two-layer matrix factorization problem, parameterized by learnable weights $\mathbf{X} \in \mathbb{R}^{m\times p}$, $\mathbf{Y} \in \mathbb{R}^{p\times q}$, and the target matrix is $\mathbf{C} \in \mathbb{R}^{m\times q}$. The loss $L$ is defined as

$$L(\mathbf{X}, \mathbf{Y}) = \frac{1}{2}\|\mathbf{XY} - \mathbf{C}\|_F^2. \tag{149}$$

Obviously $\{\mathbf{X}, \mathbf{Y} : \mathbf{X}\mathbf{Y} = \mathbf{C}\}$ forms a minimum manifold. Focusing on this manifold, our targets are: 1) to prove our condition for stable oscillation on 1D functions holds at the minimum of $L$ for any setting of dimensions, and 2) to prove a convergence result with initialization close to a minimum beyond the edge of stability for the symmetric case $\mathbf{Y} = \mathbf{X}^\top$.

### J.1    ASYMMETRIC CASE: 1D FUNCTION AT THE MINIMA

Before looking into the theorem, we would like to clarify the definition of the loss Hessian. Inherently, we squeeze $\mathbf{X}, \mathbf{Y}$ into a vector $\theta = \text{vec}(\mathbf{X}, \mathbf{Y}) \in \mathbb{R}^{mp+pq}$, which vectorizes the concatnation. As a result, we are able to represent the loss Hessian w.r.t. $\theta$ as a matrix in $\mathbb{R}^{(mp+pq) \times (mp+pq)}$. Meanwhile, the support of the loss landscape is in $\mathbb{R}^{mp+pq}$. In the following theorem, we are to show the leading eigenvector $\Delta \triangleq \text{vec}(\Delta\mathbf{X}, \Delta\mathbf{Y}) \in \mathbb{R}^{mp+pq}$ of the loss Hessian. Since the cross section of the loss landscape and $\Delta$ forms a 1D function $f_\Delta$, we would also show the stable-oscillation condition on 1D function holds at the minima of $f_\Delta$.

**Theorem 12.** *For a matrix factorization problem, assume $\mathbf{X}\mathbf{Y} = \mathbf{C}$. Consider SVD of both matrices as $\mathbf{X} = \sum_{i=1}^{\min\{m,p\}} \sigma_{x,i} u_{x,i} v_{x,i}^\top$ and $\mathbf{Y} = \sum_{i=1}^{\min\{p,q\}} \sigma_{y,i} u_{y,i} v_{y,i}^\top$, where both groups of $\sigma_{\cdot,i}$'s are in descending order and both top singular values $\sigma_{x,1}$ and $\sigma_{y,1}$ are unique. Also assume $v_{x,1}^\top u_{y,1} \neq 0$. Then the leading eigenvector of the loss Hessian is $\Delta = \text{vec}(C_1 u_{x,1} u_{y,1}^\top, C_2 v_{x,1} v_{y,1}^\top)$ with $C_1 = \frac{\sigma_{y,1}}{\sqrt{\sigma_{x,1}^2 + \sigma_{y,1}^2}}, C_2 = \frac{\sigma_{x,1}}{\sqrt{\sigma_{x,1}^2 + \sigma_{y,1}^2}}$. Denote $f_\Delta$ as the 1D function at the cross section of the loss landscape and the line following the direction of $\Delta$ passing $\text{vec}(\Delta\mathbf{X}, \Delta\mathbf{Y})$. Then, at the minima of $f_\Delta$, it satisfies*

$$3[f_\Delta^{(3)}]^2 - f_\Delta^{(2)} f_\Delta^{(4)} > 0. \tag{150}$$

*Proof.* To obtain the direction of the leading Hessian eigenvector at parameters $(\mathbf{X}, \mathbf{Y})$, consider a small deviation of the parameters as $(\mathbf{X} + \Delta\mathbf{X}, \mathbf{Y} + \Delta\mathbf{Y})$. With $\mathbf{X}\mathbf{Y} = \mathbf{C}$, evaluate the loss function as

$$L(\mathbf{X} + \Delta\mathbf{X}, \mathbf{Y} + \Delta\mathbf{Y}) = \frac{1}{2} \|\Delta\mathbf{X}\mathbf{Y} + \mathbf{X}\Delta\mathbf{Y} + \Delta\mathbf{X}\Delta\mathbf{Y}\|_F^2. \tag{151}$$

Expand these terms and split them by orders of $\Delta\mathbf{X}, \Delta\mathbf{Y}$ as follows:

$$\Theta(\|\Delta\mathbf{X}\|^2 + \|\Delta\mathbf{Y}\|^2): \quad \frac{1}{2} \|\Delta\mathbf{X}\mathbf{Y} + \mathbf{X}\Delta\mathbf{Y}\|_F^2, \tag{152}$$

$$\Theta(\|\Delta\mathbf{X}\|^3 + \|\Delta\mathbf{Y}\|^3): \quad \langle\Delta\mathbf{X}\mathbf{Y} + \mathbf{X}\Delta\mathbf{Y}, \Delta\mathbf{X}\Delta\mathbf{Y}\rangle, \tag{153}$$

$$\Theta(\|\Delta\mathbf{X}\|^4 + \|\Delta\mathbf{Y}\|^4): \quad \frac{1}{2} \|\Delta\mathbf{X}\Delta\mathbf{Y}\|_F^2. \tag{154}$$

From the second-order terms, the leading eigenvector of $\nabla^2 L$ is the solution of

$$\text{vec}(\Delta\mathbf{X}, \Delta\mathbf{Y}) = \underset{\|\Delta\mathbf{X}\|_F^2 + \|\Delta\mathbf{Y}\|_F^2 = 1}{\arg\max} \|\Delta\mathbf{X}\mathbf{Y} + \mathbf{X}\Delta\mathbf{Y}\|_F^2. \tag{155}$$

Since both the top singular values of $\mathbf{X}, \mathbf{Y}$ are unique, the solution shall have both $\Delta\mathbf{X}, \Delta\mathbf{Y}$ of rank 1. Actually the solution is (here for simplicity we eliminate the sign of both)

$$\Delta\mathbf{X} = \frac{\sigma_{y,1}}{\sqrt{\sigma_{x,1}^2 + \sigma_{y,1}^2}} u_{x,1} u_{y,1}^\top, \quad \Delta\mathbf{Y} = \frac{\sigma_{x,1}}{\sqrt{\sigma_{x,1}^2 + \sigma_{y,1}^2}} v_{x,1} v_{y,1}^\top. \tag{156}$$

Equipped with the top eigenvector of Hessian, $\text{vec}(\Delta\mathbf{X}, \Delta\mathbf{Y})$, we consider the 1-D function $f_\Delta$ generated by the cross-section of the loss landscape and the eigenvector, passing the minima $(\mathbf{X}, \mathbf{Y})$. Define the function as

$$f_\Delta(\mu) = L(\mathbf{X} + \mu\Delta\mathbf{X}, \mathbf{Y} + \mu\Delta\mathbf{Y}), \quad \mu \in \mathbb{R}. \tag{157}$$

Then, around $\mu = 0$, we have

$$f_\Delta(\mu) = \frac{1}{2} \|\Delta\mathbf{X}\mathbf{Y} + \mathbf{X}\Delta\mathbf{Y}\|_F^2 \cdot \mu^2 + \langle\Delta\mathbf{X}\mathbf{Y} + \mathbf{X}\Delta\mathbf{Y}, \Delta\mathbf{X}\Delta\mathbf{Y}\rangle \cdot \mu^3 + \frac{1}{2} \|\Delta\mathbf{X}\Delta\mathbf{Y}\|_F^2 \cdot \mu^4. \tag{158}$$

Therefore, the several order derivatives of $f_\Delta(\mu)$ at $\mu = 0$ can be obtained from Taylor expansion as

$$f_\Delta^{(2)}(0) = \|\Delta\mathbf{X}\mathbf{Y} + \mathbf{X}\Delta\mathbf{Y}\|_F^2, \tag{159}$$

$$f_\Delta^{(3)}(0) = 6\langle\Delta\mathbf{X}\mathbf{Y} + \mathbf{X}\Delta\mathbf{Y}, \Delta\mathbf{X}\Delta\mathbf{Y}\rangle, \tag{160}$$

$$f_\Delta^{(4)}(0) = 12\|\Delta\mathbf{X}\Delta\mathbf{Y}\|_F^2. \tag{161}$$

Then we compute the condition of stable oscillation of 1-D function as

$$\left[3[f_\Delta^{(3)}]^2 - f_\Delta^{(2)}f_\Delta^{(4)}\right](0) = 108\langle\Delta\mathbf{X}\mathbf{Y} + \mathbf{X}\Delta\mathbf{Y}, \Delta\mathbf{X}\Delta\mathbf{Y}\rangle^2 - 12\|\Delta\mathbf{X}\mathbf{Y} + \mathbf{X}\Delta\mathbf{Y}\|_F^2\|\Delta\mathbf{X}\Delta\mathbf{Y}\|_F^2 \tag{162}$$

$$= 96\|\Delta\mathbf{X}\mathbf{Y} + \mathbf{X}\Delta\mathbf{Y}\|_F^2\|\Delta\mathbf{X}\Delta\mathbf{Y}\|_F^2 > 0, \tag{163}$$

because all of $\Delta\mathbf{X}\mathbf{Y}, \mathbf{X}\Delta\mathbf{Y}, \Delta\mathbf{X}\Delta\mathbf{Y}$ are parallel to $u_{x,1}v_{y,1}^\top$ and $v_{x,1}^\top u_{y,1} \neq 0$.

$\square$

## J.2 SYMMETRIC CASE: CONVERGENCE ANALYSIS AROUND THE MINIMA

In this section, we focus on the symmetric case of matrix factorization where $\mathbf{Y} = \mathbf{X}^\top$. Accordingly, we rescale the loss function as $L(\mathbf{X}, \mathbf{Y}) = \frac{1}{4}\|\mathbf{X}\mathbf{X}^\top - \mathbf{C}\|_F^2$. Denote the target as $\mathbf{C} = \mathbf{X}_0\mathbf{X}_0^\top$, and assume we are around the minima $\mathbf{X}_1 = \mathbf{X}_0 + \Delta\mathbf{X}_1$ with small $\|\Delta\mathbf{X}_1\|$. Consider SVD as $\mathbf{X}_0 = \sum_{i=1}^{\min\{m,p\}} \sigma_i u_i v_i^\top \triangleq \sigma_1 u_1 v_1^\top + \widetilde{\mathbf{X}}_0$. Then the EoS-learning-rate threshold at $\mathbf{X} = \mathbf{X}_0$ is $\eta = \frac{1}{\sigma_1^2}$. Therefore, we are to show the convergence initializing from $\mathbf{X}_1 = \mathbf{X}_0 + \Delta\mathbf{X}_1$ with learning rate $\eta = \frac{1}{\sigma_1^2} + \beta$ with $\beta > 0$.

**Theorem 13.** *Consider the above symmetric matrix factorization with learning rate $\eta = \frac{1}{\sigma_1^2} + \beta$. Assume $0 < \beta\sigma_1^2 < \sqrt{4.5} - 1 \approx 1.121$ and $\eta\sigma_2^2 < 1$. The initialization is around the minimum, as $\mathbf{X}_1 = \mathbf{X}_0 + \Delta\mathbf{X}_1$, with the deviation satisfying $u_1^\top\Delta\mathbf{X}_1 v_1 \neq 0$ and $\|\Delta\mathbf{X}_1\| \leq \epsilon$ bounded by a small value. Then GD would converge to a period-2 orbit $\gamma_\eta$ by a small margin in $O(\epsilon)$, as*

$$\mathbf{X}_t \to \gamma_\eta + \Delta\mathbf{X}, \quad \|\Delta\mathbf{X}\| = O(\epsilon), \tag{164}$$

$$\gamma_\eta = (\mathbf{X}_0 + \delta_1\sigma_1 u_1 v_1^\top, \mathbf{X}_0 + \delta_2\sigma_1 u_1 v_1^\top), \tag{165}$$

*where $\delta_1 \in (0, 1), \delta_2 \in (-1, 0)$ are the two solutions of*

$$1 + \beta\sigma_1^2 = \frac{1}{(\delta+1)^2\left(\sqrt{\frac{1}{(\delta+1)^2} - \frac{3}{4}} + \frac{1}{2}\right)}. \tag{166}$$

*Proof.* The update rule of gradient descent gives

$$\mathbf{X}_{t+1} = \mathbf{X}_t - \eta\left(\mathbf{X}_t\mathbf{X}_t^\top - \mathbf{X}_0\mathbf{X}_0^\top\right)\mathbf{X}_t. \tag{167}$$

Denoting $\Delta\mathbf{X}_t = \mathbf{X}_t - \mathbf{X}_0$, the update rule is equivalent to

$$\Delta\mathbf{X}_{t+1} = \Delta\mathbf{X}_t - \eta\left(\Delta\mathbf{X}_t\mathbf{X}_0^\top + \mathbf{X}_0\Delta\mathbf{X}_t^\top + \Delta\mathbf{X}_t\Delta\mathbf{X}_t^\top\right)(\mathbf{X}_0 + \Delta\mathbf{X}_t) \tag{168}$$

Consider a decomposition of $\Delta\mathbf{X}_t = \epsilon_t\tilde{u}_t v_1^\top + \widetilde{\Delta\mathbf{X}_t}$ where $\widetilde{\Delta\mathbf{X}_t}v_1 = \mathbf{0}$ and $\|\tilde{u}_t\| = 1$. We also control the sign of $\tilde{u}_t$ by claiming $\langle\tilde{u}_t, u_1\rangle > 0$. Then, the update rule is again equivalent to

$$\Delta\mathbf{X}_{t+1} = \epsilon_t\tilde{u}_t v_1^\top + \widetilde{\Delta\mathbf{X}_t} - \eta\Bigg[\left(\epsilon_t\tilde{u}_t v_1^\top + \widetilde{\Delta\mathbf{X}_t}\right)\mathbf{X}_0^\top + \mathbf{X}_0\left(\epsilon_t\tilde{u}_t v_1^\top + \widetilde{\Delta\mathbf{X}_t}\right)^\top + \tag{169}$$

$$\left(\epsilon_t\tilde{u}_t v_1^\top + \widetilde{\Delta\mathbf{X}_t}\right)\left(\epsilon_t\tilde{u}_t v_1^\top + \widetilde{\Delta\mathbf{X}_t}\right)^\top\Bigg]\left(\mathbf{X}_0 + \epsilon_t\tilde{u}_t v_1^\top + \widetilde{\Delta\mathbf{X}_t}\right). \tag{170}$$

After expanding $\mathbf{X}_0 = \sigma_1 u_1 v_1^\top + \widetilde{\mathbf{X}}_0$ and projecting $\Delta\mathbf{X}_{t+1}$ onto $v_1$, we have

$$\Delta\mathbf{X}_{t+1}v_1 = \epsilon_t \tilde{u}_t - \eta\Big[\sigma_1^2 \epsilon_t \tilde{u}_t + \sigma_1 \epsilon_t^2 \tilde{u}_t u_1^\top \tilde{u} + \epsilon_t^3 \tilde{u}_t + \sigma_1^2 \epsilon_t u_1 \tilde{u}_t^\top u_1 + \sigma_1 \epsilon_t^2 u_1 + \sigma_1 \epsilon_t^2 \tilde{u}_t \tilde{u}_t^\top u_1 \tag{171}$$

$$+ \sigma_1 \widetilde{\Delta\mathbf{X}_t}\widetilde{\Delta\mathbf{X}_t}^\top u_1 + \sigma_1 \widetilde{\mathbf{X}}_0 \widetilde{\Delta\mathbf{X}_t}^\top u_1 + \epsilon_t \widetilde{\Delta\mathbf{X}_t}\widetilde{\mathbf{X}}_0^\top \tilde{u}_t + \epsilon_t \widetilde{\Delta\mathbf{X}_t}\widetilde{\Delta\mathbf{X}_t}^\top \tilde{u}_t + \epsilon_t \widetilde{\mathbf{X}}_0 \widetilde{\Delta\mathbf{X}_t}^\top \tilde{u}_t \Big]. \tag{172}$$

**Phase I: $\tilde{u}_t$ gets close to $u_1$ sharply from a random direction**

For now, let's assume $\left\|\widetilde{\Delta\mathbf{X}_t}\right\|$ stays small (which we will show later), then ignoring high-order small values gives

$$\epsilon_{t+1}\tilde{u}_{t+1} = \epsilon_t \tilde{u}_t - \eta\left(\sigma_1^2 \epsilon_t \tilde{u}_t + \sigma_1^2 \epsilon_t u_1 \tilde{u}_t^\top u_1 + \sigma_1 \widetilde{\mathbf{X}}_0 \widetilde{\Delta\mathbf{X}_t}^\top u_1\right) + O(\epsilon^2), \tag{173}$$

$$\langle \epsilon_{t+1}\tilde{u}_{t+1}, u_1\rangle = (1 - 2\eta\sigma_1^2)\langle \epsilon_t \tilde{u}_t, u_1\rangle + O(\epsilon^2) = (-1 - \beta\sigma_1^2)\langle \epsilon_t \tilde{u}_t, u_1\rangle + O(\epsilon^2), \tag{174}$$

$$m_{t+1} \triangleq \epsilon_{t+1}\tilde{u}_{t+1} - \langle \epsilon_{t+1}\tilde{u}_{t+1}, u_1\rangle u_1 = (1 - \eta\sigma_1^2)(\epsilon_t \tilde{u}_t - \langle \epsilon_t \tilde{u}_t, u_1\rangle u_1) - \eta\sigma_1 \widetilde{\mathbf{X}}_0 \widetilde{\Delta\mathbf{X}_t}^\top u_1 + O(\epsilon^2), \tag{175}$$

where (174) is the projection of $\Delta\mathbf{X}_{t+1}$ onto $u_1 v_1^\top$, and (175) is the orthogonal-to-$u_1$ component of $\Delta\mathbf{X}_{t+1}v_1$. Meanwhile, we have the following iteration of $\widetilde{\Delta\mathbf{X}_t}$ following the update rule,

$$\widetilde{\Delta\mathbf{X}_{t+1}} = \widetilde{\Delta\mathbf{X}_t} - \eta\Big[\sigma_1 \epsilon_t \tilde{u}_t u_1^\top \widetilde{\Delta\mathbf{X}_t} + \widetilde{\Delta\mathbf{X}_t}\widetilde{\mathbf{X}}_0^\top \widetilde{\mathbf{X}}_0 + \widetilde{\Delta\mathbf{X}_t}\widetilde{\mathbf{X}}_0^\top \widetilde{\Delta\mathbf{X}_t} + \sigma_1 \epsilon_t u_1 \tilde{u}_t^\top \widetilde{\mathbf{X}}_0 \tag{176}$$

$$+ \sigma_1 \epsilon_t u_1 \tilde{u}_t^\top \widetilde{\Delta\mathbf{X}_t} + \widetilde{\mathbf{X}}_0 \widetilde{\Delta\mathbf{X}_t}^\top \widetilde{\mathbf{X}}_0 + \widetilde{\mathbf{X}}_0 \widetilde{\Delta\mathbf{X}_t}^\top \widetilde{\Delta\mathbf{X}_t} + \epsilon_t^2 \tilde{u}_t \tilde{u}_t^\top \widetilde{\mathbf{X}}_0 \tag{177}$$

$$+ \epsilon_t^2 \tilde{u}_t \tilde{u}_t^\top \widetilde{\Delta\mathbf{X}_t} + \widetilde{\Delta\mathbf{X}_t}\widetilde{\Delta\mathbf{X}_t}^\top \widetilde{\mathbf{X}}_0 + \widetilde{\Delta\mathbf{X}_t}\widetilde{\Delta\mathbf{X}_t}^\top \widetilde{\Delta\mathbf{X}_t}\Big]. \tag{178}$$

Ignoring higher-order small values, it is

$$\widetilde{\Delta\mathbf{X}_{t+1}} = \widetilde{\Delta\mathbf{X}_t} - \eta\left(\sigma_1 \epsilon_t u_1 \tilde{u}_t^\top \widetilde{\mathbf{X}}_0 + \widetilde{\Delta\mathbf{X}_t}\widetilde{\mathbf{X}}_0^\top \widetilde{\mathbf{X}}_0 + \widetilde{\mathbf{X}}_0 \widetilde{\Delta\mathbf{X}_t}^\top \widetilde{\mathbf{X}}_0\right) + O(\epsilon^2). \tag{179}$$

Now we are to verify two facts:

1. The orthogonal-to-$u_1$ component of $\Delta\mathbf{X}_t v_1$, denoted as $m_t$, stays small. Then combining the exponential growth of parallel-to-$u_1$ component in (174) gives $\langle \tilde{u}_t, u_1\rangle \to 1$ quickly.

2. $\left\|\widetilde{\Delta\mathbf{X}_t}\right\|$ stays small.

First is the bound of $\|m_t\|$. From (175), we have $m_{t+1} = (1 - \eta\sigma_1^2)m_t - \eta\sigma_1 \widetilde{\mathbf{X}}_0 \widetilde{\Delta\mathbf{X}_t}^\top u_1 + O(\epsilon^2)$. Combining with (179) and noticing $u_1^\top \widetilde{\mathbf{X}}_0 = \mathbf{0}$, it holds

$$m_{t+1} \approx (1 - \eta\sigma_1^2)m_t - \eta\sigma_1 \widetilde{\mathbf{X}}_0 \widetilde{\Delta\mathbf{X}_t}^\top u_1 \tag{180}$$

$$\sigma_1 \widetilde{\mathbf{X}}_0 \widetilde{\Delta\mathbf{X}_{t+1}}^\top u_1 \approx \sigma_1 \widetilde{\mathbf{X}}_0 \widetilde{\Delta\mathbf{X}_t}^\top u_1 - \eta\sigma_1 \left(\sigma_1 \epsilon_t \widetilde{\mathbf{X}}_0 \widetilde{\mathbf{X}}_0^\top \tilde{u}_t + \widetilde{\mathbf{X}}_0 \widetilde{\mathbf{X}}_0^\top \widetilde{\mathbf{X}}_0 \widetilde{\Delta\mathbf{X}_t}^\top u_1\right) \tag{181}$$

$$= \left(\mathbf{I} - \eta\widetilde{\mathbf{X}}_0 \widetilde{\mathbf{X}}_0^\top\right)\sigma_1 \widetilde{\mathbf{X}}_0 \widetilde{\Delta\mathbf{X}_t}^\top u_1 - \eta\sigma_1^2 \epsilon_t \widetilde{\mathbf{X}}_0 \widetilde{\mathbf{X}}_0^\top \tilde{u}_t. \tag{182}$$

Furthermore, we have $\epsilon_t \widetilde{\mathbf{X}_0}^\top \tilde{u}_t = \widetilde{\mathbf{X}_0}^\top m_t$ due to $\widetilde{\mathbf{X}_0}^\top u_1 = \mathbf{0}$, so (182) can be rewritten as

$$\sigma_1 \widetilde{\mathbf{X}_0} \widetilde{\Delta \mathbf{X}_{t+1}}^\top u_1 \approx \left(\mathbf{I} - \eta \widetilde{\mathbf{X}_0} \widetilde{\mathbf{X}_0}^\top\right) \sigma_1 \widetilde{\mathbf{X}_0} \widetilde{\Delta \mathbf{X}_t}^\top u_1 - \eta \sigma_1^2 \widetilde{\mathbf{X}_0} \widetilde{\mathbf{X}_0}^\top m_t. \tag{183}$$

Combining (180, 183) gives a form of $m_{t+1}$ as a combination of all previous terms as

$$m_{t+1} \approx (1 - \eta \sigma_1^2) m_t + \sum_{i=1}^{t} \eta^2 \sigma_1^2 \left(\mathbf{I} - \eta \widetilde{\mathbf{X}_0} \widetilde{\mathbf{X}_0}^\top\right)^{i-1} \widetilde{\mathbf{X}_0} \widetilde{\mathbf{X}_0}^\top m_{t-i}. \tag{184}$$

Since our goal is to verify that $\|m_t\|$ is bounded, pick any eigenvector $v_p$ of $\widetilde{\mathbf{X}_0} \widetilde{\mathbf{X}_0}^\top$ with associated eigenvalue $\lambda_p$. Then we have

$$\langle m_{t+1}, v_p \rangle = (1 - \eta \sigma_1^2) \langle m_t, v_p \rangle + \sum_{i=1}^{t} \eta^2 \sigma_1^2 v_p^\top \left(\mathbf{I} - \eta \widetilde{\mathbf{X}_0} \widetilde{\mathbf{X}_0}^\top\right)^{i-1} \widetilde{\mathbf{X}_0} \widetilde{\mathbf{X}_0}^\top m_{t-i} \tag{185}$$

$$= (1 - \eta \sigma_1^2) \langle m_t, v_p \rangle + \eta^2 \sigma_1^2 \sum_{i=1}^{t} (1 - \eta \lambda_p)^{i-1} \lambda_p \langle m_{t-i}, v_p \rangle. \tag{186}$$

Obviously $\langle m_{t+1}, v_p \rangle$ shall converge to a series with exponential growth. Assume the ratio as $\langle m_{t+1}, v_p \rangle / \langle m_t, v_p \rangle = r$. The above equation is equivalent to

$$r = -\beta \sigma_1^2 + \eta^2 \sigma_1^2 \sum_{i=1}^{t} (1 - \eta \lambda_p)^{i-1} \lambda_p r^{-i} = -\beta \sigma_1^2 + \frac{\eta^2 \sigma_1^2 \lambda_p / r}{1 - (1 - \eta \lambda_p)/r} \tag{187}$$

$$= -\beta \sigma_1^2 + \frac{\eta^2 \sigma_1^2 \lambda_p}{r - 1 + \eta \lambda_p}. \tag{188}$$

The solutions for this equation are $r = 1$ or $r = 1 - \eta \lambda_p - \eta \sigma_1^2$, both of which are in $[-1, 1]$ once $\lambda_p \leq (1 - \beta \sigma_1^2)/\eta$. Hence, it is safe to say $\langle m_{t+1}, v_p \rangle$ is bounded as non-increasing. In fact, $|\langle m_{t+1}, v_p \rangle|$ is bounded by $|\langle m_0, v_p \rangle|$ because $|\langle m_1, v_p \rangle| < |\langle m_0, v_p \rangle|$ due to the scaling factor $\beta \sigma_1^2 < 1$. Therefore, after picking any eigenvector $v_p$ of $\widetilde{\mathbf{X}_0} \widetilde{\mathbf{X}_0}^\top$, we can conclude $\|m_t\| \leq \|m_0\| \leq |\epsilon_0|$. A further result is that $|\langle m_1, v_p \rangle| \propto \lambda_p$. Notice that

$$\langle m_{t+1}, v_p \rangle + \beta \sigma_1^2 \langle m_t, v_p \rangle = \eta^2 \sigma_1^2 \sum_{i=1}^{t} (1 - \eta \lambda_p)^{i-1} \lambda_p \langle m_{t-i}, v_p \rangle, \tag{189}$$

where RHS is a constant due to the proven non-increasing $|\langle m_t, v_p \rangle|$. But this constant is proportional to $\lambda_p$ because $\langle m_2, v_p \rangle + \beta \sigma_1^2 \langle m_1, v_p \rangle = \eta \sigma_1^2 \lambda_p \langle m_0, v_p \rangle \propto \lambda_p$ and all further iterations follow a similar factor. Therefore, we have $|\langle m_1, v_p \rangle| \propto \lambda_p$.

Now let's show $\left\|\widetilde{\Delta \mathbf{X}_t}\right\|$ stays small. Consider $u_1^\top \widetilde{\Delta \mathbf{X}_t}$, (179) gives

$$u_1^\top \widetilde{\Delta \mathbf{X}_{t+1}} = u_1^\top \widetilde{\Delta \mathbf{X}_t} \left(\mathbf{I} - \eta \widetilde{\mathbf{X}_0}^\top \widetilde{\mathbf{X}_0}\right) - \eta \sigma_1 m_t^\top \widetilde{\mathbf{X}_0} \tag{190}$$

$$= \sum_{i=1}^{t} -\eta \sigma_1 m_{t-i+1}^\top \widetilde{\mathbf{X}_0} \left(\mathbf{I} - \eta \widetilde{\mathbf{X}_0}^\top \widetilde{\mathbf{X}_0}\right)^{i-1}, \tag{191}$$

so the norm is bounded as, for some $\sigma$ of singular value of $\widetilde{\mathbf{X}_0}$,

$$\left\|u_1^\top \widetilde{\Delta \mathbf{X}_{t+1}}\right\| \leq \eta \sigma_1 |\epsilon_0| \frac{\sigma}{1 - (1 - \eta \sigma^2)} C \sigma^2 = |\epsilon_0| \sigma_1 C \sigma, \tag{192}$$

where $C\sigma^2$ are from the previous discussion of $|\langle m_1, v_p \rangle| \propto \lambda_p$, which follows $\sigma = \sqrt{\lambda_p}$. Hence, it is fair to say $\left\|u_1^\top \widetilde{\Delta \mathbf{X}_t}\right\|$ stays small.

Meanwhile, the residual component of $\widetilde{\Delta\mathbf{X}_t}$ that is orthogonal to $u_1$ on the left, denoted as $\widetilde{\Delta\mathbf{X}_{t,\perp}}$, iterates following

$$\widetilde{\Delta\mathbf{X}_{t+1,\perp}} = \widetilde{\Delta\mathbf{X}_{t,\perp}} - \eta\left(\widetilde{\Delta\mathbf{X}_{t,\perp}}\widetilde{\mathbf{X}_0}^\top\widetilde{\mathbf{X}_0} + \widetilde{\mathbf{X}_0}\widetilde{\Delta\mathbf{X}_{t,\perp}}^\top\widetilde{\mathbf{X}_0}\right) \qquad (193)$$

$$\widetilde{\Delta\mathbf{X}_{t+1,\perp}}\widetilde{\mathbf{X}_0}^\top + \widetilde{\mathbf{X}_0}\widetilde{\Delta\mathbf{X}_{t+1,\perp}}^\top = \widetilde{\Delta\mathbf{X}_{t,\perp}}\widetilde{\mathbf{X}_0}^\top + \widetilde{\mathbf{X}_0}\widetilde{\Delta\mathbf{X}_{t,\perp}}^\top - \eta\left[\widetilde{\Delta\mathbf{X}_{t,\perp}}\widetilde{\mathbf{X}_0}^\top\widetilde{\mathbf{X}_0}\widetilde{\mathbf{X}_0}^\top \quad (194)\right.$$

$$\left. + \widetilde{\mathbf{X}_0}\widetilde{\Delta\mathbf{X}_{t,\perp}}^\top\widetilde{\mathbf{X}_0}\widetilde{\mathbf{X}_0}^\top + \widetilde{\mathbf{X}_0}\widetilde{\mathbf{X}_0}^\top\widetilde{\Delta\mathbf{X}_{t,\perp}}\widetilde{\mathbf{X}_0}^\top + \widetilde{\mathbf{X}_0}\widetilde{\mathbf{X}_0}^\top\widetilde{\mathbf{X}_0}\widetilde{\Delta\mathbf{X}_{t,\perp}}^\top\right] \tag{195}$$

$$= \left(\widetilde{\Delta\mathbf{X}_{t,\perp}}\widetilde{\mathbf{X}_0}^\top + \widetilde{\mathbf{X}_0}\widetilde{\Delta\mathbf{X}_{t,\perp}}^\top\right)\left(0.5\mathbf{I} - \eta\widetilde{\mathbf{X}_0}\widetilde{\mathbf{X}_0}^\top\right) \tag{196}$$

$$+ \left(0.5\mathbf{I} - \eta\widetilde{\mathbf{X}_0}\widetilde{\mathbf{X}_0}^\top\right)\left(\widetilde{\Delta\mathbf{X}_{t,\perp}}\widetilde{\mathbf{X}_0}^\top + \widetilde{\mathbf{X}_0}\widetilde{\Delta\mathbf{X}_{t,\perp}}^\top\right). \tag{197}$$

As a result, due to $\left\|\widetilde{\mathbf{X}_0}\widetilde{\mathbf{X}_0}^\top\right\| \leq \sigma_2^2 < 1/\eta$, the following norm is recursively bounded as

$$\left\|\widetilde{\Delta\mathbf{X}_{t+1,\perp}}\widetilde{\mathbf{X}_0}^\top + \widetilde{\mathbf{X}_0}\widetilde{\Delta\mathbf{X}_{t+1,\perp}}^\top\right\| \leq \left\|\widetilde{\Delta\mathbf{X}_{t,\perp}}\widetilde{\mathbf{X}_0}^\top + \widetilde{\mathbf{X}_0}\widetilde{\Delta\mathbf{X}_{t,\perp}}^\top\right\|. \tag{198}$$

Since $\widetilde{\Delta\mathbf{X}_{t+1,\perp}}$ is a polynomial of $\widetilde{\mathbf{X}_0}$ and its transpose, the above bound tells that $\left\|\widetilde{\Delta\mathbf{X}_{t+1,\perp}}\right\|$ does not grow, which means

$$\left\|\widetilde{\Delta\mathbf{X}_{t+1,\perp}}\right\| \leq \left\|\widetilde{\Delta\mathbf{X}_{t,\perp}}\right\|. \tag{199}$$

Combining (192, 199), we can conclude that $\left\|\widetilde{\Delta\mathbf{X}_t}\right\|$ stays small.

After proving that both $m_t$ and $\left\|\widetilde{\Delta\mathbf{X}_t}\right\|$ stay small, from (174), the only term growing fast is $\langle\epsilon_t\tilde{u}_t, u_1\rangle$ exponentially, which means the projection of , $\tilde{u}_t$ onto $u_1$ is also growing sharply.

**Phase II: after $\tilde{u}_1 \parallel u_1$ approximately**

In the first phase, $\langle\epsilon_t\tilde{u}_t, u_1\rangle$ grows exponentially with all other components in $\widetilde{\Delta\mathbf{X}_t}$ keeping small. The consequences of such a growth are

1. $\tilde{u}_1$ gets close to $u_1$ in direction, with the cosine similarity between them growing like $\cos(\arctan(\exp(t)))$, where $t$ is a time variable starting from some constant.

2. while $\tilde{u}_t$ is a unit vector, the growth of $\langle\epsilon_t\tilde{u}_t, u_1\rangle$ makes $\epsilon_t$ grow sharply as well. In the phase I, the proof is strongly dependent of the assumption that $\epsilon_t$ is small. But in the phase II, $\epsilon_t$ is not small any more.

We would like to assume that the second consequence comes later than the first one, which is feasible once we make the initialization smaller. Then, we would like to verify the dynamics of $\langle\epsilon_t\tilde{u}_t, u_1\rangle$ when $\epsilon_t$ is relatively large.

After keeping higher-order terms of $\epsilon_t$ in (172) and considering $u_1^\top\tilde{u}_t \approx 1$, we have

$$\langle\epsilon_{t+1}\tilde{u}_t, u_1\rangle = \langle\epsilon_t\tilde{u}_t, u_1\rangle - \eta\left(\sigma_1^2\epsilon_t + \sigma_1\epsilon_t^2 + \epsilon_t^3 + \sigma_1^2\epsilon_t + \sigma_1\epsilon_t^2 + \sigma_1\epsilon_t^2 + \epsilon_t u_1^\top\widetilde{\Delta\mathbf{X}_t}\widetilde{\mathbf{X}_0}^\top\tilde{u}_t\right) \tag{200}$$

$$= \langle\epsilon_t\tilde{u}_t, u_1\rangle - \eta\left(2\sigma_1^2\epsilon_t + 3\sigma_1\epsilon_t^2 + \epsilon_t^3 + \epsilon_t u_1^\top\widetilde{\Delta\mathbf{X}_t}\widetilde{\mathbf{X}_0}^\top\tilde{u}_t\right). \tag{201}$$

We are to show that $u_1^\top \widetilde{\Delta \mathbf{X}_t} \widetilde{\mathbf{X}_0}^\top \tilde{u}_t$ is small, so that $\langle \epsilon_{t+1} \tilde{u}_t, u_1 \rangle$ is a function approximately of only $\eta, \epsilon_t, \sigma_1$.

After re-introducing higher-order terms around (178), we have

$$\widetilde{\Delta \mathbf{X}_{t+1}} = \widetilde{\Delta \mathbf{X}_t} - \eta \left( \sigma_1 \epsilon_t u_1 \tilde{u}_1^\top \widetilde{\mathbf{X}_0} + \widetilde{\Delta \mathbf{X}_t} \widetilde{\mathbf{X}_0}^\top \widetilde{\mathbf{X}_0} + \widetilde{\mathbf{X}_0} \widetilde{\Delta \mathbf{X}_t}^\top \widetilde{\mathbf{X}_0} + \epsilon_t^2 \tilde{u}_t \tilde{u}_t^\top \widetilde{\mathbf{X}_0} \right) \tag{202}$$

$$= \widetilde{\Delta \mathbf{X}_t} - \eta \left( \sigma_1 \epsilon_t u_1 \tilde{u}_1^\top \widetilde{\mathbf{X}_0} + \widetilde{\Delta \mathbf{X}_t} \widetilde{\mathbf{X}_0}^\top \widetilde{\mathbf{X}_0} + \widetilde{\mathbf{X}_0} \widetilde{\Delta \mathbf{X}_t}^\top \widetilde{\mathbf{X}_0} + \left( m_t m_t^\top + \epsilon_t u_1 m_t^\top \right) \widetilde{\mathbf{X}_0} \right). \tag{203}$$

Then, it holds

$$u_1^\top \widetilde{\Delta \mathbf{X}_t} \widetilde{\mathbf{X}_0}^\top \tilde{u}_t = u_1^\top \widetilde{\Delta \mathbf{X}_{t-1}} \widetilde{\mathbf{X}_0}^\top m_t \tag{204}$$

$$= u_1^\top \widetilde{\Delta \mathbf{X}_{t-1}} \widetilde{\mathbf{X}_0}^\top \left( \mathbf{I} - \eta \widetilde{\mathbf{X}_0} \widetilde{\mathbf{X}_0}^\top \right) m_{t-1} - \eta (\sigma_1 + \epsilon_t) m_t^\top \widetilde{\mathbf{X}_0} \widetilde{\mathbf{X}_0}^\top m_t \tag{205}$$

$$\leq u_1^\top \widetilde{\Delta \mathbf{X}_{t-1}} \widetilde{\mathbf{X}_0}^\top m_{t-1} + \eta (\sigma_1 + \epsilon_t) m_t^\top \widetilde{\mathbf{X}_0} \widetilde{\mathbf{X}_0}^\top m_t. \tag{206}$$

So we have to ensure that $\|m_t\|$ keeps small. To obtain this, we start to re-write the update of $m_t$ by $m_t = \epsilon_t \tilde{u}_t - \langle \epsilon_t \tilde{u}_t, u_1 \rangle u_1$, which is

$$m_t = \left( 1 - \eta(\sigma^2 + \epsilon_t^2) \right) m_{t-1} - \eta \sigma_1 \widetilde{\mathbf{X}_0} \widetilde{\Delta \mathbf{X}_{t-1}}^\top u_1 \tag{207}$$

$$- \eta \widetilde{\mathbf{X}_0} \widetilde{\Delta \mathbf{X}_{t-1}}^\top m_{t-1} - \eta \left( \mathbf{I} - u_1 u_1^\top \right) \widetilde{\Delta \mathbf{X}_{t-1}} \widetilde{\mathbf{X}_0}^\top m_{t-1}. \tag{208}$$

Compared with (180), the above equation has two differences, the first is the coefficient of $m_{t-1}$ being $\left( 1 - \eta(\sigma^2 + \epsilon_t^2) \right)$ instead of $\left( 1 - \eta \sigma^2 \right)$, and the second is additional two terms in (208). Let's first discuss the second difference. Actually these two terms somehow act as the coefficient as well, so if $\left\| \widetilde{\mathbf{X}_0} \right\| \ll 1$ then it is safe to neglect them and only consider $\left( 1 - \eta \sigma^2 \right)$. To show this, we again look into the stability of $\left\| u^\top \widetilde{\Delta \mathbf{X}_{t+1}} \right\|$ and $\left\| \widetilde{\Delta \mathbf{X}_{t+1,\perp}} \right\|$. We have

$$u_1^\top \widetilde{\Delta \mathbf{X}_{t+1}} = u_1^\top \widetilde{\Delta \mathbf{X}_t} \left( \mathbf{I} - \eta \widetilde{\mathbf{X}_0}^\top \widetilde{\mathbf{X}_0} \right) - \eta (\sigma_1 + \epsilon_t) m_t^\top \widetilde{\mathbf{X}_0} \tag{209}$$

$$\widetilde{\Delta \mathbf{X}_{t+1,\perp}} = \widetilde{\Delta \mathbf{X}_{t,\perp}} - \eta \left( \widetilde{\Delta \mathbf{X}_{t,\perp}} \widetilde{\mathbf{X}_0}^\top \widetilde{\mathbf{X}_0} + \widetilde{\mathbf{X}_0} \widetilde{\Delta \mathbf{X}_{t,\perp}}^\top \widetilde{\mathbf{X}_0} \right), \tag{210}$$

where the first line simply differs from (191) with the coefficient as $\eta(\sigma_1 + \epsilon_t)$ instead of $\eta \sigma_1$ and the second is the same as (193). Therefore, following the same arguments in the phase I, if both of $\|m_{t-1}\|, \left\| \widetilde{\Delta \mathbf{X}_t} \right\| = O(\epsilon)$ are small, we have the following step they are still in $O(\epsilon)$.

Therefore, we have the dynamics of $\epsilon_t$ as

$$\langle \epsilon_{t+1} \tilde{u}_t, u_1 \rangle = \epsilon_{t+1} = \epsilon_t - \eta \left( 2\sigma_1^2 \epsilon_t + 3\sigma_1 \epsilon_t^2 + \epsilon_t^3 \right), \tag{211}$$

which corresponds to the update rule of gradient descent on a 1D function $f(\epsilon) = \frac{1}{4} \left( (\epsilon + \sigma_1)^2 - \sigma_1^2 \right)^2$ with learning rate $\eta$. Since $f''(0) = 2\sigma_1^2$, the learning rate $\eta = \frac{1}{\sigma_1^2} + \beta > \frac{2}{2\sigma_1^2}$ by a margin $\beta \sigma_1^2$. Since for 1D function $f(x) = \frac{1}{4}(x^2 - \mu)^2$ with learning rate $\eta$, it converges to the two positive solutions, $\epsilon$, in (39)

$$\eta = \frac{1}{x^2 \left( \sqrt{\frac{\mu}{x^2} - \frac{3}{4}} + \frac{1}{2} \right)}. \tag{212}$$

Therefore, for 1D function $f(\epsilon) = \frac{1}{4} \left( (\epsilon + \sigma_1)^2 - \sigma_1^2 \right)^2$ with learning rate $\eta$, it converges to the solutions (one in $(-\sigma_1, 0)$, one in $(0, \sigma_2)$) of

$$1 + \beta \sigma_1^2 = \frac{1}{\left( \frac{\epsilon}{\sigma_1} + 1 \right)^2 \left( \sqrt{\frac{1}{\left( \frac{\epsilon}{\sigma_1} + 1 \right)^2} - \frac{3}{4}} + \frac{1}{2} \right)}. \tag{213}$$

Along with the above argument of stability $\left\| \widetilde{\Delta \mathbf{X}_t} \right\| = O(\epsilon)$, it concludes that the trajectory converge to the above solution with deviation upper-bounded by $O(\epsilon)$. $\qquad \square$

### J.3 QUASI-SYMMETRIC CASE: WALK TOWARDS FLATTEST MINIMA

**Theorem 14** (Restatement of Theorem 4). *Consider the above quasi-symmetric matrix factorization with learning rate $\eta = \frac{1}{\sigma_1^2} + \beta$. Assume $0 < \beta\sigma_1^2 < \sqrt{4.5} - 1 \approx 1.121$. Consider a minimum $(\mathbf{Y}_0 = \alpha\mathbf{X}_0, \mathbf{Z}_0 = 1/\alpha\mathbf{X}_0), \alpha > 0$. The initialization is around the minimum, as $\mathbf{Y}_1 = \mathbf{Y}_0 + \Delta\mathbf{Y}_1, \mathbf{Z}_1 = \mathbf{Z}_0 + \Delta\mathbf{Z}_1$, with the deviations satisfying $u_1^\top \Delta\mathbf{Y}_1 v_1 \neq 0, u_1^\top \Delta\mathbf{Z}_1 v_1 \neq 0$ and $\|\Delta\mathbf{Y}_1\|, \|\Delta\mathbf{Z}_1\| \leq \epsilon$. The second largest singular value of $\mathbf{X}_0$ needs to satisfy*

$$\max\left\{\eta\frac{\sigma_1^2}{\alpha^2}\left(1 + \alpha^4\frac{\sigma_2^2}{\sigma_1^2}\right), \eta\sigma_1^2\alpha^2\left(1 + \frac{\sigma_2^2}{\alpha^4\sigma_1^2}\right)\right\} \leq 2. \tag{214}$$

*Then GD would converge to a period-2 orbit $\gamma_\eta$ approximately with error in $\mathcal{O}(\epsilon)$, formally written as*

$$(\mathbf{Y}_t, \mathbf{Z}_t) \to \gamma_\eta + (\Delta\mathbf{Y}, \Delta\mathbf{Z}), \qquad \|\Delta\mathbf{Y}\|, \|\Delta\mathbf{Z}\| = \mathcal{O}(\epsilon), \tag{215}$$

$$\gamma_\eta = \left\{\left(\mathbf{Y}_0 + (\rho_i - \alpha)\sigma_1 u_1 v_1^\top, \mathbf{Z}_0 + (\rho_i - 1/\alpha)\sigma_1 u_1 v_1^\top\right)\right\}, \qquad (i = 1, 2) \tag{216}$$

*where $\rho_1 \in (1, 2), \rho_2 \in (0, 1)$ are the two solutions of solving $\rho$ in*

$$1 + \beta\sigma_1^2 = \frac{1}{\rho^2\left(\sqrt{\frac{1}{\rho^2} - \frac{3}{4}} + \frac{1}{2}\right)}. \tag{217}$$

*Proof.* The update rule of gradient descent gives (for $t \geq 1$)

$$\mathbf{Y}_{t+1} = \mathbf{Y}_t - \eta\left(\mathbf{Y}_t\mathbf{Z}_t^\top - \mathbf{X}_0\mathbf{X}_0^\top\right)\mathbf{Z}_t, \tag{218}$$

$$\mathbf{Y}_{t+1} = \mathbf{Z}_t - \eta\left(\mathbf{Z}_t\mathbf{Y}_t^\top - \mathbf{X}_0\mathbf{X}_0^\top\right)\mathbf{Y}_t. \tag{219}$$

Denoting $\Delta\mathbf{Y}_t = \mathbf{Y}_t - \mathbf{Y}_0, \Delta\mathbf{Z}_t = \mathbf{Z}_t - \mathbf{Z}_0$, for $t \geq 1$, the update rule is equivalent to

$$\Delta\mathbf{Y}_{t+1} = \Delta\mathbf{Y}_t - \eta\left(\Delta\mathbf{Y}_t\mathbf{Z}_0^\top + \mathbf{Y}_0\Delta\mathbf{Z}_t^\top + \Delta\mathbf{Y}_t\Delta\mathbf{Z}_t^\top\right)(\mathbf{Z}_0 + \Delta\mathbf{Z}_t), \tag{220}$$

$$\Delta\mathbf{Z}_{t+1} = \Delta\mathbf{Z}_t - \eta\left(\Delta\mathbf{Z}_t\mathbf{Y}_0^\top + \mathbf{Z}_0\Delta\mathbf{Y}_t^\top + \Delta\mathbf{Z}_t\Delta\mathbf{Y}_t^\top\right)(\mathbf{Y}_0 + \Delta\mathbf{Y}_t). \tag{221}$$

Consider the decompositions $\Delta\mathbf{Y}_t = \epsilon_{y,t}\tilde{u}_{y,t}v_1^\top + \widetilde{\Delta\mathbf{Y}}_t, \Delta\mathbf{Z}_t = \epsilon_{z,t}\tilde{u}_{z,t}v_1^\top + \widetilde{\Delta\mathbf{Z}}_t$ where we assume $\widetilde{\Delta\mathbf{Y}}_t v_1 = \mathbf{0}, \widetilde{\Delta\mathbf{Z}}_t v_1 = \mathbf{0}, \|\tilde{u}_{y,t}\| = 1, \|\tilde{u}_{z,t}\| = 1$. We also control the sign of $\tilde{u}_{y,t}, \tilde{u}_{z,t}$ by claiming $\langle\tilde{u}_{y,t}, u_1\rangle > 0, \langle\tilde{u}_{z,t}, u_1\rangle > 0$. Then, the update rule is again equivalent to

$$\Delta\mathbf{Y}_{t+1} = \epsilon_{y,t}\tilde{u}_{y,t}v_1^\top + \widetilde{\Delta\mathbf{Y}}_t - \eta\left[\left(\epsilon_{y,t}\tilde{u}_{y,t}v_1^\top + \widetilde{\Delta\mathbf{Y}}_t\right)\mathbf{Z}_0^\top + \mathbf{Y}_0\left(\epsilon_{z,t}\tilde{u}_{z,t}v_1^\top + \widetilde{\Delta\mathbf{Z}}_t\right)^\top + \right.$$

$$\tag{222}$$

$$\left.\left(\epsilon_{y,t}\tilde{u}_{y,t}v_1^\top + \widetilde{\Delta\mathbf{Y}}_t\right)\left(\epsilon_{z,t}\tilde{u}_{z,t}v_1^\top + \widetilde{\Delta\mathbf{Z}}_t\right)^\top\right](\mathbf{Z}_0 + \epsilon_{z,t}\tilde{u}_{z,t}v_1^\top + \widetilde{\Delta\mathbf{Z}}_t),$$

$$\tag{223}$$

$$\Delta\mathbf{Z}_{t+1} = \epsilon_{z,t}\tilde{u}_{z,t}v_1^\top + \widetilde{\Delta\mathbf{Z}}_t - \eta\left[\left(\epsilon_{z,t}\tilde{u}_{z,t}v_1^\top + \widetilde{\Delta\mathbf{Z}}_t\right)\mathbf{Y}_0^\top + \mathbf{Z}_0\left(\epsilon_{y,t}\tilde{u}_{y,t}v_1^\top + \widetilde{\Delta\mathbf{Y}}_t\right)^\top + \right.$$

$$\tag{224}$$

$$\left.\left(\epsilon_{y,t}\tilde{u}_{y,t}v_1^\top + \widetilde{\Delta\mathbf{Y}}_t\right)\left(\epsilon_{z,t}\tilde{u}_{z,t}v_1^\top + \widetilde{\Delta\mathbf{Z}}_t\right)^\top\right](\mathbf{Y}_0 + \epsilon_{y,t}\tilde{u}_{y,t}v_1^\top + \widetilde{\Delta\mathbf{Y}}_t).$$

$$\tag{225}$$

**Phase I: $\tilde{u}_{y,t}$ and $\tilde{u}_{z,t}$ get close to $u_1$ sharply from random directions**

Decompose the three matrices at the around-initialization minimum $\mathbf{X}_0 = \sigma_1 u_1 v_1^\top + \widetilde{\mathbf{X}}_0, \mathbf{Y}_0 = \alpha\sigma_1 u_1 v_1^\top + \widehat{\mathbf{Y}}_0, \mathbf{Z}_0 = \frac{1}{\alpha}\sigma_1 u_1 v_1^\top + \widehat{\mathbf{Z}}_0$. Obviously they satisfy $\widehat{\mathbf{X}}_0 = \frac{1}{\alpha}\widehat{\mathbf{Y}}_0 = \alpha\widehat{\mathbf{Z}}_0$. For now, let's

assume $\left\|\widetilde{\Delta\mathbf{Y}_t}\right\|, \left\|\widetilde{\Delta\mathbf{Z}_t}\right\|$ stay small, then ignoring high-order small values gives

$$\epsilon_{y,t+1}\tilde{u}_{y,t+1} = \Delta\mathbf{Y}_{t+1}v_1 = \epsilon_{y,t}\tilde{u}_{y,t} - \eta\left(\epsilon_{y,t}\tilde{u}_{y,t}\frac{\sigma_1^2}{\alpha^2} + \frac{\sigma_1}{\alpha}\widetilde{\mathbf{Y}_0}\widetilde{\Delta\mathbf{Z}_t}^\top u_1 + \epsilon_{z,t}\sigma_1^2 u_1\tilde{u}_{z,t}^\top u_1\right), \tag{226}$$

$$\epsilon_{z,t+1}\tilde{u}_{z,t+1} = \Delta\mathbf{Z}_{t+1}v_1 = \epsilon_{z,t}\tilde{u}_{z,t} - \eta\left(\epsilon_{z,t}\tilde{u}_{z,t}\sigma_1^2\alpha^2 + \sigma_1\alpha\widetilde{\mathbf{Z}_0}\widetilde{\Delta\mathbf{Y}_t}^\top u_1 + \epsilon_{y,t}\sigma_1^2 u_1\tilde{u}_{y,t}^\top u_1\right), \tag{227}$$

then we have

$$P_{y,t+1} \triangleq \langle\epsilon_{y,t+1}\tilde{u}_{y,t+1}, u_1\rangle = \left(1 - \eta\frac{\sigma_1^2}{\alpha^2}\right)P_{y,t} - \eta\sigma_1^2 P_{z,t}, \tag{228}$$

$$P_{z,t+1} \triangleq \langle\epsilon_{z,t+1}\tilde{u}_{z,t+1}, u_1\rangle = \left(1 - \eta\frac{\sigma_1^2}{\alpha^2}\right)P_{z,t} - \eta\sigma_1^2 P_{y,t}. \tag{229}$$

From the above two, we would like to find a lower bound of $P_{y,t}, P_{z,t}$. Note that

$$P_{z,t+1} - \alpha^2 P_{y,t+1} = P_{z,t} - \alpha^2 P_{y,t} \triangleq k, \tag{230}$$

$$P_{y,t+1} = \left(1 - \eta\frac{\sigma_1^2}{\alpha^2}\right)P_{y,t} - \eta\sigma_1^2\left(\alpha^2 P_{y,t} + k\right) = \left(1 - \eta\sigma_1^2\alpha^2 - \eta\frac{\sigma_1^2}{\alpha^2}\right)P_{y,t} - \eta\sigma_1^2 \cdot k. \tag{231}$$

Since $1 - \eta\sigma_1^2\alpha^2 - \eta\frac{\sigma_1^2}{\alpha^2} \leq 1 - 2\eta\sigma_1^2 < -1$, we can see $|P_{y,t}|$ is growing exponentially with the ratio of $2\eta\sigma_1^2 - 1$ at least. Meanwhile, with $P_{z,t} - \alpha^2 P_{y,t}$ fixed along time, it holds $|P_{z,t}|$ is growing exponentially with the same ratio as well.

Now let's see how other things stay small when $P_{y,t}$ and $P_{z,t}$ are the only two terms growing exponentially. If that holds, we can conclude that $\tilde{u}_{y,t}$ and $\tilde{u}_{z,t}$ get close to $u_1$ sharply from random directions. Similar to the discussion of the symmetric case, we have two remaining components as follows (which are wished to be bounded)

$$m_{y,t+1} \triangleq \epsilon_{y,t+1}\tilde{u}_{y,t+1} - \langle\epsilon_{y,t+1}\tilde{u}_{y,t+1}, u_1\rangle u_1 = \left(1 - \eta\frac{\sigma_1^2}{\alpha^2}\right)m_{y,t} - \eta\frac{\sigma_1}{\alpha}\widetilde{\mathbf{Y}_0}\widetilde{\Delta\mathbf{Z}_t}^\top u_1 \tag{232}$$

$$m_{z,t+1} \triangleq \epsilon_{z,t+1}\tilde{u}_{z,t+1} - \langle\epsilon_{z,t+1}\tilde{u}_{z,t+1}, u_1\rangle u_1 = \left(1 - \eta\sigma_1^2\alpha^2\right)m_{z,t} - \eta\sigma_1\alpha\widetilde{\mathbf{Z}_0}\widetilde{\Delta\mathbf{Y}_t}^\top u_1 \tag{233}$$

$$u_1^\top\widetilde{\Delta\mathbf{Y}_{t+1}} = u_1^\top\widetilde{\Delta\mathbf{Y}_t} - \eta\left(u_1^\top\widetilde{\Delta\mathbf{Z}_t}\widetilde{\mathbf{Y}_0}^\top\widetilde{\mathbf{Y}_0} + \sigma_1 m_{y,t}^\top\widetilde{\mathbf{X}_0}\right) \tag{234}$$

$$u_1^\top\widetilde{\Delta\mathbf{Z}_{t+1}} = u_1^\top\widetilde{\Delta\mathbf{Z}_t} - \eta\left(u_1^\top\widetilde{\Delta\mathbf{Y}_t}\widetilde{\mathbf{Z}_0}^\top\widetilde{\mathbf{Z}_0} + \sigma_1 m_{z,t}^\top\widetilde{\mathbf{X}_0}\right) \tag{235}$$

Take any eigenvector $v_p$ of $\widetilde{\mathbf{X}_0}\widetilde{\mathbf{X}_0}^\top$ with associated eigenvalue $\sigma_p^2$ with $\sigma_p > 0$. Then the above system can be written as

$$\langle m_{y,t+1}, v_p\rangle = \left(1 - \eta\frac{\sigma_1^2}{\alpha^2}\right)\langle m_{y,t}, v_p\rangle - \eta\sigma_1\sigma_p\langle u_1^\top\widetilde{\Delta\mathbf{Z}_t}, v_p\rangle \tag{236}$$

$$\langle m_{z,t+1}, v_p\rangle = \left(1 - \eta\sigma_1^2\alpha^2\right)\langle m_{z,t}, v_p\rangle - \eta\sigma_1\sigma_p\langle u_1^\top\widetilde{\Delta\mathbf{Y}_t}, v_p\rangle \tag{237}$$

$$\langle u_1^\top\widetilde{\Delta\mathbf{Y}_{t+1}}, v_p\rangle = \langle u_1^\top\widetilde{\Delta\mathbf{Y}_t}, v_p\rangle - \eta\left(\sigma_p^2\alpha^2\langle u_1^\top\widetilde{\Delta\mathbf{Z}_t}, v_p\rangle + \sigma_1\sigma_p\langle m_{y,t}, v_p\rangle\right) \tag{238}$$

$$\langle u_1^\top\widetilde{\Delta\mathbf{Z}_{t+1}}, v_p\rangle = \langle u_1^\top\widetilde{\Delta\mathbf{Z}_t}, v_p\rangle - \eta\left(\frac{\sigma_p^2}{\alpha^2}\langle u_1^\top\widetilde{\Delta\mathbf{Y}_{t+1}}, v_p\rangle + \sigma_1\sigma_p\langle m_{z,t+1}, v_p\rangle\right) \tag{239}$$

Then the above system can be re-written as matrix $\mathbf{A} \in \mathbb{R}^{4\times4}$, which maps from $(\langle m_{y,t}, v_p\rangle, \langle m_{z,t}, v_p\rangle, \langle u_1^\top\widetilde{\Delta\mathbf{Y}_t}, v_p\rangle, \langle u_1^\top\widetilde{\Delta\mathbf{Z}_t}, v_p\rangle)$ to $(\langle m_{y,t+1}, v_p\rangle, \langle m_{z,t+1}, v_p\rangle, \langle u_1^\top\widetilde{\Delta\mathbf{Y}_{t+1}}, v_p\rangle, \langle u_1^\top\widetilde{\Delta\mathbf{Z}_{t+1}}, v_p\rangle)$. The explicit form of $\mathbf{A}$ is

$$\mathbf{A} = \begin{bmatrix} 1 - \eta\frac{\sigma_1^2}{\alpha^2} & 0 & 0 & -\eta\sigma_1\sigma_p \\ 0 & 1 - \eta\sigma_1^2\alpha^2 & -\eta\sigma_1\sigma_p & 0 \\ -\eta\sigma_1\sigma_p & 0 & 1 & -\eta\alpha^2\sigma_p^2 \\ 0 & -\eta\sigma_1\sigma_p & -\eta\frac{\sigma_p^2}{\alpha^2} & 1 \end{bmatrix} \tag{240}$$

$$= \mathbf{I} - \eta\frac{\sigma_1^2}{\alpha^2}\begin{bmatrix} 1 \\ 0 \\ \alpha^2\frac{\sigma_p}{\sigma_1} \\ 0 \end{bmatrix}\begin{bmatrix} 1 & 0 & 0 & \frac{\sigma_p}{\sigma_1}\alpha^2 \end{bmatrix} - \eta\sigma_1^2\alpha^2\begin{bmatrix} 0 \\ 1 \\ 0 \\ \frac{\sigma_p}{\sigma_1\alpha^2} \end{bmatrix}\begin{bmatrix} 0 & 1 & \frac{\sigma_p}{\sigma_1\alpha^2} & 0 \end{bmatrix} \tag{241}$$

$$\triangleq \mathbf{I} - \mathbf{B}. \tag{242}$$

So we have a rank-2 matrix $\mathbf{B}$ with all elements as non-negative. Our target is to show $\mathbf{A}^n$ is bounded with $n \to \infty$. Hence, we require the spectral norm $\|\mathbf{B}\| \leq 2$ by Lemma 8, which gives

$$\max\left\{\eta\frac{\sigma_1^2}{\alpha^2}\left(1 + \alpha^4\frac{\sigma_p^2}{\sigma_1^2}\right), \eta\sigma_1^2\alpha^2\left(1 + \frac{\sigma_p^2}{\alpha^4\sigma_1^2}\right)\right\} \leq 2. \tag{243}$$

Therefore, this gives an upper bound for $\sigma_2$, as

$$\max\left\{\eta\frac{\sigma_1^2}{\alpha^2}\left(1 + \alpha^4\frac{\sigma_2^2}{\sigma_1^2}\right), \eta\sigma_1^2\alpha^2\left(1 + \frac{\sigma_2^2}{\alpha^4\sigma_1^2}\right)\right\} \leq 2. \tag{244}$$

Hence, with the above discussion, we have shown $\|m_{y,t}\|, \|m_{z,t}\|, \left\|u_1^\top\widetilde{\Delta\mathbf{Y}_t}\right\|, \left\|u_1^\top\widetilde{\Delta\mathbf{Z}_t}\right\|$ stay small in this phase. Meanwhile, the residual components of $\widetilde{\Delta\mathbf{Y}_t}, \widetilde{\Delta\mathbf{Z}_t}$ that are orthogonal to $u_1$ on the left, denoted as $\widetilde{\Delta\mathbf{Y}_{t,\perp}}, \widetilde{\Delta\mathbf{Z}_{t,\perp}}$, iterate following

$$\widetilde{\Delta\mathbf{Y}_{t+1,\perp}} = \widetilde{\Delta\mathbf{Y}_{t,\perp}} - \eta\left(\widetilde{\Delta\mathbf{Z}_{t,\perp}}\widetilde{\mathbf{Y}_0}^\top\widetilde{\mathbf{Y}_0} + \widetilde{\mathbf{Z}_0}\widetilde{\Delta\mathbf{Y}_{t,\perp}}^\top\widetilde{\mathbf{Y}_0}\right), \tag{245}$$

$$\widetilde{\Delta\mathbf{Z}_{t+1,\perp}} = \widetilde{\Delta\mathbf{Z}_{t,\perp}} - \eta\left(\widetilde{\Delta\mathbf{Y}_{t,\perp}}\widetilde{\mathbf{Z}_0}^\top\widetilde{\mathbf{Z}_0} + \widetilde{\mathbf{Y}_0}\widetilde{\Delta\mathbf{Z}_{t,\perp}}^\top\widetilde{\mathbf{Z}_0}\right). \tag{246}$$

Then we have

$$\mathbf{M}_{t+1} \triangleq \widetilde{\Delta\mathbf{Y}_{t+1,\perp}}\widetilde{\mathbf{X}_0}^\top + \widetilde{\mathbf{X}_0}\widetilde{\Delta\mathbf{Y}_{t+1,\perp}}^\top + \alpha^2(\widetilde{\Delta\mathbf{Z}_{t+1,\perp}}\widetilde{\mathbf{X}_0}^\top + \widetilde{\mathbf{X}_0}\widetilde{\Delta\mathbf{Z}_{t+1,\perp}}^\top) \tag{247}$$

$$= \widetilde{\Delta\mathbf{Y}_{t,\perp}}\widetilde{\mathbf{X}_0}^\top + \widetilde{\mathbf{X}_0}\widetilde{\Delta\mathbf{Y}_{t,\perp}}^\top + \alpha^2\widetilde{\Delta\mathbf{Z}_{t,\perp}}\widetilde{\mathbf{X}_0}^\top + \alpha^2\widetilde{\mathbf{X}_0}\widetilde{\Delta\mathbf{Z}_{t,\perp}}^\top \tag{248}$$

$$- \eta\widetilde{\Delta\mathbf{Z}_{t,\perp}}\widetilde{\mathbf{Y}_0}^\top\widetilde{\mathbf{Y}_0}\widetilde{\mathbf{X}_0}^\top - \eta\widetilde{\mathbf{X}_0}\widetilde{\mathbf{Y}_0}^\top\widetilde{\mathbf{Y}_0}\widetilde{\Delta\mathbf{Z}_{t,\perp}}^\top - \eta\widetilde{\mathbf{X}_0}\widetilde{\Delta\mathbf{Y}_{t,\perp}}^\top\widetilde{\mathbf{X}_0}\widetilde{\mathbf{X}_0}^\top \tag{249}$$

$$- \eta\widetilde{\mathbf{X}_0}\widetilde{\mathbf{X}_0}^\top\widetilde{\Delta\mathbf{Y}_{t,\perp}}\widetilde{\mathbf{X}_0}^\top - \eta\alpha^2\widetilde{\Delta\mathbf{Y}_{t,\perp}}\widetilde{\mathbf{Z}_0}^\top\widetilde{\mathbf{Z}_0}\widetilde{\mathbf{X}_0}^\top - \eta\alpha^2\widetilde{\mathbf{X}_0}\widetilde{\mathbf{Z}_0}^\top\widetilde{\mathbf{Z}_0}\widetilde{\Delta\mathbf{Y}_{t,\perp}}^\top \tag{250}$$

$$- \eta\alpha^2\widetilde{\mathbf{X}_0}\widetilde{\Delta\mathbf{Z}_{t,\perp}}^\top\widetilde{\mathbf{X}_0}\widetilde{\mathbf{X}_0}^\top - \eta\alpha^2\widetilde{\mathbf{X}_0}\widetilde{\mathbf{X}_0}^\top\widetilde{\Delta\mathbf{Z}_{t,\perp}}\widetilde{\mathbf{X}_0}^\top \tag{251}$$

$$= \widetilde{\Delta\mathbf{Y}_{t,\perp}}\widetilde{\mathbf{X}_0}^\top + \widetilde{\mathbf{X}_0}\widetilde{\Delta\mathbf{Y}_{t,\perp}}^\top + \alpha^2\widetilde{\Delta\mathbf{Z}_{t,\perp}}\widetilde{\mathbf{X}_0}^\top + \alpha^2\widetilde{\mathbf{X}_0}\widetilde{\Delta\mathbf{Z}_{t,\perp}}^\top \tag{252}$$

$$- \eta\alpha^2\widetilde{\Delta\mathbf{Z}_{t,\perp}}\widetilde{\mathbf{X}_0}^\top\widetilde{\mathbf{X}_0}\widetilde{\mathbf{X}_0}^\top - \eta\alpha^2\widetilde{\mathbf{X}_0}\widetilde{\mathbf{X}_0}^\top\widetilde{\mathbf{X}_0}\widetilde{\Delta\mathbf{Z}_{t,\perp}}^\top - \eta\widetilde{\mathbf{X}_0}\widetilde{\Delta\mathbf{Y}_{t,\perp}}^\top\widetilde{\mathbf{X}_0}\widetilde{\mathbf{X}_0}^\top \tag{253}$$

$$- \eta\widetilde{\mathbf{X}_0}\widetilde{\mathbf{X}_0}^\top\widetilde{\Delta\mathbf{Y}_{t,\perp}}\widetilde{\mathbf{X}_0}^\top - \eta\widetilde{\Delta\mathbf{Y}_{t,\perp}}\widetilde{\mathbf{X}_0}^\top\widetilde{\mathbf{X}_0}\widetilde{\mathbf{X}_0}^\top - \eta\widetilde{\mathbf{X}_0}\widetilde{\mathbf{X}_0}^\top\widetilde{\mathbf{X}_0}\widetilde{\Delta\mathbf{Y}_{t,\perp}}^\top \tag{254}$$

$$- \eta\alpha^2\widetilde{\mathbf{X}_0}\widetilde{\Delta\mathbf{Z}_{t,\perp}}^\top\widetilde{\mathbf{X}_0}\widetilde{\mathbf{X}_0}^\top - \eta\alpha^2\widetilde{\mathbf{X}_0}\widetilde{\mathbf{X}_0}^\top\widetilde{\Delta\mathbf{Z}_{t,\perp}}\widetilde{\mathbf{X}_0}^\top \tag{255}$$

$$= \mathbf{M}_t\left(0.5\mathbf{I} - \eta\widetilde{\mathbf{X}_0}\widetilde{\mathbf{X}_0}^\top\right) + \left(0.5\mathbf{I} - \eta\widetilde{\mathbf{X}_0}\widetilde{\mathbf{X}_0}^\top\right)\mathbf{M}_t. \tag{256}$$

Then, by reorganizing terms, $\|\mathbf{M}_{t+1}\|$ can be well bounded as $\|\mathbf{M}_{t+1}\| \leq \|\mathbf{M}_t\|$.

Similarly, we need to the bound the norm of the following term

$$\mathbf{N}_{t+1} \triangleq -(\widetilde{\Delta\mathbf{Y}_{t+1,\perp}}\widetilde{\mathbf{X}_0}^\top + \widetilde{\mathbf{X}_0}\widetilde{\Delta\mathbf{Y}_{t+1,\perp}}^\top) + \alpha^2(\widetilde{\Delta\mathbf{Z}_{t+1,\perp}}\widetilde{\mathbf{X}_0}^\top + \widetilde{\mathbf{X}_0}\widetilde{\Delta\mathbf{Z}_{t+1,\perp}}^\top) \quad (257)$$

$$= -\widetilde{\Delta\mathbf{Y}_{t,\perp}}\widetilde{\mathbf{X}_0}^\top - \widetilde{\mathbf{X}_0}\widetilde{\Delta\mathbf{Y}_{t,\perp}}^\top + \alpha^2\widetilde{\Delta\mathbf{Z}_{t,\perp}}\widetilde{\mathbf{X}_0}^\top + \alpha^2\widetilde{\mathbf{X}_0}\widetilde{\Delta\mathbf{Z}_{t,\perp}}^\top \quad (258)$$

$$+ \eta\alpha^2\widetilde{\Delta\mathbf{Z}_{t,\perp}}\widetilde{\mathbf{X}_0}^\top\widetilde{\mathbf{X}_0}\widetilde{\mathbf{X}_0}^\top + \eta\alpha^2\widetilde{\mathbf{X}_0}\widetilde{\mathbf{X}_0}^\top\widetilde{\mathbf{X}_0}\widetilde{\Delta\mathbf{Z}_{t,\perp}}^\top + \eta\widetilde{\mathbf{X}_0}\widetilde{\Delta\mathbf{Y}_{t,\perp}}^\top\widetilde{\mathbf{X}_0}\widetilde{\mathbf{X}_0}^\top \quad (259)$$

$$+ \eta\widetilde{\mathbf{X}_0}\widetilde{\mathbf{X}_0}^\top\widetilde{\Delta\mathbf{Y}_{t,\perp}}\widetilde{\mathbf{X}_0}^\top - \eta\widetilde{\Delta\mathbf{Y}_{t,\perp}}\widetilde{\mathbf{X}_0}^\top\widetilde{\mathbf{X}_0}\widetilde{\mathbf{X}_0}^\top - \eta\widetilde{\mathbf{X}_0}\widetilde{\mathbf{X}_0}^\top\widetilde{\mathbf{X}_0}\widetilde{\Delta\mathbf{Y}_{t,\perp}}^\top \quad (260)$$

$$- \eta\alpha^2\widetilde{\mathbf{X}_0}\widetilde{\Delta\mathbf{Z}_{t,\perp}}^\top\widetilde{\mathbf{X}_0}\widetilde{\mathbf{X}_0}^\top - \eta\alpha^2\widetilde{\mathbf{X}_0}\widetilde{\mathbf{X}_0}^\top\widetilde{\Delta\mathbf{Z}_{t,\perp}}\widetilde{\mathbf{X}_0}^\top \quad (261)$$

$$= \alpha^2\widetilde{\Delta\mathbf{Z}_{t+1,\perp}}\widetilde{\mathbf{X}_0}^\top\left(\mathbf{I} - \eta\widetilde{\mathbf{X}_0}\widetilde{\mathbf{X}_0}^\top\right) + \left(-\eta\widetilde{\mathbf{X}_0}\widetilde{\mathbf{X}_0}^\top\right)\alpha^2\widetilde{\Delta\mathbf{Z}_{t+1,\perp}}\widetilde{\mathbf{X}_0}^\top \quad (262)$$

$$+ \alpha^2\widetilde{\mathbf{X}_0}\widetilde{\Delta\mathbf{Z}_{t+1,\perp}}^\top\left(-\eta\widetilde{\mathbf{X}_0}\widetilde{\mathbf{X}_0}^\top\right) + \left(\mathbf{I} - \eta\widetilde{\mathbf{X}_0}\widetilde{\mathbf{X}_0}^\top\right)\alpha^2\widetilde{\mathbf{X}_0}\widetilde{\Delta\mathbf{Z}_{t+1,\perp}}^\top \quad (263)$$

$$- \widetilde{\Delta\mathbf{Y}_{t+1,\perp}}\widetilde{\mathbf{X}_0}^\top\left(\mathbf{I} - \eta\widetilde{\mathbf{X}_0}\widetilde{\mathbf{X}_0}^\top\right) - \left(-\eta\widetilde{\mathbf{X}_0}\widetilde{\mathbf{X}_0}^\top\right)\widetilde{\Delta\mathbf{Y}_{t+1,\perp}}\widetilde{\mathbf{X}_0}^\top \quad (264)$$

$$- \widetilde{\mathbf{X}_0}\widetilde{\Delta\mathbf{Y}_{t+1,\perp}}^\top\left(-\eta\widetilde{\mathbf{X}_0}\widetilde{\mathbf{X}_0}^\top\right) + \left(\mathbf{I} - \eta\widetilde{\mathbf{X}_0}\widetilde{\mathbf{X}_0}^\top\right)\widetilde{\mathbf{X}_0}\widetilde{\Delta\mathbf{Y}_{t+1,\perp}}^\top. \quad (265)$$

Regarding the above equation, there are several key observations:

(a) $\mathbf{N}_t$ is symmetric.

(b) For any eigenvector $v$ of $\widetilde{\mathbf{X}_0}\widetilde{\mathbf{X}_0}^\top$, we have $v^\top\mathbf{N}_{t+1}v = v^\top\mathbf{N}_t v$.

(c) For any distinct eigenvectors $v_p, v_q$ of $\widetilde{\mathbf{X}_0}\widetilde{\mathbf{X}_0}^\top$, we have $v_p^\top\mathbf{N}_{t+1}v_q + v_q^\top\mathbf{N}_{t+1}v_p = v_p^\top\mathbf{N}_t v_q + v_q^\top\mathbf{N}_t v_p$.

Combining these three observations, we have $v^\top\mathbf{N}_{t+1}v = v^\top\mathbf{N}_t v$ for any $v$ with $v$ decomposed as a linear combination of the eigenvectors of $\widetilde{\mathbf{X}_0}\widetilde{\mathbf{X}_0}^\top$. Therefore, it is fair to say $\|\mathbf{N}_{t+1}\| = \|\mathbf{N}_t\|$.

So combining $\|\mathbf{M}_{t+1}\| \leq \|\mathbf{M}_t\|$ and $\|\mathbf{N}_{t+1}\| = \|\mathbf{N}_t\|$ tells that both $\|\widetilde{\Delta\mathbf{Y}_{t+1,\perp}}\widetilde{\mathbf{X}_0}^\top + \widetilde{\mathbf{X}_0}\widetilde{\Delta\mathbf{Y}_{t+1,\perp}}^\top\|$, $\|\widetilde{\Delta\mathbf{Z}_{t+1,\perp}}\widetilde{\mathbf{X}_0}^\top + \widetilde{\mathbf{X}_0}\widetilde{\Delta\mathbf{Z}_{t+1,\perp}}^\top\|$ stay small, which tells $\left\|\widetilde{\Delta\mathbf{Y}_{t,\perp}}\right\|, \left\|\widetilde{\Delta\mathbf{Z}_{t,\perp}}\right\|$ also stay small.

Therefore, we can conclude that both $\left\|\widetilde{\Delta\mathbf{Y}_t}\right\|, \left\|\widetilde{\Delta\mathbf{Z}_t}\right\|$ stay small.

After proving that all of $\|m_{y,t}\|, \|m_{z,t}\|, \left\|\widetilde{\Delta\mathbf{Y}_t}\right\|, \left\|\widetilde{\Delta\mathbf{Z}_t}\right\|$ stay small, from (174), the only terms growing fast are $\langle\epsilon_t\tilde{u}_{y,t}, u_1\rangle, \langle\epsilon_t\tilde{u}_{z,t}, u_1\rangle$ exponentially, which means the projections of $\tilde{u}_{y,t}, \tilde{u}_{z,t}$ onto $u_1$ is also growing sharply.

**Phase II: after $\tilde{u}_y, \tilde{u}_z \parallel u_1$ approximately**

Following the same spirit of everything in the symmetric case, we re-introduce higher-order terms, as

$$\Delta\mathbf{Y}_{t+1}v_1 \triangleq \epsilon_{y,t+1}\tilde{u}_{y,t} = \epsilon_{y,t}\tilde{u}_{y,t} - \eta\bigg[\epsilon_{y,t}\tilde{u}_{y,t}\frac{\sigma_1^2}{\alpha^2} + \epsilon_{y,t}\epsilon_{z,t}\tilde{u}_{y,t}\frac{\sigma_1}{\alpha} \quad (266)$$

$$+ \sigma_1^2\epsilon_{z,t}u_1 + \alpha\sigma_1\epsilon_{z,t}^2 u_1 + \frac{\sigma}{\alpha}\epsilon_{y,t}\epsilon_{z,t}\tilde{u}_{y,t} + \epsilon_{y,t}\epsilon_{z,t}^2\tilde{u}_{y,t}.\bigg], \quad (267)$$

$$\epsilon_{y,t+1} = \langle\epsilon_{y,t+1}\tilde{u}_{y,t+1}, u_1\rangle = \langle\epsilon_{y,t}\tilde{u}_{y,t}, u_1\rangle - \eta\bigg[\frac{\sigma_1^2}{\alpha^2}\epsilon_{y,t} + 2\frac{\sigma_1}{\alpha}\epsilon_{y,t}\epsilon_{z,t} + \sigma_1^2\epsilon_{z,t} + \sigma_1\alpha\epsilon_{z,t}^2 + \epsilon_{y,t}\epsilon_{z,t}^2\bigg],$$
$$\quad (268)$$

$$\epsilon_{z,t+1} = \langle\epsilon_{z,t+1}\tilde{u}_{z,t+1}, u_1\rangle = \langle\epsilon_{z,t}\tilde{u}_{z,t}, u_1\rangle - \eta\bigg[\sigma_1^2\alpha^2\epsilon_{z,t} + 2\sigma_1\alpha\epsilon_{y,t}\epsilon_{z,t} + \sigma_1^2\epsilon_{y,t} + \frac{\sigma_1}{\alpha}\epsilon_{y,t}^2 + \epsilon_{y,t}^2\epsilon_{z,t}\bigg].$$
$$\quad (269)$$

Then we have

$$\sigma_1\alpha + \epsilon_{y,t+1} = \sigma_1\alpha + \epsilon_{y,t} - \eta\left(\frac{\sigma_1}{\alpha} + \epsilon_{z,t}\right)\left(\frac{\sigma_1}{\alpha}\epsilon_{y,t} + \epsilon_{y,t}\epsilon_{z,t} + \sigma_1\alpha\epsilon_{z,t}\right) \tag{270}$$

$$\frac{\sigma_1}{\alpha} + \epsilon_{z,t+1} = \frac{\sigma_1}{\alpha} + \epsilon_{z,t} - \eta\left(\sigma_1\alpha + \epsilon_{y,t}\right)\left(\frac{\sigma_1}{\alpha}\epsilon_{y,t} + \epsilon_{y,t}\epsilon_{z,t} + \sigma_1\alpha\epsilon_{z,t}\right). \tag{271}$$

Note that the shared factor $\left(\frac{\sigma_1}{\alpha}\epsilon_{y,t} + \epsilon_{y,t}\epsilon_{z,t} + \sigma_1\alpha\epsilon_{z,t}\right) = \left(\sigma_1\alpha + \epsilon_{y,t}\right)\left(\frac{\sigma_1}{\alpha} + \epsilon_{z,t}\right) - \sigma_1^2$. Hence, the above system is equivalent to

$$y = y - \eta(yz - \sigma_1^2)z, \tag{272}$$

$$z = z - \eta(yz - \sigma^2)y. \tag{273}$$

Furthermore, this is equivalent to $f(y,z) = \frac{1}{2}(yz-1)^2$ with learning rate $\eta' = 1 + \beta\sigma_1^2$. Therefore, from Theorem 5, we know $y, z$ will converge to the same values, which make the problem with the same solution of 1D functions. Fortunately, it follows the solution for (39). In summary, after straightforward computation, it converges to

$$(\mathbf{Y}_t, \mathbf{Z}_t) \to \gamma_\eta + (\Delta\mathbf{Y}, \Delta\mathbf{Z}), \qquad \|\Delta\mathbf{Y}\|, \|\Delta\mathbf{Z}\| = O(\epsilon), \tag{274}$$

$$\gamma_\eta = \left\{\left(\mathbf{Y}_0 + (\rho_i - \alpha)\,\sigma_1 u_1 v_1^\top, \mathbf{Z}_0 + (\rho_i - {}^1\!/\!\alpha)\,\sigma_1 u_1 v_1^\top\right)\right\}, \qquad (i = 1, 2) \tag{275}$$

where $\rho_1 \in (1, 2), \rho_2 \in (0, 1)$ are the two solutions of solving $\rho$ in

$$1 + \beta\sigma_1^2 = \frac{1}{\rho^2\left(\sqrt{\frac{1}{\rho^2} - \frac{3}{4}} + \frac{1}{2}\right)}. \tag{276}$$

Note that $\|\Delta\mathbf{Y}\|, \|\Delta\mathbf{Z}\| = \mathcal{O}(\epsilon)$ can achieved by the already presented discussion in Phase I, and here we omit similar discussion for Phase II, very close to the symmetric case. $\square$

## K  USEFUL LEMMAS

**Lemma 7.** *Assume $a \cdot \Delta a \geq b \cdot \Delta b$ and $a \geq b$. All of $a, b, \Delta a, \Delta b$ are positive. If $\Delta b \leq a$, then $a + \Delta a \geq b + \Delta b$.*

*Proof.* $(a + \Delta a) - (b + \Delta b) \geq a + b\frac{\Delta b}{a} - b - \Delta b = (\frac{\Delta b}{a} - 1)(b - a) \geq 0.$ $\square$

**Lemma 8.** *In (242), if we have $\|\mathbf{B}\| \leq 2$ and with $\sigma_2 < \sigma_1$, then $\mathbf{A}^t$ is bounded entry-wise, for any $t \geq 1$.*

*Proof.* Denote $u_1 = \frac{1}{z_1}\left[1, 0, \alpha^2\frac{\sigma_p}{\sigma_1}, 0\right]^\top, u_2 = \frac{1}{z_2}\left[0, 1, 0, \frac{\sigma_p}{\sigma_1\alpha^2}\right]^\top, v_1 = \frac{1}{z_3}\left[1, 0, 0, \frac{\sigma_p}{\sigma_1}\alpha^2\right]^\top,$
$v_2 = \frac{1}{z_4}\left[0, 1, \frac{\sigma_p}{\sigma_1\alpha^2}, 0\right]^\top$, where $z_1, z_2, z_3, z_4$ are positive normalization terms to ensure $\|u_i\| = 1, \|v_i\| = 1$ for $i = 1, 2$. Then $\mathbf{A}$ can be re-written as

$$\mathbf{A} = \mathbf{I} - \eta\frac{\sigma_1^2}{\alpha^2}\left(1 + \left(\alpha^2\frac{\sigma_p}{\sigma_1}\right)^2\right)u_1 v_1^\top - \eta\sigma_1^2\alpha^2\left(1 + \left(\frac{\sigma_p}{\sigma_1\alpha^2}\right)^2\right)u_2 v_2^\top. \tag{277}$$

Let's denote $a_1 = \eta\frac{\sigma_1^2}{\alpha^2}\left(1 + \left(\alpha^2\frac{\sigma_p}{\sigma_1}\right)^2\right), a_2 = \eta\sigma_1^2\alpha^2\left(1 + \left(\frac{\sigma_p}{\sigma_1\alpha^2}\right)^2\right)$. Obviously, for any $t \geq 1$, the $t$-th power of $\mathbf{A}$, can be represented as

$$\mathbf{A}^t = \mathbf{I} + k_1^{(t)}u_1 v_1^\top + k_2^{(t)}u_2 v_2^\top + k_3^{(t)}u_1 v_2^\top + k_4^{(t)}u_2 v_1^\top, \tag{278}$$

and the update rule of $k_i^{(t)}$'s is

$$k_1^{(t+1)} = k_1^{(t)}(1 - a_1 v_1^\top u_1) - k_3^{(t)}a_1 v_2^\top u_1 - a_1,$$
$$k_2^{(t+1)} = k_2^{(t)}(1 - a_2 v_2^\top u_2) - k_4^{(t)}a_2 v_1^\top u_2 - a_2,$$
$$k_3^{(t+1)} = k_3^{(t)}(1 - a_2 v_2^\top u_2) - k_1^{(t)}a_2 v_1^\top u_2,$$
$$k_4^{(t+1)} = k_4^{(t)}(1 - a_1 v_1^\top u_1) - k_2^{(t)}a_1 v_2^\top u_1.$$

Since $\mathbf{A}^t$ is bounded $\forall\, t$ if and only if $k_i^{(t)}$ is bounded for any $t \geq 1$, we only need to show the below matrix $\mathbf{C}$ with $\|\mathbf{C}\|_{\text{spectral}} < 1$, with $\mathbf{C}$ defined as

$$\mathbf{C} = \begin{bmatrix} 1 - a_1 v_1^\top u_1 & 0 & -a_1 v_2^\top u_1 & 0 \\ 0 & 1 - a_2 v_2^\top u_2 & 0 & -a_2 v_1^\top u_2 \\ -a_2 v_1^\top u_2 & 0 & 1 - a_2 v_2^\top u_2 & 0 \\ 0 & -a_1 v_2^\top u_1 & 0 & 1 - a_1 v_1^\top u_1 \end{bmatrix}, \tag{279}$$

where

$$v_1^\top u_1 = \frac{1}{\sqrt{1 + \left(\frac{\sigma_p}{\sigma_1}\alpha^2\right)^2}\sqrt{1 + \left(\frac{\sigma_p}{\sigma_1}\alpha^2\right)^2}}, \quad v_2^\top u_1 = \frac{\left(\frac{\sigma_p}{\sigma_1}\right)^2}{\sqrt{1 + \left(\frac{\sigma_p}{\sigma_1}\alpha^2\right)^2}\sqrt{1 + \left(\frac{\sigma_p}{\sigma_1\alpha^2}\right)^2}}, \tag{280}$$

$$v_2^\top u_2 = \frac{1}{\sqrt{1 + \left(\frac{\sigma_p}{\sigma_1\alpha^2}\right)^2}\sqrt{1 + \left(\frac{\sigma_p}{\sigma_1\alpha^2}\right)^2}}, \quad v_1^\top u_2 = \frac{\left(\frac{\sigma_p}{\sigma_1}\right)^2}{\sqrt{1 + \left(\frac{\sigma_p}{\sigma_1}\alpha^2\right)^2}\sqrt{1 + \left(\frac{\sigma_p}{\sigma_1\alpha^2}\right)^2}}. \tag{281}$$

To obtain $\|\mathbf{C}\|_2 < 1$, we only need to compute $\|\mathbf{C}_{i,:}\| < 1$ for $i \in [4]$. Taking $i = 1$ as an example, we have

$$\|\mathbf{C}_{1,:}\|^2 = (1 - a_1 v_1^\top u_1)^2 + (a_2 v_2^\top u_1)^2 = \left(1 - \eta\frac{\sigma_1^2}{\alpha^2}\right)^2 + \left(\eta\frac{\sigma_p^2}{\alpha^2}\frac{\sqrt{1 + \left(\frac{\sigma_p}{\sigma_1}\alpha^2\right)^2}}{\sqrt{1 + \left(\frac{\sigma_p}{\sigma_1\alpha^2}\right)^2}}\right)^2. \tag{282}$$

If $\alpha < 1$, the above RHS $< (1 - \eta\frac{\sigma_1^2}{\alpha^2})^2 + (\eta\frac{\sigma_p^2}{\alpha^2})^2 < 1$ where the second inequality is due to $\sigma_p^2 \leq \sigma_2^2 < \sigma_1^2$. If $\alpha \geq 1$, the condition of $a_1 \leq 2$ (from $\|\mathbf{B}\| \leq 2$) gives $\sigma_p^2 \leq \frac{2/\eta - \sigma_1^2/\alpha^2}{\alpha^2}$, which is further $\sigma_p^4 < \frac{2/\eta - \sigma_1^2/\alpha^2}{\alpha^2}\sigma_1^2$ due to $\sigma_p^2 < \sigma_1^2$. As a result, it holds $\eta\sigma_1^4 - 2\alpha^2\sigma_1^2 + \eta\sigma_p^4\alpha^4 < 0$, which means $(1 - \eta\sigma_1^2/\alpha^2)^2 + (\eta\sigma_p^2)^2 < 1$. Noting the above RHS$\leq (1 - \eta\sigma_1^2/\alpha^2)^2 + (\eta\sigma_p^2/\alpha^2 \cdot \alpha^2)^2 = (1 - \eta\sigma_1^2/\alpha^2)^2 + (\eta\sigma_p^2)^2$ when $\alpha \geq 1$, it finishes the proof. $\qquad\square$

## L    ILLUSTRATION OF PERIOD-2 AND PERIOD-4 ORBITS

In the proof of local convergence to the period-2 orbit in (**??**), we give a bound of learning rate as $\sqrt{5} - 1 \approx 1.236$. Local convergence is guaranteed if the learning rate is smaller than it. Conversely, if the learning rate is larger than it, although the period-2 orbit still exists, GD starting from a point infinitesimally close to the orbit still escapes from it. This is when GD converges to a higher-order orbit.

Figure 9 precisely shows the effectiveness of such a bound where GD converges to the period-2 orbit when $\eta = 1.235 < \sqrt{5} - 1$ and a period-4 orbit when $\eta = 1.237 > \sqrt{5} - 1$.

## M    DISCUSSIONS

First, we provide a general roadmap of our theoretical results in Section M.1, as illustrated in Figure 10. Then, in Section M.2 we discuss three implications from our current low-dimensional settings to more complicated models for future understanding of EoS in pratical NNs, where low-dimension theorems are enhanced with high-dim experiments.

### M.1    CONNECTIONS BETWEEN THEORETICAL RESULTS

In this section, we discuss the connections between our presented theoretical results, as illustrated in Figure 10.

Theorem 1 and Lemma 1 present (local) intrinsic geometric properties for a 1-D function to allow stable oscillations. Such properties provide us the 1-D function $f(x) = (\mu - x^2)^2$ and, furthermore,

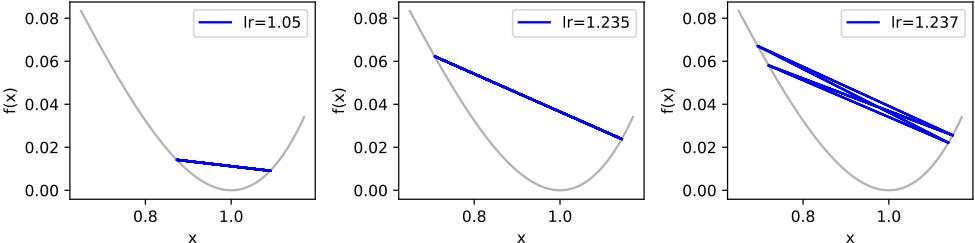

Figure 9: The convergent orbits of GD on $f(x) = \frac{1}{4}(x^2 - 1)^2$ with learning rate=1.05, 1.235 and 1.237. The first two smaller learning rates drive to period-2 orbits while the last one goes to an period-4 orbit. The significant bound between period-2 and period-4 is predicted by our proof in (**??**), as $\sqrt{5} - 1 \approx 1.236$.

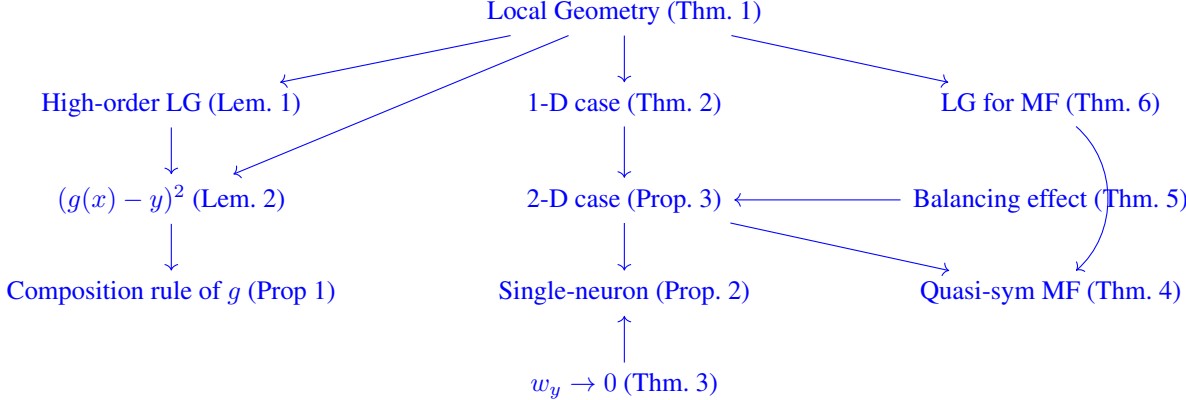

Figure 10: Connections between our presented theoretical results. The arrows stand for "implies". LG stands for Local Geometry. MF stands for Matrix Factorization.

we generalize the local property to a global convergence result in Theorem 2. Then we are to generalize the 1-D analysis to cases of i) multi-parameter, ii) nonlinear and iii) high-dimension.

(a) **Multi-parameter**. Compared with 1-D $f(x) = (\mu - x^2)^2$, the 2-D function $f(x, y) = (\mu - xy)^2$ can be viewed as the simplest setting of two-layer models. We prove that the 2-D case converges to the region of $x = y$ in Theorem 5 in Appendix A.1, which means it shares the same convergence as the 1-D model. Also, $x = y$ means its sharpness is the flattest.

(b) **Nonlinear**. We extend the 2-D model to a two-layer single-neuron ReLU model in Section 5. Although the student neuron can be initialized far from the direction of the teacher neuron, we prove the student neuron converges to the correct direction (as $w_y \to 0$) in Theorem 3. Then the problem degenerates to the above 2-D analysis, which means it shares the same convergence with the 2-D, where $(v, w_x)$ corresponds to $(x, y)$ in 2-D.

(c) **High-dimension**. We extend the 2-D model to quasi-symmetric matrix factorization in Section 6. Although the parameters are initialized near a sharp minima, GD still walks towards the flattest minima, as shown in Theorem 4. At convergence, the top singular values of $\mathbf{Y}, \mathbf{Z}$ are the same, following the 2-D analysis. So the singular values are in the same period-2 orbit as the 1-D case.

Meanwhile, from Theorem 1 and Lemma 1, we prove a condition for base models $g$ in regression tasks to allow stable oscillation in Lemma 2. Furthermore, we provide a composition rule of two base models to find a more complicated model that allows stable oscillation in Prop 1.

### M.2    IMPLICATIONS FROM LOW-DIMENSION TO HIGH-DIMENSION

We would like to emphasize that, although our current simple settings are a little far from practical NNs, it still helps understand the ability of GD at large LRs to discover flat minima in three steps as follows. We include more experiments in Appendix B.2 to present the following hopes for complicated networks:

(a) By Theorem 1, especially its second condition, we wish to discover an intrinsic geometric property around local minima of more complicated models. The key is to investigate the 1-D function at the cross-section of the leading eigenvector and the loss landscape.

◆ Theoretical: we prove the 1-D condition holds at any minima for non-trivial matrix factorization, shown as Theorem 6 in Appendix A.2.

✽ Empirical: we show the 1-D condition holds around minima of 3,4,5-layer ReLU MLPs on MNIST, shown in Figure 6(d), 7(d), 8(d) in Appendix B.2.2.

(b) With the above intrinsic geometric property, the next question is whether the training trajectory utilizes this property.

◆ Theoretical: in the case of quasi-symmetric matrix factorization, we prove that the training trajectory follows the leading eigenvector of the Hessian (i.e. the leading component of $\mathbf{X}_0$) in Theorem 4, where the only top components of weights are changing in $\omega(\epsilon)$.

✽ Empirical: for MLPs on MNIST, we show the almost perfect alignment of the gradient and the top Hessian eigenvector in Figure 6(c), 7(c), 8(c).

(c) The final implication is the implicit bias of EoS after such oscillation. It turns out GD is driven to flatter minima from sharper minima. In the 1-D case, obviously there is nothing about implicit bias since the only thing GD is doing is to approximate the target value. However, an implicit bias from the oscillation appears starting from the 2-D case.

◆ Theoretical 1: in the 2-D case in Theorem 5, we prove the two learnable parameters $x, y$ will converge to the same values after oscillations of their product $xy$. Actually in the minimum manifold, smaller $|x - y|$ means a flatter minimizer.

◆ Theoretical 2: in the single-neuron ReLU network in Theorem 3 and Prop 2, we show the model degenerates to the 2-D case since $w_y \to 0$. The 2-D argument tells that this nonlinear model also walks towards the balanced situation, verified with experiments in Figure 2.

◆ Theoretical 3: in the quasi-symmetric MF in Theorem 4, although the initialization is around a sharp minima, GD is still driven towards the flattest minima where $\sigma_{\max}(\mathbf{Y}) = \sigma_{\max}(\mathbf{Z})$.

✽ Empirical 1: for 2-layer 16-neuron ReLU network in a student-teacher setting, it turns out learning rate decay after beyond-EoS oscillations drives the model very close to the flattest minima, as shown in Figure 5 and in Appendix B.2.1.

✽ Empirical 2: for 3,4,5-layer MLPs on MNIST, larger learning rate drives to a flatter minima, as shown in Figure 6(b).

## N    CONCLUSIONS

In this work, we investigate gradient descent with a large step size that crosses the threshold of local stability. In the low dimensional setting, we provide conditions on high-order derivatives that allow stable oscillation around local minima. For a two-layer single-neuron ReLU network, we prove its convergence to align with the teacher neuron under population loss. For matrix factorization, we prove that the necessary 1-D condition holds around any minima. Furthermore, we conduct an analysis of GD in symmetric matrix factorization, which converges to a period-2 orbit aligned with the 1-D convergence. Moreover, we generalize the analysis to quasi-symmetric cases where GD walks towards the flattest minimiser although initialized near sharp ones. A further discussion is provided in Appendix M.

While these are encouraging results that contribute to the growing understanding of gradient descent beyond the Edge of Stability, our analysis suffers from important limitations that require further work.

An important item for future work is therefore to extend it to general dimensions with nonlinearity, which will enable the analysis of empirical landscapes as well as multiple neurons. Next, the understanding of the implicit bias of GD in the large-learning rate regime won't be complete without integrating the noise, either in the classic SGD sense or in the labels, as done in Damian et al. (2021); Li et al. (2021).

