# OpenReview forum: "On Gradient Descent Convergence beyond the Edge of Stability"
_ICLR.cc/2023/Conference — Submitted to ICLR 2023_

### Official Review · Reviewer_zoGM · 2022-10-22

**Confidence:** 3
**Correctness:** 2
**Technical Novelty And Significance:** 2
**Empirical Novelty And Significance:** 2
**Recommendation:** 6

**Clarity, Quality, Novelty And Reproducibility:**

The clarity and novelty are major concerns of the paper: there are no clear takeaways, and some theoretical statements lack rigor.

**Strength And Weaknesses:**

Strength:

- The authors provided many intersting observations on specific examples.

Weaknesses:

- A major issue is that the work seems like collecting a number of scattered examples where GD can converge with step size larger than $\frac{1}{L}$, without any coherent observation.

- The results on the 1-diemsnional functions are qualitatively different from the high-dimensional ones: In Section 4 (barring the special case of Theorem 2), the statements are "There **exists** a large step size such that XXX happens", whereas for higher-dimensional ones, they change to "**For all** step size in this region, XXX happens." Moreover, the existence proofs in Section 4 produce step sizes that are of Lebesgue measure 0, which shows that it is almost impossible for any practical algorithm to satisfy such conditions. In this sense, contrary to what the authors claimed, there is a disconnection between the low-dimensional and high-dimensional analysis.

- All the proofs, especially that of Theorem 3, rely on tedious and highly-specific calculations that do not reveal much about the general behavior of GD.

- The presentation of Section 5 is quite opaque. Can the authors explain what is meant by *"$\tilde{w}_\perp$ as the unit-length orthogonal residual of $w$ after projecting onto $\tilde{w}$"*? I understood it as the normalized version of $w- \text{proj}_\tilde{w} w$, but then this would violate the followup statement *"it is clear that updates of $w$ always stay in the plane spanned by $\tilde{w}$ and $w^{(0)}$"* since $\tilde{w}_\perp$ is not in the span of these two vectors. In addition, unless I missed something, the meaning of Theorem 3 is not clearly stated: it states that the orthogonal component of $w_t$ will decrease to 0, but it does not prove convergence (i.e., $vw\rightarrow \tilde{w}$). Can the authors provide a more clear interpretation than *"... the dynamics of the single-neuron ReLU network is getting closer to the 2-D case in Section A.1"*?

- Theorem 5 in Appendix A is not correct. The authors seem to have assumed that $xy > \mu$ always happens in the GD process, which is not guaranteed. If the authors wish to investigate the case of only those iterates satisfying $x^{(t)}y^{(t)} > \mu$, then one cannot assume an GD update from $x^{(t)}y^{(t)}$ to $x^{(t+1)}y^{(t+1)}, which the authors implicitly did.

**Summary Of The Paper:**

This work investigates gradient descent with a step size larger than the $\frac{2}{L}$ predicted by the standard theory, a regime known as the edge of stability. The authors provided sufficient conditions for ensuring an oscillatory (instead of diverging) behavior around the local minima for 1-dimensional functions. For higher-dimensional problems, the authors considered two scenarios: a two-layer single-neuron ReLU network in a teacher-student setup, and quasi-symmetric matrix factorization. The main results are that these objectives can still converge or stabilize for some step size $\eta = \frac{2K}{L}$, $K>1$.



**Summary Of The Review:**

This work studies the EoS phenomenon and provides several interesting examples. However, the lack of a global picture, clarity of the presentation, and mathematical rigor makes me feel that a major revision is required before accepting it.

---

> ### Author Response · Authors · 2022-11-09
> **Authors' response (1/2)**
>
> Thank you for your time and careful review!
>
> The following discussions on i) implications from our low-dim settings to more complicated models, and ii) a roadmap of our theoretical results are added in Appendix M and Figure 10.
>
> ---
> ---
>
> **Q1**. Missing coherent observations as a collection of scattered examples
>
> **A1**. The main phenomena we are trying to understand is the ability of GD at large LRs to discover flat minima. This includes two necessary conditions: (i) oscillations need to stabilize, and (ii) trend towards flatter minimisers. As it turns out, (i) is captured by the behavior of the function along a certain one-dimensional projection/cross-section, and this is our starting point in Theorem 1 and lemma 1, revealing the relationship of several-order derivatives. The second property (ii) requires 'overparameterization', and we formally study it in matrix factorization, with the scalar factorization case being a particular case.
>
>
> Then, how can they help understand EoS in NNs starting from low-dim results? We would like to emphasize that they still help in the following three steps. We include more experiments in Appendix B.2 to present the following hopes for complicated networks:
>
> 1. By Theorem 1, especially its second condition, we discover an intrinsic geometric property around local minima of more complicated models. The key is to investigate the 1-D function at the cross-section of the leading eigenvector and the loss landscape.
>     * Theoretical: we prove the 1-D condition holds at any minima for non-trivial matrix factorization, shown as Theorem 6 in Appendix A.2.
>     * Empirical: we show the 1-D condition holds around minima of 3,4,5-layer ReLU MLP on MNIST, shown in Figure 6(d), 7(d), 8(d) in Appendix B.2.2.
> 2. With the above intrinsic geometric property, the next question is whether the training trajectory utilizes this property.
>     * Theoretical: in the case of quasi-symmetric matrix factorization, we prove that the training trajectory follows the leading eigenvector of the Hessian (i.e. the leading component of $\mathbf{X}_0$) in Theorem 4, where the only top components of weights are changing in $\omega(\epsilon)$.
>     * Empirical: for MLPs on MNIST, we show the almost perfect alignment of the gradient and the top Hessian eigenvector in Figure 6(c), 7(c), 8(c).
>
> 3. The final implication is the implicit bias of EoS after such oscillation. It turns out GD is driven to flatter minima from sharper minima. In the 1-D case, obviously there is nothing about implicit bias since the only thing GD is doing is to approximate the target value. However, an implicit bias from the oscillation appears starting from the 2-D case.
>     * Theoretical 1: in the 2-D case in Theorem 5, we prove the two learnable parameters $x,y$ will converge to the same values after oscillations of their product $xy$. Actually in the minimizer manifold, smaller $|x-y|$ means a flatter minimizer.
>     * Theoretical 2: in the single-neuron ReLU network in Theorem 3 and Prop 2, we show the model degenerates to the 2-D case since $w_y\rightarrow 0$. The 2-D argument tells that this nonlinear model also walks towards the balanced situation, verified with experiments in Figure 2.
>     * Theoretical 3: in the quasi-symmetric MF, although the initialization is around a sharp minima, GD is still driven towards the flattest minima where $\sigma_{\max}(Y)=\sigma_{\max}(Z)$.
>     * Empirical 1: for 2-layer 16-neuron ReLU network in a student-teacher setting, it turns out learning rate decay after beyond-EoS oscillations drives the model very close to the flattest minima, as shown in Figure 5 and in Appendix B.2.1.
>     * Empirical 2: for 3,4,5-layer MLPs on MNIST, larger learning rate drives to a flatter minima, as shown in Figure 6(b).
>
> ---
>
>
> **Q2**. step sizes of Lebesgue measure 0 in low dim
>
> **A2**. We have reformulated Theorem 1 and Lemma 1 to clarify our meaning. We agree that these two results include learning rates of Lebesgue measure 0. Nevertheless, they provide conditions of local geometry at minima to allow stable oscillations, which stand as foundations for our following analysis and all these results are strictly beyond EoS. It would be impossible to discover the function of interest in Theorem 2 without Theorem 1.
>
> Meanwhile, we develop Lemma 2 and Prop 1 based on Theorem 1 to show the richness of such a family of functions. We find an extension that the regression problem naturally allows stable oscillations with a list of popular base models in Corollary 1-8. This opens a direction for the future, which is to conduct global analysis of such functions.

---

> > ### Author Response · Authors · 2022-11-09
> > **Authors' response (2/2)**
> >
> > **Q3**. Tedious proof
> >
> > **A3**. We agree that the proof seems tedious. But it is necessary to provide a convergence result with such a large learning rate due to the following reasons:
> >
> > 1. In such a large learning rate, one of the most powerful tools in classic optimization – descent lemma – fails. Actually in our setting, it **never** happens $\eta\le 2/L$ where $L$ is the sharpness.
> > 2. Moreover, in such an extreme setting, we would like to prove something uniform for a region instead of some special points of measure zero. For example, in Theorem 3, we allow the student neuron to be initialized in half-space which otherwise would not be meaningful in high-dimension cases.
> > 3. We have made efforts to work with simplest targets and notations. For example, in Section 5, we simplify the high-dimension problem to three parameters $w_x,w_y,v$ because it enjoys rotational symmetry.
> >
> > ---
> >
> > **Q4**. Presentation of Section 5 is opaque
> >
> > **A4**. We have improved the description of $\tilde{w}_\perp$ closer to yours. Actually $w$ is always in the spanned plane of $\tilde{w}$ and $w^{(0)}$ because at initialization $\tilde{w}_\perp$
> >
> > is in the spanned plane due to the definition of projection. Then all the three vectors in the second row of $\nabla_{\theta}~L$ above Eq(6) are in this spanned plane. Then by induction, we have $w$ always in the plane.
> >
> > We have added a Prop 2 to formally state where the model converges to. Please note that we don’t have $vw\rightarrow \tilde{w}$. Instead, the convergence direction $vw$ is fixed as $w_y\rightarrow 0$, and $v=w_x$ oscillates in a period-2 orbit, similar to the 2-D case.
> >
> > ---
> >
> > **Q5**. Is Theorem 5 correct?
> >
> > **A5**. Thank you for your careful review again! Here we did not include any implicit assumptions. Actually for $x^{(t)}y^{(t)}>\mu$, it takes **at least two GD steps** to go back to the next $xy>\mu$.
> >
> > With such a lower bound, we are to prove the balancing effect in two steps, as acknowledged around Eq(57-58):
> >
> > 1. First prove $\left|\frac{y^{(t+2)}-x\^{(t+2)}}{y^{(t)}-x^{(t)}}\right|<1$ although it may happen $x^{(t+2)}y^{(t+2)}<\mu$.
> > 2. To complete the corner case of the first point, it holds that, for any $x^{(k)}y^{(k)}<\mu$, we always have $\left|\frac{y^{(k+1)}-x\^{(k+1)}}{y^{(k)}-x^{(k)}}\right|<1$. In other words, as long as $xy<\mu$, the next step always has smaller $|x-y|$ until it reaches $xy>\mu$.
> >
> > The guarantee that $xy>\mu$ surely happens in some steps is because: in Eq(56), the first term is the same update rule of 1-D case by considering $x_t y_t$ as the $x^2$ in Theorem 2, and the second term is always positive if $xy<\mu$ and $x\neq y$. In other words, it is easier for $xy$ to grow beyond $\mu$ than the 1-D case, except a measure-zero set that $xy$ falls into the minima $\mu$, as we acknowledged under the remark of Theorem 5.
> >
> > ---
> >
> > We hope our explanation could address your concerns, and we look forward to hearing more from you.

---

> > > ### Comment · Reviewer_zoGM · 2022-12-05
> > > **Thank you for the rebuttal**
> > >
> > > I thank the authors for the clear rebuttal. After revisiting the proof of Theorem 5, I realized that my initial argument was wrong. Also, the presentation is massively improved in the revision. The only concern that was not adequately addressed (and I believe is a weakness of the paper) is the 0-Lebesgue measure for the step-size.
> > >
> > > All things considered, I have increased my score from 5 to 6.

---

### Official Review · Reviewer_RMoF · 2022-10-23

**Confidence:** 3
**Clarity, Quality, Novelty And Reproducibility:** 1. The presentation could be more cle…
**Correctness:** 3
**Technical Novelty And Significance:** 2
**Empirical Novelty And Significance:** 2
**Recommendation:** 5

**Strength And Weaknesses:**

Strengths: The topic is very interesting and important to the theoretical deep learning community and has not been addressed properly. The paper contains several interesting observations. The numerical experiments support the theoretical findings and are presented in a clear manner.

Weaknesses:
1. The paper misses a clear storyline. It feels that the paper presents several examples in the EoS regime but it is not clear how they are connected.
2. It is not clear how the findings or the setting under consideration is important to understand deep learning theoretically. The high-learning rate is often used in the early stages of neural network training and then the step size is decreased. It is not at all clear to me how this connects to the findings in the paper which seem rather to appear very locally, i.e. in the later stages of training.

Furthermore, I have the following list of specific comments.

1. page 4: "$x_0$ will back to $x_0$ in two steps"; Can you make this statement more formal
2. page 5, Lemma 1: "stably oscillate" has not been defined yet
3. page 5, Lemma 1: "except the $f''$"; what is meant by that?
4. page 6: $v^{(t)}=v^{(t)}$; Is this a typo?!
5. page 6; proof sketch; I cannot find where $\Delta w_y$ is defined.
6. page 7: typo: "...condition holds around (the) minimum"
7. page 7: "period-$2$ orbit $\gamma_\eta$; I cannot find the definition of a period-$2$ orbit
8. page 7: What is $Z_0$ in eq. (11)?

**Summary Of The Paper:**

This paper is concerned with gradient descent beyond the so-called Edge of Stability (EoS). That is, the step size is large than what standard theory would suggest (using the Lipschitz constant of the loss function).  The EoS is interesting to the (theoretical) deep learning community because very often neural networks are trained in this regime.

This paper studies this regime by focusing on the following three simple settings.

1. One-dimensional functions
2. Learning a single neutron
3. Non-convex matrix factorization

Several results are established. For example: In Setting 1 it is shown that there exists a step size in the EoS regime such that gradient descent converges to an 2-periodic orbit. In Setting 2 it is shown that even for large learning rate, the single neuron can be learned. In the symmetric matrix factorization setting, it is shown that the gradient descent iterates converge to a 2-periodic orbit.

Moreover, there are several numerical experiments provide which support the theoretical findings.


**Summary Of The Review:**

It is not clear how the theoretical results can amplify our current understand of the EoS regime in deep learning practice. Moreover, the paper misses a clear storyline and the quality of writing should be improved in my opinion. For this reason,  I cannot recommend acceptance.

--------------------------

I have read all the reviews and the rebuttals by the authors. While the changes made by the authors improve the paper in my opinion, I am still not convinced that the toy examples can give sufficient into the high-dimensional phenomena which appear in modern machine learning. Also the storyline still needs to be more coherent. (Although this would rather amount to a major revision and conference rebuttals are not for major revisions of papers.)

Thus, I can only increase my score to 5.

---

> ### Author Response · Authors · 2022-11-09
> **Authors' response**
>
> Thank you for your time and careful review!
>
> Following your suggestions, we add the following discussions on i) a roadmap of our theoretical results, and ii) implications from our low-dim settings to more complicated models in Appendix M and Figure 10.
>
> ---
> ---
>
> **Q1**. Missing a clear storyline. Not clear how several examples are connected
>
> **A1**. Our target is to understand the key phenomena of interest: large learning rate finds flat minima. We provide a detailed discussion along with a roadmap of our theoretical results in Appendix M and Figure 10.
>
> To understand finding flat minima, our framework provides two necessary conditions: (i) oscillations need to stabilize, and (ii) trend towards flatter minimisers. As it turns out, (i) is captured by the behavior of the function along a certain one-dimensional projection/cross-section, and this is our starting point in Theorem 1 and lemma 1, revealing the relationship of several-order derivatives. The second property (ii) requires 'overparameterization', and we formally study it in matrix factorization, with the scalar factorization case being a particular case.
>
> Besides theoretical analysis, we provide empirical evidence in Appendix B.2 for such a framework:
>
> 1. The one-dimensional cross-section along the gradient at a linear search minima near the training trajectory enjoys the 1-D condition $3[f^{(3)}]^2-f’’f^{(4)}>0$ in Theorem 1 (cf Figures 6-8(d)). The definition and settings of line search minima are around Eq(27-28) in a setting of MLPs on MNIST.
> 2. The training trajectory is approximately low-dimensional when GD is beyond EoS: the gradient and the top Hessian eigenvector are aligned well, in Figures 6-8(c).
> 3. GD beyond EoS has a trend toward the flattest minima, as shown in Figure 5(b). The learning rate decays after sufficient oscillations, and then GD converges near the flattest minima in Figure 5(c-e). The setting of this experiment is in Appendix B.2.1.
>
> Analogous to the third observation, our theoretical results in Theorem 3,4,5 can also be extended as LR decay after oscillations drives to flattest minima.
>
> ---
>
> **Q2**.Not clear how the considered findings or settings help understand deep learning. Typically high learning rate is used in earlier stages, while the current setting in later stages.
>
> **A2**.We include discussion on implications from our low-dim settings to complicated models in Appendix M.
>
> We agree that high learning rate is more often in earlier stages. Our analysis of stable oscillation is to show that sufficient oscillations drive the trajectory towards flat minima. In other words, our results can be extended by decaying the learning rate to very small (e.g., infinitesimal) after sufficient oscillations. For the 2-D in Theorem 5, single-neuron in Theorem 3 and matrix factorization in Theorem 4, it is straightforward to reach a result by starting gradient flow from the presented orbit and converging near the flattest minima. To support this argument, we also include experiments in Appendix B.2.1, where GD with LR decay converges near the flattest minima, as shown in Figure 5(c-e).
>
> ---
>
> **Q3**. $x_0$ will back to $x_0$ in two steps
>
> **A3**. Here it means GD travels through $x_0 \rightarrow x_1 \rightarrow x_0 \rightarrow \dots$. Please see our definition of period-2 stable oscillation. We also make it more formal in Theorem 1 in the revision, as you suggested.
>
> ---
>
> **Q4**. Lemma 1: stably oscillation not defined
>
> **A4**. We have added a definition of period-2 stable oscillation in Section 4 in the revision. A more general definition of stable oscillation is that GD is contained in a bounded level set but does not converge to a stationary point.
>
> ---
>
> **Q5**. Lemma 1: what is ``except $f’’$``
>
> **A5**. $f’’$ is the second order derivative of $f$. We require a high-order non-zero derivative $f^{(k)}$ with $k\ge4$ in Lemma 1, since Theorem 1 has covered $k=3$.
>
> ---
>
> **Q6**. The meaning of $v^{(t)}=v^{(t)}$
>
> **A6**.It is an intensional expression to keep $v^{(t)}$ un-changed because we are simplifying the high-dim neuron to a 2-D neuron with new notations $w_x,w_y$ introduced. Therefore, it is simplified as a three-parameter problem with $v,w_x,w_y$.
>
> ---
>
> **Q7**. Where $\Delta w_y$ is defined
>
> **A7**. We add the definition of $\Delta w_y$ in the proof sketch of Theorem 3.
>
> ---
>
> **Q8**. Period-2 orbit $\gamma_\eta$
>
> **A8**. It is defined in Eq(12) as two pairs, where each pair is two learnable matrices for $(\mathbf{Y},\mathbf{Z})$.
>
> ---
>
> **Q9**. What is $Z_0$ in eq(11)
>
> **A9**. We add the definition of $\mathbf{Y}_0, \mathbf{Z}_0$ in the statement of Theorem 4. They are rescaled forms of $\mathbf{X}_0$ to define a sharp minima.
>
> ---
> ---
>
> We hope our explanation could address your concerns, and we look forward to hearing more from you.

---

### Official Review · Reviewer_WSV5 · 2022-10-25

**Confidence:** 4
**Correctness:** 3
**Technical Novelty And Significance:** 2
**Empirical Novelty And Significance:** Not applicable
**Recommendation:** 5

**Clarity, Quality, Novelty And Reproducibility:**

Quality: The quality of the work is okay. There is some issues:
1. I have a question regarding equation (40) in the appendix. I do not think you can ignore the higher order term and eventually give a precise bound on $\eta$. If you want to give a precise bound on $\eta$ such as $\eta \mu <\sqrt{5}-1$ you need not to neglect the higher order terms. Otherwise, you will just be able to say that $\eta \mu$ can be larger than $1$ without explicitly giving the upper bound. I would like the author to address this point in the discussion.
2. Regarding Theorem 1 and Lemma 1:
- First I do not understand the notation $f^{(3)}(\bar x)/f''(\bar x) = \mathcal{O}(1)$. It seems that everything is fixed here. Why do you use an asymptotic notation?
- Second, it seems that these results say that for specific initialization there exists **one** step-size for which the optimization oscillates. It is a very weak result since this phenomenon can be very brittle (you need to find the exact step-size that corresponds to your initialization). The important question (which is addressed in Theorem 2) is: Is there a range of step-size for which the method is stable?
- Around Theorem 3 some statements are unclear:
- The connection between $\lambda_1 = \frac{\|w\|^2+ v^2}{d}$ and  $\lambda_1 = \frac{(\|w\|- v)^2 + 2 \|\tilde w\|^2}{d}$ is unclear.

3. (Minor) Around Theorem 4 many notations are missing:
- Who are $\simga_1$, $u_1$ , $v_1$
- $Y_0$ and $Z_0$ are not defined.

Clarity: The results are clearly presented. However, I feel that Theorem 1 and Lemma 1 are a distraction from the main results. (see my points about quality)
Novelty: I am not an expert in the very recent topic of optimization with step-size larger than the inverse Lipschitz constant, so it is hard for me to address precisely the novelty of this work. However, from the perspective of an expert in optimization for neural networks, these results seem quite novel.
Question: In the introduction, you mention that "We estimate our learning rate is at least 3× theirs (Wang et al., 2021)." but you do not expand on this when you present your results. Could you be more precise on that claim?
Reproducibility: These results seem reproducible.


**Strength And Weaknesses:**

## strenght:
- This paper tries to tackle a complex and important problem: explaining the experimental evidence that GD convergence for NN training even with a step-size that is too large according to the optimization literature.
- The presentation is relatively clear, and the (simple) experiments illustrate the theory.
## Weaknesses:
- The setting is simplistic. It is unclear whether such a setting captures the phenomenon that occurs in deep learning.
- Lemma 1 and Theorem 1 seem very weak results, relatively disconnected from the rest of the paper. (the key 1D result seems to be Theorem 2).


**Summary Of The Paper:**

This paper studies the dynamics of gradient descent with a large step size.  In particular, the authors are interested in the study beyond the edge of stability, i.e., the step-size regime above the inverse of the Lipschitz constant.
In that regime, local convergence is not guaranteed anymore. However, they show that under some additional assumptions on the higher derivative, one can actually show convergence to a cycle close to the optimum. They then apply the techniques to a one hidden neuron student teach problem and to a matrix factorization problem, showing that such a notion of convergence to a cycle can be obtained in that situation.


**Summary Of The Review:**

This paper is providing results regarding an important question in our field.
The quality and clarity of the paper are okay, but they could be improved.

My main concern is the significance of the results which are very toyish to a certain extent. It is not clear that these results can be related to the behavior of neural network training

---

> ### Author Response · Authors · 2022-11-09
> **Authors' response (1/2)**
>
> Thank you for your time and careful review!
>
> The following discussions on i) implications from our low-dim settings to more complicated models, and ii) a roadmap of our theoretical results are added in Appendix M and Figure 10.
>
> ---
> ---
>
> **Q1**. unclear whether such a setting captures the phenomenon that occurs in deep learning.
>
> **A1**. We would like to emphasize that, although our current simple settings are a little far from practical NNs, it still helps understand the key phenomena of interest: large learning rate finds flat minima in three steps as follows. We include more experiments in Appendix B.2 to present the following implications for complicated networks:
> 1. By Theorem 1, especially its second condition, we wish to discover an intrinsic geometric property around local minima of more complicated models. The key is to investigate the 1-D function at the cross-section of the leading eigenvector and the loss landscape.
>
>     * Theoretical: we prove the 1-D condition $3[f^{(3)}]^2-f’’f^{(4)}>0$ holds at any minima for non-trivial matrix factorization, shown as Theorem 6 in Appendix A.2.
>
>     * Empirical: we show the 1-D condition $3[f^{(3)}]^2-f’’f^{(4)}>0$ holds around minima of 3,4,5-layer ReLU MLP on MNIST, shown in Figure 6(d), 7(d), 8(d) in Appendix B.2.2.
> 2. With the above intrinsic geometric property, the next question is whether the training trajectory utilizes this property.
>
>     * Theoretical: in the case of quasi-symmetric matrix factorization, we prove that the training trajectory follows the leading eigenvector of the Hessian (i.e. the leading component of $\mathbf{X}_0$) in Theorem 4, where the only top components of weights $\mathbf{Y},\mathbf{Z}$ are changing in $\omega(\epsilon)$.
>
>     * Empirical: for MLPs on MNIST, we show the almost perfect alignment of the gradient and the top Hessian eigenvector in Figure 6(c), 7(c), 8(c).
>
> 3. The final implication is the implicit bias of EoS after such oscillation, which turns out GD is driven to flatter minima from sharper minima. This implicit bias motivates us to investigate the 2-D case, as well as single-neuron model and matrix factorization.
>     * Theoretical 1: in the 2-D case in Theorem 5, we prove the two learnable parameters $x,y$ will converge to the same values after oscillations of their product $xy$. Actually in the minimizer manifold, smaller $|x-y|$ means a flatter minimizer.
>     * Theoretical 2: in the single-neuron ReLU network in Theorem 3 and Prop 2, we show the model degenerates to the 2-D case since $w_y\rightarrow 0$. The 2-D argument tells that this nonlinear model also walks towards the balanced situation, verified with experiments in Figure 2.
>     * Theoretical 3: in the quasi-symmetric MF, although the initialization is around a sharp minima, GD is still driven towards the flattest minima where $\sigma_{\max}(\mathbf{Y})=\sigma_{\max}(\mathbf{Z})$.
>     * Empirical 1: for 2-layer 16-neuron ReLU network in a student-teacher setting, it turns out learning rate decay after beyond-EoS oscillations drives the model very close to the flattest minima, as shown in Figure 5 and in Appendix B.2.1.
>     * Empirical 2: for 3,4,5-layer MLPs on MNIST, larger learning rate drives to a flatter minima, as shown in Figure 6(b).
>
> ---
>
> **Q2**. Lemma 1 and Theorem 1 seem unclear,  weak and disconnected from the rest results
>
>
> **A2**. We have reformulated Theorem 1 and Lemma 1 to clarify our meaning. We agree that these two results are more local since they include learning rates of Lebesgue measure 0. Nevertheless, they provide conditions of local geometry at minima to allow stable oscillations, which stand as foundations for our following analysis and all these results are strictly beyond EoS. It would be impossible to discover the function of interest in Theorem 2 without Theorem 1.
>
> Meanwhile, we develop Lemma 2 and Prop 1 based on Theorem 1 to show the richness of such a family of functions. We find an extension that the regression problem naturally allows stable oscillations with a list of popular base models in Corollary 1-8. This opens a direction for the future, which is to conduct global analysis of such functions.
>
> Furthermore, such a geometric property is empirically true in Figure 6(d), 7(d), 8(d) in Appendix B.2.2., which might be proven in the future.
>
> ---
>
> **Q3**. Upper bound of $\eta\mu<\sqrt{5}-1$ in local convergence analysis around Eq(40)
>
> **A3**. Thank you for pointing this out! We have resolved this issue! Please see the revision.
>
> In the proof of Theorem 2, the functionality of the original discussion on $\eta\mu<\sqrt{5}-1$ is show $F^2_{\eta}(x)>x$ for $x\in[x_s,\bar{x}_0]$ and $F^2_\eta (x)<x$ for $x\in(\bar{x}_0,\bar{x})$ to support the global convergence. Now we reach the same conclusion using the uniqueness of $\bar{x}_0$, the continuity of $F^2_{\eta}$ and $F^2_{\eta}(x_s)>x_s$ when $\eta\mu<\sqrt{4.5}-1\approx 1.121$.

---

> > ### Author Response · Authors · 2022-11-09
> > **Authors' response (2/2)**
> >
> > **Q4**. Unclear statement around Theorem 3
> >
> > **A4**. $\lambda_1=\frac{|w|^2+v^2}{d}$ and $\lambda_1=\frac{(|w|-v)^2+2\|\tilde{w}\|}{d}$ are the same in the minimum manifold where $vw=\tilde{w}$. We mention the latter equation to emphasize that $\lambda_1$ measures the sharpness of different minimizers. Later, with $w_y\rightarrow 0$ in Theorem 3, the one-neuron ReLU model degenerates to the 2-D problem, with $v, w_x$ corresponding to $x,y$ in Theorem 5. Hence, GD beyond EoS gives $v=w_x$.
> >
> > ---
> >
> > **Q5**. Notations around Theorem 4
> >
> > **A5**. Thank you for pointing this out! We have fixed them in the revision.
> >
> > ---
> >
> > **Q6**. Comparison with Wang et al.
> >
> > **A6**. We agree there is some ambiguity here. The following arguments are following their notation.
> >
> > Our claim on ``at least 3x theirs`` is from the comparison of our 2-D result in Theorem 5 and their Theorem 3.1. Our result allows the learning rate $h$ to be $h>1/\mu$ when they require $h\le \frac{1}{3\mu}$.
> >
> > Also, there is some ambiguity about the sharpness $L$. Our setting is to set a learning rate larger than $2/L$ with the sharpness $L$ at the flattest minima, while they set $L$ to be that at initialization.
> >
> > However, it is safe to say our learning rate is strictly larger than their case. Because they define a Phase 2 where GD converges in a flat region, while in our case such a convergence never happens because $h>2/L$ for any point in the minimum manifold. Moreover, their Theorem 3.2 implicitly assumes $h\mu <1$ which is smaller than $h\mu>1$ in our setting.
> >
> > ---
> > ---
> >
> > We hope our explanation could address your concerns, and we look forward to hearing more from you!

---

> > > ### Comment · Reviewer_WSV5 · 2022-12-09
> > > **Thanks you for your answer**
> > >
> > > Dear authors,
> > >
> > > thank you for your answer, while these updates improves the overall quality of the paper, It does not change my overall recommendation.

---

### Official Review · Reviewer_Hb8V · 2022-10-27

**Confidence:** 3
**Correctness:** 2
**Technical Novelty And Significance:** 4
**Empirical Novelty And Significance:** Not applicable
**Recommendation:** 5

**Clarity, Quality, Novelty And Reproducibility:**

The writing of this paper could be improved.
Although not practical, I find the results in this paper interesting and not present in previous work.

**Strength And Weaknesses:**

Strength:

Although it has been known that GD on a function with non-uniform curvature (or specifically, step size guarantee local stability on some region but not for other region) could lead to period-2 orbit, e.g. in Ma et al. (2022), this paper provide a more detailed analysis on several specific examples.

Weakness:

1. I cannot verify the soundness of Theorem 4/14 due to confusing notation and missing definition.

The first conflict I observe is that authors define $\Delta \mathbf{Y}_t=\mathbf{Y}_t-\mathbf{Y}_0$ below eq.(218), which implies $\Delta \mathbf{Y}_0=0$, but in the statement of Theorem 4/14, authors require $u_1^{\top} \Delta \mathbf{Y}_0 v_1 \neq0$.

The second difficulty is that $\widetilde{\mathbf{Y}}_t$, which shows up multiple times in the proof, is not defined.

Maybe these problems are just due to some typos, and I am willing to re-check the proof if authors could fix it during review process.

2. The writing need to be improved.

Theorem 3 doesn't directly state anything about edge of stability. Although the oscillating behaviors is discussed later informally, it would be better to express them rigorously in a theorem.

$\sigma_1$ is used in Section 6 without definition. Although in appendix J.2, authors relate it with SVD, it is better to give an explicit definition in the main paper.

> we study non-asymptotic properties of GD beyond EoS

It is not clear to me why this paper is "beyond" EoS.

3. The EoS in this paper is established by carefully selecting a step size that slightly break the local stability at the flattest minimizer. This is not how EoS works in DNN, where progressive sharpening drives curvature up until local stability is violated. Thus it is not clear how the results in this paper could contribute to the understanding of EoS in deep learning.

4. Figure 2 right shows oscillating w_y. However, according to Theorem 3, $w_y^{(t)}$ should exponentially decay to 0. Why they are different? Is it a consequence of finite number of points instead of explicit expectation?

**Summary Of The Paper:**

This paper studies the (non)convergence of gd on general 1d function, $f(x)=\frac{1}{4}(x^2-\mu)^2$, two-layer single-neuron homogeneous network and matrix factorization, and shows a similar patten among then, i.e. GD converges to a period-2 orbit around the minimizer when step size slightly exceed the critical value for establishing local stability at the minimizer.

**Summary Of The Review:**

This paper demonstrate EoS phenomenon in several specific examples. Some confusing notations, missing definitions and unclear writing affect the quality of the article, but I believe they could be fixed by a slight revision and I am willing to re-evaluate during review process.

---

> ### Author Response · Authors · 2022-11-09
> **Authors' response**
>
> Thank you for your time and careful review!
>
> We add the following in Appendix M and Figure 10 as i) implications from our low-dim settings to more complicated models, and ii) a roadmap of our theoretical results.
>
> ---
> ---
>
> **Q1**. Soundness of Theorem 4/14 due to notations and definitions. $\sigma_1$ is not defined properly.
>
> **A1**. Thank you for pointing out these typos! We have fixed notations in the proof of Theorem 4/14 accordingly, including the initialization and $\sigma_1, \widetilde{\Delta \mathbf{Y}_t}, \widetilde{\mathbf{Y}_0}$.
>
> ---
>
> **Q2**. Theorem 3 does not directly state anything about EoS.
>
> **A2**. We add the convergence of the single-neuron model as Prop 2 in the revision. Here $K=1$ corresponds to $\eta=2/L$ where $L$ is the sharpness at the flattest minima $(w_x=1, w_y=0, v=1)$. Therefore our result with $K>1$ is strictly ‘’beyond’’ EoS. Nevertheless, in such an extreme setting, the student neuron still turns out to be aligned with the teacher neuron (i.e., $w_y\rightarrow 0$), which induces a global convergence result beyond EoS.
>
> ---
>
> **Q3**. Why is this work ‘’beyond’’ EoS?
>
> **A3**. Typically EoS states the local sharpness of hessian along the training trajectory *hovers* around $2/\eta$. In some recent work [1], the sharpness is found to oscillate above and below $2/\eta$. Such a setting is **at** EoS, where there exists flat minima in the minimum manifold and then GD converges to it from some sharper regions.
>
> But our interest is in a more extreme setting: the learning rate is **beyond** the EoS learning rate even for the flattest minima, which means GD cannot converge to any stationary point. Then we provide theoretical analysis of this in cases of 1-D, 2-D, single-neuron ReLU and matrix factorization. To our knowledge, such analysis beyond EoS was never explored before.
>
> One natural extension of such a setting ‘’beyond’’ EoS is to discuss what if the learning rate decays after a period of oscillations. In this favor, our results in Theorem 3, 4, 5 can be extended that learning rate decay (e.g., to infinitesimal) after sufficient oscillations drives the trajectory to the flattest minima. But such a guarantee cannot be reached in the analysis *at* EoS because GD converges before sufficient rounds of oscillation.
>
> [1] Self-Stabilization: The Implicit Bias of Gradient Descent at the Edge of Stability.
>
> ---
>
> **Q4**. Contribution to understanding of EoS in NNs
>
> **A4**. We agree that progressive sharpening is a significant topic of EoS. We do not include it explicitly but it is implicitly contained in the optimization process of the single-neuron model in Section 5. Then the gap between our analysis and the practice is low-dim vs high-dim. We would like to emphasize that our analysis still helps understand the key phenomena of interest: large learning rate finds flat minima. We provide a detailed discussion along with a roadmap of our theoretical results in Appendix M and Figure 10.
>
> To understand finding flat minima, our framework provides two necessary conditions: (i) oscillations need to stabilize, and (ii) trend towards flatter minimisers. As it turns out, (i) is captured by the behavior of the function along a certain one-dimensional projection/cross-section, and this is our starting point in Theorem 1 and lemma 1, revealing the relationship of several-order derivatives. The second property (ii) requires 'overparameterization', and we formally study it in matrix factorization, with the scalar factorization case being a particular case.
>
> Besides theoretical analysis, we provide empirical evidence in Appendix B.2 for such a framework:
>
> 1. The one-dimensional cross-section along the gradient at a linear search minima near the training trajectory enjoys the 1-D condition $3[f^{(3)}]^2-f’’f^{(4)}>0$ in Theorem 1 (cf Figures 6-8(d)). The definition and settings of line search minima are around Eq(27-28) in a setting of MLPs on MNIST.
> 2. The training trajectory is approximately low-dimensional when GD is beyond EoS: the gradient and the top Hessian eigenvector are aligned well, in Figures 6-8(c).
> 3. GD beyond EoS has a trend toward the flattest minima, as shown in Figure 5(b). The learning rate decays after sufficient oscillations, and then GD converges near the flattest minima in Figure 5(c-e). The setting of this experiment is in Appendix B.2.1.
>
> ---
>
>
> **Q5**. $w_y$ in experiments
>
> **A5**. Yes! It is because our analysis is on population loss while the experiment has a finite number of samples. It is a future direction to generalize the results to empirical loss.
>
> ---
> ---
>
> We hope our explanation could address your concerns, and we look forward to hearing more from you!

---

> > ### Comment · Reviewer_Hb8V · 2022-11-18
> > **Thanks for the response**
> >
> > Thanks for the new revision.
> >
> > (229) contains a typo. The position of $\alpha$ is wrong.
> >
> > I tried to recheck the proof of Theorem 14, and was stuck since eq (226).
> > This equation dropped all terms containing more than two terms of $\varepsilon_{y,t}$, $\widetilde{\Delta \mathbf{Y}_t}$ $\widetilde{\Delta \mathbf{Z}_t}$ or their product.
> > Therefore rigorously, all equation below (226) should be modified with terms $\mathcal{O}(\varepsilon\_{y,t}^2+\lVert \widetilde{\Delta \mathbf{Y}\_t} \rVert^2+\lVert \widetilde{\Delta \mathbf{X}\_t} \rVert^2)$.
> > More critically, k in (230) is no longer an invariant.
> >
> > Although author "conclude that both $\lVert \widetilde{\Delta \mathbf{Y}_t} \rVert$, $\lVert \widetilde{\Delta \mathbf{Z}_t} \rVert$ stay small." at end of Phase I, I don't think the analysis is rigorous in its current form and we need to be more cautious in order to avoid circular reasoning.
> >
> > Other questions:
> > How (234) and (235) is derived?
> > The left part of inner product in (238) seems wrong. Does a transpose fix it?

---

### Author Response · Authors · 2022-11-09
**Update in the revision**

Dear AC and Reviewers,

Thank you for your attention to this work!

---

We would like to list the following updates in the uploaded revision. All key updates are in blue color.

1. Add: **Figure 10 to illustrate the coherent connections of proven results** in Appendix M.1
2. Add: discussion on **connections of our theoretical results** in Appendix M.1
3. Add: discussion on **implications from our low-dim settings to more complicated networks** in Appendix M.2, with **low-dim theorems enhanced with high-dim experiments**
4. Add: Prop 1 to show a composition rule of two base models to find a more complicated model that allows stable oscillation in Section 4
5. Add: Prop 2 to formally state the convergence of single-neuron model in Section 5, as suggested by Reviewer Hb8v
6. Add: Prop 3 to formally state the convergence of 2-D case in Appendix A.1
7. Fix: reformulate Theorem 1 and Lemma 1 as suggested by Reviewer zoGM
8. Fix: proof of Theorem 2 as suggested by Reviewer WSV5
9. Add and fix: all missing or ambiguous notations and descriptions as pointed out by Reviewers
10. Add: remarks on why our results reveal GD walks towards the flattest minima under Theorem 3,4,5
11. Reshape: the section Conclusions is moved to Appendix due to space limit

---

### Decision · Program_Chairs · 2023-01-20

**Decision:**

Reject

**Justification For Why Not Higher Score:**

All reviewers agree this is a clear reject.

**Justification For Why Not Lower Score:**

N/A

**Metareview: Summary, Strengths And Weaknesses:**

This paper studies the Edge of Stability (EoS) phenomenon, which has recently attracted a lot of attention in the community. Specifically, the paper focuses on three different problems: one-dimensional functions, learning a single neutron, and non-convex matrix factorization.

The major common concern of the reviewers is the clear lack of a storyline. Instead, the paper presents a set of somewhat disconnected results. It's hard to draw of consistent conclusion and the impact on neural networks is rather weak and not well justified.

On the plus side, the authors are tackling an interesting problem but the paper is currently not ready for publication, all reviewers agree on this point. I strongly encourage the authors to rethink the overall presentation of the paper, making the paper more accessible and above all try to present a coherent story.


**Summary Of Ac-Reviewer Meeting:**

I initiated a discussion in which no reviewer participated (also sent emails to the reviewers). I, therefore, took a decision based on my own reading of the paper and reviews (which I believe to lean towards rejection).